# Evaluating Robustness of Predictive Uncertainty Estimation: Are Dirichlet-based Models Reliable?

## Abstract

Robustness to adversarial perturbations and accurate uncertainty estimation are crucial for reliable application of deep learning in real world settings. Dirichlet-based uncertainty (DBU) models are a family of models that predict the parameters of a Dirichlet distribution (instead of a categorical one) and promise to signal when *not* to trust their predictions. Untrustworthy predictions are obtained on unknown or ambiguous samples and marked with a high uncertainty by the models.

In this work, we show that DBU models with standard training are not robust w.r.t. three important tasks in the field of uncertainty estimation. First, we evaluate how useful the uncertainty estimates are to **(1)** indicate correctly classified samples. Our results show that while they are a good indicator on unperturbed data, performance on perturbed data decreases dramatically. **(2)** We evaluate if uncertainty estimates are able to detect adversarial examples that try to fool classification. It turns out that uncertainty estimates are able to detect FGSM attacks but not able to detect PGD attacks. We further evaluate the reliability of DBU models on the task of **(3)** distinguishing between in-distribution (ID) and out-of-distribution (OOD) data. To this end, we present the first study of *certifiable* robustness for DBU models. Furthermore, we propose novel *uncertainty attacks* that fool models into assigning high confidence to OOD data and low confidence to ID data, respectively. Both approaches show that detecting OOD samples and distinguishing between ID-data and OOD-data is not robust.

Based on our results, we explore the first approaches to make DBU models more robust. We use adversarial training procedures based on label attacks, uncertainty attacks, or random noise and demonstrate how they affect robustness of DBU models on ID data and OOD data.

## 1 Introduction

Neural networks achieve high predictive accuracy in many tasks, but they are known to have two substantial weaknesses: First, neural networks are not robust against adversarial perturbations, i.e., semantically meaningless input changes that lead to wrong predictions (Szegedy et al., 2014; Goodfellow et al., 2015). Second, neural networks tend to make over-confident predictions at test time (Lakshminarayanan et al., 2017). Even worse, standard neural networks are unable to identify samples that are different from the samples they were trained on. In these cases, they provide uninformed decisions instead of abstaining. These two weaknesses make them impracticable in sensitive domains like financial, autonomous driving or medical areas which require trust in predictions.

To increase trust in neural networks, models that provide predictions along with the corresponding uncertainty have been proposed. There are three main families of models that aim to provide meaningful estimates of their predictive uncertainty. The first family are Bayesian Neural Networks (Blundell et al., 2015; Osawa et al., 2019; Maddox et al., 2019), which have the drawback that they are computationally demanding. The second family consists of Monte-Carlo drop-out based models (Gal & Ghahramani, 2016) and ensembles (Lakshminarayanan et al., 2017) that estimate uncertainty by computing statistics such as mean and variance by aggregating forward passes of multiple models. A disadvantage of all of these models is that uncertainty estimation at inference time is expensive.

In contrast to these, the recently growing family of Dirichlet-based uncertainty (DBU) models (Malinin & Gales, 2018a; 2019; Sensoy et al., 2018; Malinin et al., 2019; Charpentier et al., 2020) directly predict the parameters of a Dirichlet distribution over categorical probability distributions. They provide efficient uncertainty estimates at test time since they only require a single forward pass.

DBU models bring the benefit of providing both, aleatoric and epistemic uncertainty estimates. Aleatoric uncertainty is irreducible and caused by the natural complexity of the data, such as class overlap or noise. Epistemic uncertainty results from the lack of knowledge about unseen data, e.g. when the model is presented an image of an unknown object. Both uncertainty types can be quantified using different uncertainty measures based on a Dirichlet distribution, such as differential entropy, mutual information, or pseudo-counts (Malinin & Gales, 2018a; Charpentier et al., 2020). These uncertainty measures have been shown outstanding performance in, e.g., the detection of OOD samples and thus are superior to softmax based confidence (Malinin & Gales, 2019; Charpentier et al., 2020).

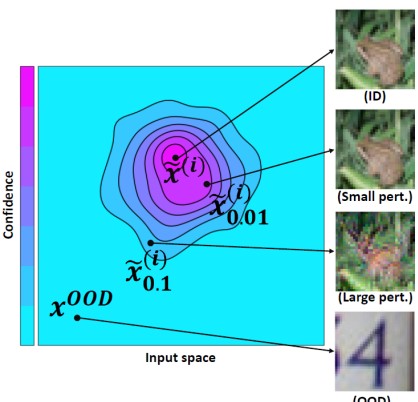

Figure 1: Visualization of the desired uncertainty estimates.

Neural networks from the families outlined above are expected to *know what they don't know*, i.e., notice when they are unsure about a prediction. This raises questions with regards to adversarial examples: should uncertainty estimates *detect* these corrupted samples and abstain from making a prediction (i.e. indicated by high uncertainty in the prediction), or should they be *robust* to adversarial examples and produce the correct output even under perturbations? Using humans as the gold standard of image classification and assuming that the perturbations are semantically meaningless, which is typically implied by small $L_p$ norm of the corruption, we argue that the best option is that the models are robust to adversarial perturbations (see Figure 1). Beyond being robust w.r.t. label prediction, we expect models to robustly know what they do not know. That is, they should robustly distinguish between ID and OOD data even if those are perturbed.

In this work, we focus on DBU models and analyze their robustness capacity w.r.t. the classification decision and uncertainty estimations, going beyond simple softmax output confidence by investigating advanced measures like differential entropy. Specifically, we study the following questions:

1. *Is high certainty a reliable indicator of correct predictions?*
2. *Can we use uncertainty estimates to detect label attacks on the classification decision?*
3. *Are uncertainty estimates such as differential entropy a robust feature for OOD detection?*

In addressing these questions we place particular focus on adversarial perturbations of the input in order to evaluate the *worst case* performance of the models. We address question one by analyzing uncertainty estimation on correctly and wrongly labeled samples, without and with adversarial perturbations on the inputs. To answer question two, we study uncertainty estimates of DBU models on *label attacks*. More specifically, we analyze whether there is a difference between uncertainty estimates on perturbed and unperturbed inputs and whether DBU models are capable of recognizing successful label attacks by uncertainty estimation. Addressing question three, we use robustness verification based on randomized smoothing and propose to investigate *uncertainty attacks*. Uncertainty attacks aim at changing the uncertainty estimate such that ID data is marked as OOD data and vice versa. Finally, we propose robust training procedures that use label attacks, uncertainty attacks or random noise and analyze how they affect robustness of DBU models on ID data and OOD data.

## 2 RELATED WORK

Recently, multiple works have analyzed uncertainty estimation and robustness of neural networks. (Snoek et al., 2019) compares uncertainty estimates of models based on drop-out and ensembles under data set shifts. (Cardelli et al., 2019; Wicker et al., 2020) study probabilistic safety of Bayesian networks under adversarial perturbations by analyzing inputs sets and the corresponding mappings

in the output space. In contrast, our work focus on DBU models and analyze their robustness w.r.t. adversarial perturbations specifically designed to fool label or uncertainty predictions of the models. Furthermore, previous works on attack defenses have focused on evaluating either *robustness w.r.t. class predictions* (Carlini & Wagner, 2017; Weng et al., 2018) or *label attack detection* (Carlini & Wagner, 2017). In contrast, our work jointly evaluates both tasks by analyzing them from the uncertainty perspective. Furthermore, in addition to label attacks, we study a new type of adversarial perturbations that directly target uncertainty estimation. Those attacks are different from traditional label attacks (Madry et al., 2018; Dang-Nhu et al., 2020).

Different models have been proposed to account for uncertainty while being robust. (Smith & Gal, 2018) and (Lee et al., 2018) have tried to improve label attack detection based on uncertainty using drop-out or density estimation. In addition from improving label attack detection for large unseen perturbations, (Stutz et al., 2020) aimed at improving robustness w.r.t. class label predictions on small input perturbations. To this end, they proposed a new adversarial training with softer labels for adversarial samples further from the original input. (Qin et al., 2020) suggested a similar adversarial training where labels are soften differently depending on the input robustness. These previous works only consider the aleatoric uncertainty contained in the predicted categorical probabilities, i.e. the softmax output. They do not consider DBU models which explicitly account for both aleatoric and epistemic uncertainty. (Malinin & Gales, 2019) proposed to improve a single type of DBU model for label attack detection by assigning them high uncertainty during training.

Please note that the works (Tagasovska & Lopez-Paz, 2019; Kumar et al., 2020; Bitterwolf et al., 2020; Meinke & Hein, 2020) study a different orthogonal problem. (Tagasovska & Lopez-Paz, 2019) propose to compute confidence intervals while (Kumar et al., 2020) propose certificates on softmax predictions. (Bitterwolf et al., 2020) uses interval bound propagation to compute bounds on softmax predictions in the $L_\infty$-ball around an OOD point and for ReLU networks, (Meinke & Hein, 2020) proposes an approach to obtain certifiably low confidence for OOD data. These four studies estimate confidence based on softmax predictions, which accounts for aleatoric uncertainty only. In this paper, we provide certificates on the OOD classification task using DBU models directly which is better suited to epistemic uncertainty measures.

## 3 DIRICHLET-BASED UNCERTAINTY MODELS

Standard (softmax) neural networks predict the parameters of a categorical distribution $\boldsymbol{p}^{(i)} = [p_1^{(i)}, \ldots, p_C^{(i)}]$ for a given input $\boldsymbol{x}^{(i)} \in \mathbb{R}^d$, where $C$ is the number of classes. Given the parameters of a categorical distribution, we can evaluate its *aleatoric uncertainty*, which is the uncertainty on the class label prediction $y^{(i)} \in \{1, \ldots, C\}$. For example, when predicting the result of an unbiased coin flip, we expect the model to have high aleatoric uncertainty and predict $p(\text{head}) = 0.5$.

In contrast to standard (softmax) neural networks, DBU models predict the parameters of a Dirichlet distribution – the natural prior of categorical distributions – given input $\boldsymbol{x}^{(i)}$ (i.e. $q^{(i)} = \text{Dir}(\boldsymbol{\alpha}^{(i)})$ where $f_\theta(\boldsymbol{x}^{(i)}) = \boldsymbol{\alpha}^{(i)} \in \mathbb{R}_+^C$). Hence, the *epistemic distribution* $q^{(i)}$ expresses the *epistemic* uncertainty on $\boldsymbol{x}^{(i)}$, i.e. the uncertainty on the categorical distribution prediction $\boldsymbol{p}^{(i)}$. From the epistemic distribution, follows an estimate of the *aleatoric distribution* of the class label prediction $\text{Cat}(\bar{\boldsymbol{p}}^{(i)})$ where $\mathbb{E}_{q^{(i)}}[\boldsymbol{p}^{(i)}] = \bar{\boldsymbol{p}}^{(i)}$. An advantage of DBU models is that one pass through the neural network is sufficient to compute epistemic distribution, aleatoric distribution, and predict the class label:

$$q^{(i)} = \text{Dir}(\boldsymbol{\alpha}^{(i)}), \quad \bar{p}_c^{(i)} = \frac{\alpha_c^{(i)}}{\alpha_0^{(i)}} \text{ with } \alpha_0^{(i)} = \sum_{c=1}^{C} \alpha_c^{(i)}, \quad y^{(i)} = \arg\max [\bar{p}_1^{(i)}, ..., \bar{p}_C^{(i)}] \quad (1)$$

This parametrization allow to compute classic uncertainty measures in closed-form. As an example, the concentration parameters $\alpha_c^{(i)}$ can be interpreted as a pseudo-count of observed samples of class $c$ and, thus, are a good indicator of epistemic uncertainty. Note that further measures, such as differential entropy of the Dirichlet distribution (see Equation 2, where $\Gamma$ is the Gamma function and $\Psi$ is the Digamma function) or the mutual information between the label $y^{(i)}$ and the categorical $\boldsymbol{p}^{(i)}$ can also be computed in closed-form (App. A.2, (Malinin & Gales, 2018a)). Hence, DBU models can efficiently use these measures to assign high uncertainty for unknown data making them specifically suited for detection of OOD samples like anomalies.

$$m_{\text{diffE}} = \sum_c^K \ln \Gamma(\alpha_c) - \ln \Gamma(\alpha_0) - \sum_c^K (\alpha_c - 1) \cdot (\Psi(\alpha_c) - \Psi(\alpha_0)) \qquad (2)$$

Several recently proposed models for uncertainty estimations belong to the family of DBU models, such as PriorNet, EvNet, DDNet and PostNet. These models differ in terms of their parametrization of the Dirichlet distribution, the training, and density estimation. An overview of theses differences is provided in Table 1. We evaluate all recent versions of these models in our study.

Table 1: Summary of DBU models. Further details on the loss functions are provided in the appendix.

|  | $\alpha^{(i)}$-parametrization | Loss | OOD training data | Ensemble training | Density estimation |
|---|---|---|---|---|---|
| **PostNet** | $f_\theta(\boldsymbol{x}^{(i)}) = \boldsymbol{1} + \boldsymbol{\alpha}^{(i)}$ | Bayesian loss | No | No | Yes |
| **PriorNet** | $f_\theta(\boldsymbol{x}^{(i)}) = \boldsymbol{\alpha}^{(i)}$ | Reverse KL | Yes | No | No |
| **DDNet** | $f_\theta(\boldsymbol{x}^{(i)}) = \boldsymbol{\alpha}^{(i)}$ | Dir. Likelihood | No | Yes | No |
| **EvNet** | $f_\theta(\boldsymbol{x}^{(i)}) = \boldsymbol{1} + \boldsymbol{\alpha}^{(i)}$ | Expected MSE | No | No | No |

Contrary to the other models, Prior Networks (**PriorNet**) (Malinin & Gales, 2018a; 2019) requires OOD data for training to "teach" the neural network the difference between ID and OOD data. PriorNet is trained with a loss function consisting of two KL-divergence terms. The fist term is designed to learn Dirichlet parameters for ID, while the second one is used to learn a flat Dirichlet distribution ($\boldsymbol{\alpha} = \boldsymbol{1}$) for OOD data. There a two variants of PriorNet. The first one is trained based on reverse KL-divergence (Malinin & Gales, 2019), while the second one is trained with KL-divergence (Malinin & Gales, 2018a). We include in our experiment the most recent reverse version of PriorNet, as it shows superior performance (Malinin & Gales, 2019).

Evidential Networks (**EvNet**) (Sensoy et al., 2018) are trained with a loss that computes the sum of squares between the on-hot encoded true label $\boldsymbol{y}*^{(i)}$ and the predicted categorical $\boldsymbol{p}^{(i)}$ under the Dirichlet distribution. Ensemble Distribution Distillation (**DDNet**) (Malinin et al., 2019) is trained in two steps. First, an ensemble of $M$ classic neural networks needs to be trained. Then, the soft-labels $\{\boldsymbol{p}_m^{(i)}\}_{m=1}^M$ provided by the ensemble of networks are distilled into a Dirichlet-based network by fitting them with the maximum likelihood under the Dirichlet distribution. Posterior Network (**PostNet**) (Charpentier et al., 2020) performs density estimation for ID data with normalizing flows and uses a Bayesian loss formulation. Note that EvNet and PostNet model the Dirichlet parameters as $f_\theta(\boldsymbol{x}^{(i)}) = 1 + \boldsymbol{\alpha}^{(i)}$ while PriorNet, RevPriorNet and DDNet compute them as $f_\theta(\boldsymbol{x}^{(i)}) = \boldsymbol{\alpha}^{(i)}$.

## 4 ROBUSTNESS OF DIRICHLET-BASED UNCERTAINTY MODELS

We analyze robustness of DBU models in the field of uncertainty estimation w.r.t. the following four aspects: *accuracy*, *confidence calibration*, *label attack detection* and *OOD detection*. Uncertainty is quantified by differential entropy, mutual information and pseudo counts. A formal definition of all uncertainty estimation measures is provided in the appendix. Robustness of Dirichlet-based uncertainty models is evaluated based on *label attacks* and a newly proposed type of attacks called *uncertainty attacks*. While label attacks aim at changing the predicted class, uncertainty attacks aim at changing uncertainty assigned to a prediction. All existing works are based on label attacks and focus on robustness w.r.t. the classification decision. Thus, we are the first to propose attacks targeting uncertainty estimates such as differential entropy and analyze further desirable robustness properties of DBU models. Both attack types compute a perturbed input $\tilde{\boldsymbol{x}}^{(i)}$ close to the original input $\boldsymbol{x}^{(i)}$ i.e. $||\boldsymbol{x}^{(i)} - \tilde{\boldsymbol{x}}^{(i)}||_2 < r$ where $r$ is the attack radius. The perturbed input is obtained by optimizing a loss function $l(\boldsymbol{x})$ using Fast Gradient Sign Method (FGSM) or Projected Gradient Descent (PGD). We use also a black box attack (Noise) which generates 10 Noise samples from a Gaussian distribution with mean equal to the original sample. The pertrubed sample which fools the most the loss function is selected as an attack. To complement attacks, we propose the first study of certifiable robustness for DBU models, which is based on randomized smoothing (Cohen et al., 2019).

The following questions we address by our experiments have a common assessment metric. Distinguishing between correctly and wrongly classified samples, between non-attacked input and attacked inputs or between ID data and OOD data can be treated as binary classification problems. To quantify the performance of the models on these binary classification problems, we compute AUC-PR.

Experiments are performed on two image data sets (MNIST (LeCun & Cortes, 2010) and CIFAR10 (Krizhevsky et al., 2009)), which contain bounded inputs and two tabular data sets (Segment (Dua & Graff, 2017) and Sensorless drive (Dua & Graff, 2017)), consisting of unbounded inputs. Note that unbounded inputs are challenging since it is impossible to describe the infinitely large OOD distribution. As PriorNet requires OOD training data, we use two further image data sets (FashionMNIST (Xiao et al., 2017) and CIFAR100 (Krizhevsky et al., 2009)) for training on MNIST and CIFAR10, respectively. All other models are trained without OOD data. To obtain OOD data for the tabular data sets, we remove classes from the ID data set (class window for the Segment data set and class 9 for Sensorless drive) and use them as the OOD data. See appendix for further details on the setup.

## 4.1 Uncertainty estimation under label attacks

Label attacks aim at changing the predicted class. To obtain a perturbed input with a different label, we maximize the cross-entropy loss $\tilde{\boldsymbol{x}}^{(i)} \approx \arg\max_{\boldsymbol{x}} l(\boldsymbol{x}) = \text{CE}(\boldsymbol{p}^{(i)}, \boldsymbol{y}^{(i)})$ under the radius constraint. For the sake of completeness we also analyze label attacks regarding their performance to change class predictions and report their accuracy to show the effectiveness based on different radii (see Appendix, Table 7). As expected and partially shown by previous works, none of the DBU models is robust against label attacks. However, we noted that PriorNet is slightly more robust than the other models. This might be explained by the use of OOD data during training, which can be seen as some kind of robust training. From now on, we switch to the core focus of this work and analyze robustness properties of uncertainty estimation.

**Is high certainty a reliable indicator of correct predictions?**
*Expected behavior:* Predictions with high certainty are more likely to be correct than low certainty predictions. *Assessment metric:* We distinguish between correctly classified samples (label 0) and wrongly classified ones (label 1) based on the differential entropy scores produced by the DBU models (Malinin & Gales, 2018a). Correctly classified samples are expected to have low differential entropy, reflecting the model's confidence, and analogously that wrongly predicted samples tend to have higher differential entropy. *Observed behavior:* Note that the positive and negative classes are not balanced, thus, the use of AUC-PR scores (Saito & Rehmsmeier, 2015) are important to enable meaningful measures. While uncertainty estimates are indeed an indicator of correctly classified samples on non-perturbed data, none of the models maintains its high performance on perturbed data (see. Table 2). Thus, using uncertainty estimates as indicator for correctly labeled inputs is not robust to adversarial perturbations, although the used attacks do not target uncertainty.

Table 2: Certainty based on differential entropy under PGD label attacks (AUC-PR).

| | CIFAR10 | | | | | | | | Sensorless | | | | | |
|---|---|---|---|---|---|---|---|---|---|---|---|---|---|---|
| Att. Rad. | 0.0 | 0.1 | 0.2 | 0.5 | 1.0 | 2.0 | 4.0 | | 0.0 | 0.1 | 0.2 | 0.5 | 1.0 | 2.0 | 4.0 |
| PostNet | **98.7** | 88.6 | 56.2 | 7.8 | 1.2 | 0.4 | 0.3 | | 99.7 | 8.3 | 3.9 | 3.6 | **7.0** | **9.8** | **11.3** |
| PriorNet | 92.9 | 77.7 | 60.5 | **37.6** | **24.9** | **11.3** | **3.0** | | 99.8 | 10.5 | 3.2 | 0.7 | 0.2 | 0.2 | 2.2 |
| DDNet | 97.6 | **91.8** | **78.3** | 18.1 | 0.8 | 0.0 | 0.0 | | 99.7 | 11.9 | 1.6 | 0.4 | 0.2 | 0.1 | 0.2 |
| EvNet | 97.9 | 85.9 | 57.2 | 10.2 | 4.0 | 2.4 | 0.3 | | **99.9** | **22.9** | **13.0** | **6.0** | 3.7 | 3.2 | 3.1 |

**Can we use uncertainty estimates to detect label attacks on the classification decision?**
*Expected behavior:* Adversarial examples are not from the natural data distribution. Therefore, DBU models are expected to detect them as OOD data by assigning them a higher uncertainty. We expect perturbations with larger attack radius $r$ to be easier to detect as they differ more significantly from the data distribution. *Assessment metric:* The goal of attack-detection is to distinguish between unperturbed samples (label 0) and perturbed samples (label 1). To quantify the performance, we use the differential entropy (Malinin & Gales, 2018a). Non-perturbed samples are expected to have low differential entropy, reflecting the fact that they are from the distribution the models were trained on, while perturbed samples are expected to have a high differential entropy. Further results based on other uncertainty measures are provided in the appendix. *Observed behavior:* Table 7 shows that the accuracy of all models decreases significantly under PGD label attacks, but none of the models is able to provide an equivalently increasing high attack detection rate (see Table 3). Even larger perturbations are hard to detect for DBU models. Although PGD label attacks do not explicitly consider uncertainty, they seem to provide adversarial examples with similar uncertainty as the original input. Such high certainty adversarial examples are illustrated in Figure 2, where certainty is visualized based on the precision $\alpha_0$ that is supposed to be high for ID data and low for OOD

data. While the original input (perturbation size 0.0) is correctly classified as frog and ID data, there exist adversarial examples that are classified as deer or bird. The certainty on the prediction of these adversarial examples has a similar or even higher value than the prediction on the original input. Using the differential entropy to distinguish between ID and OOD data results in the same ID/OOD assignment since the differential entropy of the three right-most adversarial examples is similar or even smaller than on the unperturbed input.

For the less powerful FGSM and Noise attacks (see Appendix), DBU models achieve mostly better attack detection rates than for PGD attacks. This suggests that uncertainty estimation is able to detect weak attacks, which is consistent with the observations in (Malinin & Gales, 2018b). Furthermore, PostNet provides better label attack detection rate for large perturbations on tabular data sets. An explanation for this observation is that the density estimation of the ID samples has been shown to work better for tabular data sets (Charpentier et al., 2020). Standard adversarial training (based on label attacks targeting the crossentropy loss function) improves robustness w.r.t. class predictions (see Appendix, Table 32), but does not improve label attack detection performance of any model (see Table 40). Overall, none of the DBU models provides a reliable indicator for adversarial inputs that target the classification decision.

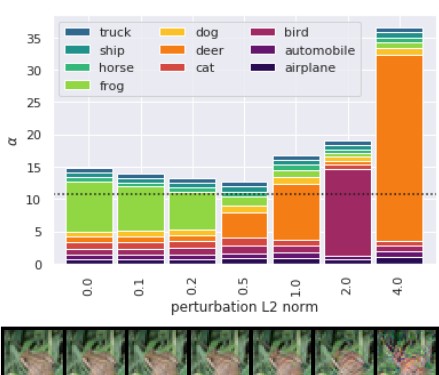

Figure 2: Input & corr. Dir.-parameters under label attacks (dotted: threshold to distinguish ID and OOD).

Table 3: Label Attack-Detection by normally trained DBU models based on differential entropy under PGD label attacks (AUC-PR).

|  | CIFAR10 | | | | | | | Sensorless | | | | | |
| --- | --- | --- | --- | --- | --- | --- | --- | --- | --- | --- | --- | --- | --- |
| Att. Rad. | 0.1 | 0.2 | 0.5 | 1.0 | 2.0 | 4.0 | | 0.1 | 0.2 | 0.5 | 1.0 | 2.0 | 4.0 |
| PostNet | **63.4** | **66.9** | 42.1 | 32.9 | 31.6 | 31.2 | | 47.7 | 42.3 | 36.9 | **48.5** | **85.0** | **99.0** |
| PriorNet | 53.3 | 56.0 | 55.6 | **49.2** | 42.2 | 35.4 | | 38.8 | 33.6 | 31.4 | 33.1 | 40.9 | 53.5 |
| DDNet | 55.8 | 60.5 | **57.3** | 38.7 | 32.3 | 31.4 | | **53.5** | 42.2 | 35.0 | 32.8 | 32.6 | 33.9 |
| EvNet | 48.4 | 46.9 | 46.3 | 46.3 | **44.5** | **42.5** | | 48.2 | **42.6** | **38.2** | 36.0 | 37.2 | 41.7 |

## 4.2 ATTACKING UNCERTAINTY ESTIMATION

DBU models are designed to provide uncertainty estimates (beyond softmax based confidence) alongside predictions and use this predictive uncertainty for OOD detection. Thus, in this section we focus on attacking these uncertainty estimates. We present result for attacks based on the differential entropy as loss function ($\tilde{\boldsymbol{x}}^{(i)} \approx \arg\max_{\boldsymbol{x}} l(\boldsymbol{x}) = \text{Diff-Ent}(\text{Dir}(\alpha^{(i)}))$), since it is the most widely used metric for ID-OOD-differentiation. Result based on further uncertainty measures, loss functions and details on the uncertainty attacks are provided in the appendix. Regarding uncertainty attacks, we analyze model performance w.r.t. two tasks. First, attacks are computed on ID data to transform them in OOD data, while OOD data is left non-attacked. Second, we attack OOD data to transform it into ID data, while ID data is not attacked. Hence, uncertainty attacks aim at posing ID data as OOD data or conversely.

**Are uncertainty estimates a robust feature for OOD detection?**
*Expected behavior:* We expect Dirichlet-based uncertainty models to be able to distinguish between ID and OOD data by providing reliable uncertainty estimates, even under small perturbations. That is, we expect the uncertainty estimates of DBU models to be robust under attacks. *Assessment metric:* We distinguish between ID data (label 0) and OOD data (label 1) based on the differential entropy as uncertainty scoring function (Malinin & Gales, 2018a). Differential entropy is expected to be small on ID samples and high on OOD samples. Experiments on further uncertainty measure and results for AUROC are provided in the appendix. *Observed behavior:* OOD samples are perturbed as illustrated in Figure 3. The left part shows an OOD-sample, which is identified as OOD. Adding adversarial perturbations $\geq 0.5$ to it changes the Dirichlet parameters such that the resulting images are identified as ID, based on precision or differential entropy as uncertainty measure. Adding adversarial perturba-

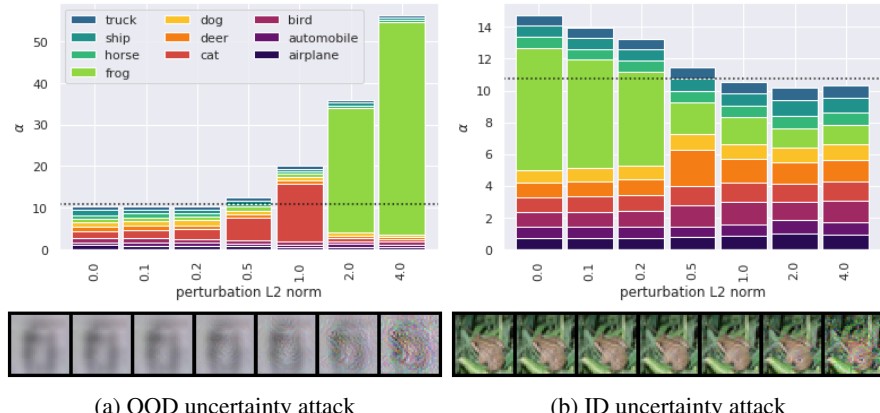

(a) OOD uncertainty attack       (b) ID uncertainty attack

Figure 3: ID and OOD input with corresponding Dirichlet-parameters under uncertainty attacks (dotted line: threshold to distinguish ID and OOD).

tions to an ID sample (right part) results in images identified as OOD. OOD detection performance of all DBU models decreases rapidly with the size of the perturbation, regardless of whether attacks are computed on ID or OOD data (Table 4). Thus, using uncertainty estimation to distinguish between ID and OOD data is not robust. PostNet and DDNet achieve slightly better performance than the other models. Further, PostNet provides better scores for large perturbations on tabular data sets which could again be explained by its density-based approach.

Table 4: OOD detection based on differential entropy under PGD uncertainty attacks against differential entropy on ID data and OOD data (AUC-PR).

| | ID-Attack (non-attacked OOD) | | | | | | | | OOD-Attack (non-attacked ID) | | | | | | |
| Att. Rad. | 0.0 | 0.1 | 0.2 | 0.5 | 1.0 | 2.0 | 4.0 | | 0.0 | 0.1 | 0.2 | 0.5 | 1.0 | 2.0 | 4.0 |
|---|---|---|---|---|---|---|---|---|---|---|---|---|---|---|---|
| | | | | | | | **CIFAR10 – SVHN** | | | | | | | | |
| PostNet | 81.8 | 64.3 | 47.2 | 22.4 | 17.6 | **16.9** | **16.4** | | 81.8 | 60.5 | 40.7 | 23.3 | 21.8 | 19.8 | 18.1 |
| PriorNet | 54.4 | 40.1 | 30.0 | 17.9 | 15.6 | 15.4 | 15.4 | | 54.4 | 40.7 | 30.7 | 19.5 | 16.5 | 15.7 | 15.4 |
| DDNet | **82.8** | **71.4** | **59.2** | **28.9** | 16.0 | 15.4 | 15.4 | | **82.8** | **72.0** | **57.2** | 20.8 | 15.6 | 15.4 | 15.4 |
| EvNet | 80.3 | 62.4 | 45.4 | 21.7 | **17.9** | 16.5 | 15.6 | | 80.3 | 58.2 | 46.5 | **34.6** | **28.0** | **23.9** | **21.0** |
| | | | | | | | **Sens. – Sens. class 10, 11** | | | | | | | | |
| PostNet | **74.5** | **39.8** | **36.1** | **36.0** | **45.9** | **46.0** | **46.0** | | **74.5** | **43.3** | **42.0** | **32.1** | **35.1** | **82.6** | **99.4** |
| PriorNet | 32.3 | 26.6 | 26.5 | 26.5 | 26.6 | 28.3 | 38.6 | | 32.3 | 26.7 | 26.6 | 26.6 | 27.0 | 30.4 | 36.8 |
| DDNet | 31.7 | 26.8 | 26.6 | 26.5 | 26.6 | 27.1 | 30.5 | | 31.7 | 27.1 | 26.7 | 26.7 | 26.8 | 26.9 | 27.3 |
| EvNet | 66.5 | 30.5 | 28.2 | 27.1 | 28.1 | 31.8 | 37.5 | | 66.5 | 38.7 | 36.1 | 30.2 | 28.2 | 28.8 | 32.2 |

### 4.3 ROBUST TRAINING FOR DBU MODELS & ID/OOD VERIFICATION

Our robustness analysis based on label attacks and uncertainty attacks shows that neither the predicted class, nor the uncertainty corresponding to a prediction, nor the differentiation between ID and OOD-data is robust. Thus, we propose adversarial training procedures to enhance robustness. During training we augment the data set by samples computed based on (i) PGD attacks against the crossentropy loss or (ii) against the differential entropy function, which is used to distinguish between ID and OOD data, or (iii) by adding random noise as proposed for randomized smoothing training.

Since attacks are used during robust training, we want to avoid tying robustness evaluation to gradient based attacks. Instead, we propose the first approach that certifies robustness of DBU models based on randomized smoothing (Cohen et al., 2019). Randomized smoothing was proposed to verify robustness w.r.t. class predictions and we modify it for ID/OOD-verification. As randomized smoothing treats classifiers as a black-box, we transform distinguishing between ID data (label 0) and OOD data (label 1) into a binary classification problem based on an uncertainty measure, which requires to set a threshold for the uncertainty measure to obtain an actual decision boundary. This is in contrast to our attack-based experiments where we avoided setting thresholds by analyzing area under the curve metrics. Thresholds for uncertainty measure are set for each model individually based on the validation set, such that the accuracy w.r.t. to ID/OOD-assignment of the model is maximized.

In the following we discuss results for ID/OOD-verification based on differential entropy on CIFAR10 (ID data) and SVHN (OOD data). Further results on other data sets, other uncertainty measures and results on the standard classification based randomized smoothing verification are shown in the appendix. Table 5 shows the percentage of samples which are correctly identified as ID (resp. OOD) data and are certifiably robust within this type (cc; certified correct) along with the corresponding mean certified radius. The higher the portion of cc samples and the larger the radius the more robust is ID/OOD-distinguishing w.r.t. the corresponding perturbation size $\sigma$.[1]

Table 5: Randomized smoothing verification for different $\sigma$ of CIFAR10 (ID data) and SVHN (OOD data). Left part: percentage of samples that is *correctly* identified and certified as ID data (cc) and corresponding mean certified radius (R). Right part: same for OOD data.

| | ID-Verification | | | | | | OOD-Verification | | | | | |
| | 0.1 | | 0.2 | | 0.5 | | 0.1 | | 0.2 | | 0.5 | |
| $\sigma$ | cc | R | cc | R | cc | R | cc | R | cc | R | cc | R |
|---|---|---|---|---|---|---|---|---|---|---|---|---|
| | | | | | | adv. train. loss: None | | | | | | |
| PriorNet | **83.2** | **0.26** | **97.8** | **0.58** | **100.0** | **1.47** | 3.7 | 0.10 | 0.0 | 0.00 | 0.0 | 0.00 |
| PostNet | 23.6 | 0.17 | 22.2 | 0.11 | 0.0 | 0.00 | **99.3** | **0.23** | **99.2** | **0.29** | **100.0** | **1.37** |
| DDNet | 63.7 | 0.24 | 88.7 | 0.50 | 53.0 | 0.32 | 27.9 | 0.17 | 8.7 | 0.16 | 77.6 | 0.58 |
| EvNet | 53.2 | 0.15 | 58.3 | 0.20 | 13.1 | 0.14 | 54.9 | 0.11 | 48.1 | 0.21 | 94.3 | 0.59 |
| | | | | | | adv. train. loss: rand. smooth. | | | | | | |
| PriorNet | 1.5 | 0.06 | 0.8 | 0.05 | **89.3** | **0.73** | **97.5** | **0.28** | **99.4** | **0.34** | 38.7 | 0.22 |
| PostNet | 63.3 | 0.26 | 51.8 | 0.46 | 65.3 | 0.86 | 93.4 | 0.26 | 92.9 | 0.48 | 73.2 | 0.63 |
| DDNet | **68.6** | **0.26** | **58.0** | **0.43** | 80.5 | 0.90 | 86.3 | 0.16 | 88.1 | 0.36 | 45.1 | 0.33 |
| EvNet | 58.9 | 0.27 | 56.6 | 0.45 | 63.9 | 0.98 | 92.9 | 0.27 | 74.4 | 0.46 | **85.6** | **0.81** |
| | | | | | | adv. train. loss: crossentropy | | | | | | |
| PriorNet | **99.8** | **0.38** | 0.0 | 0.00 | 31.1 | 0.25 | 0.0 | 0.00 | **100.0** | **0.76** | 60.7 | 0.21 |
| PostNet | 22.2 | 0.15 | 51.2 | 0.21 | 0.0 | 0.00 | **99.4** | **0.22** | 44.9 | 0.18 | 100.0 | 1.44 |
| DDNet | 49.0 | 0.20 | 33.8 | 0.25 | 0.0 | 0.00 | 45.4 | 0.18 | 61.6 | 0.39 | **100.0** | **1.91** |
| EvNet | 29.4 | 0.12 | **84.2** | **0.26** | 2.4 | 0.09 | 96.6 | 0.16 | 8.4 | 0.10 | 100.0 | 0.55 |
| | | | | | | adv. train. loss: diffE | | | | | | |
| PriorNet | 1.1 | 0.04 | 0.0 | 0.00 | **100.0** | **1.91** | **99.2** | **0.31** | **100.0** | **0.76** | 0.0 | 0.00 |
| PostNet | 30.3 | 0.17 | 6.1 | 0.13 | 0.0 | 0.00 | 94.9 | 0.17 | 99.8 | 0.55 | 100.0 | 1.17 |
| DDNet | 37.1 | 0.22 | 4.4 | 0.23 | 0.0 | 0.00 | 81.5 | 0.24 | 100.0 | 0.65 | **100.0** | **1.80** |
| EvNet | **38.6** | **0.31** | 22.6 | 0.15 | 1.0 | 0.11 | 77.9 | 0.32 | 91.8 | 0.21 | 99.8 | 0.62 |

For each model, we observe a performance jump between ID- and OOD-verification, where robustness on ID data drops from high values to low ones while the cc percentage and radius on OOD-data increase. These jumps are observed for normal training as well as adversarial training based on the crossentropy or the differential entropy. Thus, either ID-verification or OOD-verification performs well, depending on the chosen threshold. Augmenting the data set with random noise perturbed samples (randomized smoothing loss) does not result in such performance jumps (except for PriorNet), but there is also a trade-off between robustness on ID data versus robustness on OOD data and there is no parametrization where ID-verification and OOD-verification perform equally well.

Table 6: Randomized smoothing verification for different $\sigma$ of CIFAR10 (ID data) and SVHN (OOD data): percentage of samples that is *wrongly* identified as ID/OOD and certifiably robust as this *wrong* type (cw) and corresponding mean certified radius (R). The lower cw, the more robust the model.

| $\sigma$ | 0.1 | | 0.2 | | 0.5 | | 0.1 | | 0.2 | | 0.5 | |
| | cw | R | cw | R | cw | R | cw | R | cw | R | cw | R |
|---|---|---|---|---|---|---|---|---|---|---|---|---|
| | adv. train. loss: None | | | | | | adv. train. loss: rand. smooth. | | | | | |
| PriorNet | 15.9 | 0.13 | 1.9 | 0.18 | 0.0 | 0.00 | 98.2 | 0.33 | 98.6 | 0.53 | **8.0** | **0.22** |
| PostNet | 74.9 | 0.17 | 73.5 | 0.21 | 100.0 | 1.30 | 35.7 | 0.16 | 46.7 | 0.34 | 32.3 | 0.47 |
| DDNet | 35.1 | 0.14 | 10.1 | 0.17 | 41.6 | 0.35 | **29.9** | **0.11** | 40.5 | 0.31 | 17.6 | 0.32 |
| EvNet | 43.0 | 0.09 | 37.2 | 0.15 | 82.8 | 0.52 | 39.5 | 0.22 | 41.4 | 0.33 | 34.2 | 0.50 |
| | adv. train. loss: crossentropy | | | | | | adv. train. loss: diffE | | | | | |
| PriorNet | **0.1** | **0.12** | 100.0 | 0.76 | 62.2 | 0.33 | 98.4 | 0.31 | 100.0 | 0.74 | 0.0 | 0.00 |
| PostNet | 76.4 | 0.18 | 45.0 | 0.18 | 100.0 | 1.28 | 68.0 | 0.15 | 93.5 | 0.42 | 100.0 | 1.10 |
| DDNet | 49.5 | 0.16 | 64.3 | 0.37 | 100.0 | 1.91 | **61.2** | **0.19** | 95.5 | 0.57 | 100.0 | 1.84 |
| EvNet | 68.3 | 0.12 | **12.9** | **0.11** | 95.6 | 0.39 | 61.2 | 0.33 | **73.8** | **0.18** | 97.9 | 0.60 |

While Table 5 shows the percentage of samples which are correctly identified and certified as ID/OOD data (cc), Table 6 shows the percentage of samples which are wrongly identified as ID/OOD data

---

[1]We want to highlight again that attacks are here only used to enable robust training of the models. The robustness evaluation itself operates on the original data (not attacked and, thus, seemingly easy); only smoothed via randomized smoothing. The verification provides us a radius that guarantees robustness around the sample.

and certifiably robust as this wrong type (cw; certified wrong). These cw samples are worse than adversarial examples. Neither robust training based on label attacks, uncertainty attacks nor noise perturbed samples consistently reduce the portion of certifiably wrong samples, even worse it seems to increase the number of cw samples. Thus, although robust training improves DBU-model resistance against label attacks (see Appendix, Table 35), ID/OOD-verification shows that each model is either robust on ID-data or on OOD-data. Achieving robustness on both types is challenging. Our results rise the following question: How do we make DBU models robust w.r.t. class label predictions and ID/OOD-differentiation without favoring either performance on ID data or OOD data?

## 5 CONCLUSION

This work analyze robustness of uncertainty estimation by DBU models and answer multiple questions in this context. Our results show: (1) While uncertainty estimates are a good indicator to identify correctly classified samples on unperturbed data, performance decrease drastically on perturbed data-points. (2) None of the Dirichlet-based uncertainty models is able to detect PGD label attacks against the classification decision by uncertainty estimation, regardless of the used uncertainty measure. (3) Detecting OOD samples and distinguishing between ID-data and OOD-data is not robust. (4) Robust training based on label attacks or uncertainty attacks increases performance of Dirichlet-based uncertainty models w.r.t. either ID data or OOD data, but achieving high robustness on both is challenging – and poses an interesting direction for future studies.

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

# A    APPENDIX

## A.1    DIRICHLET-BASED UNCERTAINTY MODELS

In this section, we provide details on the losses used by each DBU model. PostNet uses a Bayesian loss which can be expressed as follows:

$$L_{\text{PostNet}} = \frac{1}{N} \sum_i \mathbb{E}_{q(p^{(i)})}[\text{CE}(p^{(i)}, y^{(i)})] - H(q^{(i)}) \tag{3}$$

where CE denotes the cross-entropy. Both the expectation term (i.e. $\mathbb{E}_{q(p^{(i)})}[\text{CE}(p^{(i)}, y^{(i)})]$) and the entropy term (i.e. $H(q^{(i)})$) can be computed in closed-form (Charpentier et al., 2020). PriorNet uses a loss composed of two KL divergence terms for ID and OOD data:

$$L_{\text{PriorNet}} = \frac{1}{N} \left[ \sum_{\boldsymbol{x}^{(i)} \in \text{ID data}} [\text{KL}[\text{Dir}(\alpha^{\text{ID}})||q^{(i)}]] + \sum_{\boldsymbol{x}^{(i)} \in OODdata} [\text{KL}[\text{Dir}(\alpha^{\text{OOD}})||q^{(i)}]] \right]. \tag{4}$$

Both KL divergences terms can be computed in closed-form (Malinin & Gales, 2019). The precision $\alpha^{\text{ID}}$ and $\alpha^{\text{OOD}}$ are hyper-parameters. The precision $\alpha^{\text{ID}}$ is usually set to $1e^1$ for the correct class and 1 otherwise. The precision $\alpha^{\text{OOD}}$ is usually set to **1**. DDNet uses use the Dirichlet likelihood of soft labels produce by an ensemble of $M$ neural networks:

$$L_{\text{DDNet}} = -\frac{1}{N} \sum_i \sum_{m=1}^M [\ln q^{(i)}(\pi^{im})] \tag{5}$$

where $\pi^{im}$ denotes the soft-label of $m$th neural network. The Dirichlet likelihood can be computed in closed-form (Malinin et al., 2019). EvNet uses the expected mean square error between the one-hot encoded label and the predicted categorical distribution:

$$L_{\text{EvNet}} = \frac{1}{N} \sum_i \mathbb{E}_{\boldsymbol{p}^{(i)} \sim \text{Dir}(\boldsymbol{\alpha}^{(i)})} ||\boldsymbol{y} *^{(i)} - \boldsymbol{p}^{(i)}||^2 \tag{6}$$

where $\boldsymbol{y}*^{(i)}$ denotes the one-hot encoded label. The expected MSE loss can also be computed in closed form (Sensoy et al., 2018). For more details please have a look at the original paper on PriorNet (Malinin & Gales, 2018a), PostNet (Charpentier et al., 2020), DDNet (Malinin & Gales, 2019) and EvNet (Sensoy et al., 2018).

## A.2    CLOSED-FORM COMPUTATION OF UNCERTAINTY MEASURES & UNCERTAINTY ATTACKS

Dirichlet-based uncertainty models allow to compute several uncertainty measures in closed form (see (Malinin & Gales, 2018a) for a derivation). As proposed by Malinin & Gales (2018a), we use precision $m_{\alpha_0}$, differential entropy $m_{\text{diffE}}$ and mutual information $m_{\text{MI}}$ to estimate uncertainty on predictions.

The differential entropy $m_{\text{diffE}}$ of a DBU model reaches its maximum value for equally probable categorical distributions and thus, a on flat Dirichlet distribution. It is a measure for distributional uncertainty and expected to be low on ID data, but high on OOD data.

$$m_{\text{diffE}} = \sum_c^K \ln \Gamma(\alpha_c) - \ln \Gamma(\alpha_0) - \sum_c^K (\alpha_c - 1) \cdot (\Psi(\alpha_c) - \Psi(\alpha_0)) \tag{7}$$

where $\alpha$ are the parameters of the Dirichlet-distribution, $\Gamma$ is the Gamma function and $\Psi$ is the Digamma function.

The mutual information $m_{\text{MI}}$ is the difference between the total uncertainty (entropy of the expected distribution) and the expected uncertainty on the data (expected entropy of the distribution). This uncertainty is expected to be low on ID data and high on OOD data.

$$m_{\text{MI}} = -\sum_{c=1}^K \frac{\alpha_c}{\alpha_0} \left( \ln \frac{\alpha_c}{\alpha_0} - \Psi(\alpha_c + 1) + \Psi(\alpha_0 + 1) \right) \tag{8}$$

Furthermore, we use the precision $\alpha_0$ to measure uncertainty, which is expected to be high on ID data and low on OOD data.

$$m_{\alpha_0} = \alpha_0 = \sum_{c=1}^{K} \alpha_c \tag{9}$$

As these uncertainty measures are computed in closed form and it is possible to obtain their gradients, we use them (i.e. $m_{\text{diffE}}$, $m_{\text{MI}}$, $m_{\alpha_0}$) are target function of our uncertainty attacks. Changing the attacked target function allows us to use a wide range of gradient-based attacks such as FGSM attacks, PGD attacks, but also more sophisticated attacks such as Carlini-Wagner attacks.

## A.3 DETAILS OF THE EXPERIMENTAL SETUP

**Models.** We trained all models with a similar based architecture. We used namely 3 linear layers for vector data sets, 3 convolutional layers with size of 5 + 3 linear layers for MNIST and the VGG16 Simonyan & Zisserman (2015) architecture with batch normalization for CIFAR10. All the implementation are performed using Pytorch (Paszke et al., 2019). We optimized all models using Adam optimizer. We performed early stopping by checking for loss improvement every 2 epochs and a patience of 10. The models were trained on GPUs (1 TB SSD).

We performed a grid-search for hyper-parameters for all models. The learning rate grid search was done in $[1e^{-5}, 1e^{-3}]$. For PostNet, we used Radial Flows with a depth of 6 and a latent space equal to 6. Further, we performed a grid search for the regularizing factor in $[1e^{-7}, 1e^{-4}]$. For PriorNet, we performed a grid search for the OOD loss weight in $[1, 10]$. For DDNet, we distilled the knowledge of 5 neural networks after a grid search in $[2, 5, 10, 20]$ neural networks. Note that it already implied a significant overhead at training compare to other models.

**Metrics.** For all experiments, we focused on using AUC-PR scores since it is well suited to imbalance tasks (Saito & Rehmsmeier, 2015) while bringing theoretically similar information than AUC-ROC scores (Davis & Goadrich, 2006). We scaled all scores from $[0, 1]$ to $[0, 100]$. All results are average over 5 training runs using the best hyper-parameters found after the grid search.

**Data sets.** For vector data sets, we use 5 different random splits to train all models. We split the data in training, validation and test sets (60%, 20%, 20%).

We use the segment vector data set Dua & Graff (2017), where the goal is to classify areas of images into 7 classes (window, foliage, grass, brickface, path, cement, sky). We remove class window from ID training data to provide OOD training data to PriorNet. Further, We remove the class 'sky' from training and instead use it as the OOD data set for OOD detection experiments. Each input is composed of 18 attributes describing the image area. The data set contains 2, 310 samples in total.

We further use the Sensorless Drive vector data set Dua & Graff (2017), where the goal is to classify extracted motor current measurements into 11 different classes. We remove class 9 from ID training data to provide OOD training data to PriorNet. We remove classes 10 and 11 from training and use them as the OOD dataset for OOD detection experiments. Each input is composed of 49 attributes describing motor behaviour. The data set contains 58, 509 samples in total.

Additionally, we use the MNIST image data set LeCun & Cortes (2010) where the goal is to classify pictures of hand-drawn digits into 10 classes (from digit 0 to digit 9). Each input is composed of a $1 \times 28 \times 28$ tensor. The data set contains 70, 000 samples. For OOD detection experiments, we use FashionMNIST Xiao et al. (2017) and KMNIST Clanuwat et al. (2018) containing images of Japanese characters and images of clothes, respectively. FashionMNIST was used as training OOD for PriorNet while KMNIST is used as OOD at test time.

Finally, we use the CIFAR10 image data set Krizhevsky et al. (2009) where the goal is to classify a picture of objects into 10 classes (airplane, automobile, bird, cat, deer, dog, frog, horse, ship, truck). Each input is a $3 \times 32 \times 32$ tensor. The data set contains 60, 000 samples. For OOD detection experiments, we use street view house numbers (SVHN) Netzer et al. (2011) and CIFAR100 (Krizhevsky et al., 2009) containing images of numbers and objects respectively. CIFAR100 was used as training OOD for PriorNet while SVHN is used as OOD at test time.

**Perturbations.** For all label and uncertainty attacks, we used Fast Gradient Sign Methods and Project Gradient Descent. We tried 6 different radii $[0.0, 0.1, 0.2, 0.5, 1.0, 2.0, 4.0]$. These radii operate on

the input space after data normalization. We bound perturbations by $L_\infty$-norm or by $L_2$-norm, with

$$L_\infty(x) = \max_{i=1,\dots,D} |x_i| \quad \text{and} \quad L_2(x) = (\sum_{i=1}^{D} x_i^2)^{0.5}. \tag{10}$$

For $L_\infty$-norm it is obvious how to relate perturbation size $\varepsilon$ with perturbed input images, because all inputs are standardized such that the values of their features are between $0$ and $1$. A perturbation of size $\varepsilon = 0$ corresponds to the original input, while a perturbation of size $\varepsilon = 1$ corresponds to the whole input space and allows to change all features to any value.

For $L_2$-norm the relation between perturbation size $\varepsilon$ and perturbed input images is less obvious. To justify our choice for $\varepsilon$ w.r.t. this norm, we relate perturbations size $\varepsilon_2$ corresponding to $L_2$-norm with perturbations size $\varepsilon_\infty$ corresponding to $L_\infty$-norm. First, we compute $\varepsilon_2$, such that the $L_2$-norm is the smallest super-set of the $L_\infty$-norm. Let us consider a perturbation of $\varepsilon_\infty$. The largest $L_2$-norm would be obtained if each feature is perturbed by $\varepsilon_\infty$. Thus, perturbation $\varepsilon_2$, such that $L_2$ encloses $L_\infty$ is $\varepsilon_2 = (\sum_{i=1}^{D} \varepsilon_\infty^2)^{0.5} = \sqrt{D}\varepsilon_\infty$. For the MNIST-data set, with $D = 28 \times 28$ input features $L_2$-norm with $\varepsilon_2 = 28$ encloses $L_\infty$-norm with $\varepsilon_\infty = 1$.

Alternatively, $\varepsilon_2$ can be computes such that the volume spanned by $L_2$-norm is equivalent to the one spanned by $L_\infty$-norm. Using that the volume spanned by $L_\infty$-norm is $\varepsilon_\infty^D$ and the volume spanned by $L_2$-norm is $\frac{\pi^{0.5D}\varepsilon_2^D}{\Gamma(0.5D+1)}$ (where $\Gamma$ is the Gamma-function), we obtain volume equivalence if $\varepsilon_2 = \Gamma(0.5D + 1)^{\frac{1}{D}} \sqrt{\pi}\varepsilon_\infty$. For the MNIST-data set, with $D = 28 \times 28$ input features $L_2$-norm with $\varepsilon_2 \approx 21.39$ is volume equivalent to $L_\infty$-norm with $\varepsilon_\infty = 1$.

## A.4 ADDITIONAL EXPERIMENTS

Table 7 and 8 illustrate that no DBU model maintains high accuracy under gradient-based label attacks. Accuracy under PGD attacks decreases more than under FGSM attacks, since PGD is stronger. Interestingly Noise attacks achieve also good performances with increasing Noise standard deviation. Note that the attack is not constraint to be with a given radius for Noise attacks.

Table 7: Accuracy under PGD label attacks.

| Att. Rad. | 0.0 | 0.1 | 0.2 | 0.5 | 1.0 | 2.0 | 4.0 | | 0.0 | 0.1 | 0.2 | 0.5 | 1.0 | 2.0 | 4.0 |
|---|---|---|---|---|---|---|---|---|---|---|---|---|---|---|---|
| | | | | MNIST | | | | | | | | CIFAR10 | | | |
| PostNet | **99.4** | **99.2** | **98.8** | 96.8 | 89.6 | 53.8 | 13.0 | | 89.5 | 73.5 | 51.7 | 13.2 | 2.2 | 0.8 | 0.3 |
| PriorNet | 99.3 | 99.1 | **98.8** | 97.4 | **93.9** | **75.3** | 4.8 | | 88.2 | **77.8** | **68.4** | **54.0** | **37.9** | **17.5** | **5.1** |
| DDNet | **99.4** | 99.1 | **98.8** | 97.5 | 91.6 | 48.8 | 0.2 | | 86.1 | 73.9 | 59.1 | 20.5 | 1.5 | 0.0 | 0.0 |
| EvNet | 99.2 | 98.9 | 98.4 | 96.8 | 92.4 | 73.1 | **40.9** | | **89.8** | 71.7 | 48.8 | 11.5 | 2.7 | 1.5 | 0.4 |
| | | | | Sensorless | | | | | | | | Segment | | | |
| PostNet | 98.3 | 13.1 | 6.4 | 4.0 | **7.0** | **9.8** | **11.3** | | 98.9 | 82.8 | **50.1** | **19.2** | **8.8** | **5.1** | **8.6** |
| PriorNet | **99.3** | 16.5 | 5.6 | 1.2 | 0.4 | 0.2 | 1.6 | | **99.5** | 90.7 | 47.6 | 7.8 | 0.2 | 0.0 | 0.4 |
| DDNet | **99.3** | 12.4 | 2.4 | 0.6 | 0.3 | 0.1 | 0.1 | | 99.2 | 90.8 | 45.7 | 6.9 | 0.0 | 0.0 | 0.0 |
| EvNet | 99.0 | **35.3** | **22.3** | **11.2** | **7.0** | 5.2 | 4.0 | | 99.3 | 91.8 | 54.0 | 10.3 | 0.8 | 0.5 | 0.6 |

Table 8: Accuracy under FGSM label attacks.

| Att. Rad. | 0.0 | 0.1 | 0.2 | 0.5 | 1.0 | 2.0 | 4.0 | | 0.0 | 0.1 | 0.2 | 0.5 | 1.0 | 2.0 | 4.0 |
|---|---|---|---|---|---|---|---|---|---|---|---|---|---|---|---|
| | | | | MNIST | | | | | | | | CIFAR10 | | | |
| PostNet | **99.4** | **99.2** | 98.9 | 97.7 | 95.2 | **90.1** | **79.2** | | 89.5 | 72.3 | 54.9 | 31.2 | 21.0 | 16.8 | 15.6 |
| PriorNet | 99.3 | 99.1 | 98.9 | 97.7 | **95.8** | 93.2 | 76.7 | | 88.2 | **77.3** | **70.1** | **59.4** | **52.3** | **48.5** | **46.8** |
| DDNet | **99.4** | **99.2** | 98.9 | 97.8 | 94.7 | 79.2 | 25.2 | | 86.1 | 73.0 | 60.2 | 32.5 | 14.6 | 7.1 | 6.0 |
| EvNet | 99.2 | 98.9 | 98.6 | 97.6 | **95.8** | **90.1** | 74.4 | | **89.8** | 71.4 | 54.5 | 29.6 | 18.1 | 14.4 | 13.4 |
| | | | | Sensorless | | | | | | | | Segment | | | |
| PostNet | 98.3 | 19.6 | 10.9 | 10.9 | 11.9 | 12.4 | 12.5 | | 98.9 | 79.6 | **57.3** | **31.5** | 18.4 | **20.6** | **19.9** |
| PriorNet | **99.3** | 24.7 | 11.8 | 8.6 | 8.5 | 8.1 | 8.3 | | **99.5** | 85.5 | 40.5 | 8.9 | 0.4 | 0.3 | 0.2 |
| DDNet | **99.3** | 18.0 | 8.2 | 6.5 | 5.4 | 6.7 | 7.8 | | 99.2 | 86.4 | 36.2 | 11.9 | 0.9 | 0.0 | 0.0 |
| EvNet | 99.0 | **42.0** | **28.0** | **17.5** | **13.7** | **13.6** | **14.9** | | 99.3 | **90.6** | 55.2 | 14.2 | 2.4 | 0.5 | 0.1 |

Table 9: Accuracy under Noise label attacks.

| Noise Std | 0.0 | 0.1 | 0.2 | 0.5 | 1.0 | 2.0 | 4.0 | | 0.0 | 0.1 | 0.2 | 0.5 | 1.0 | 2.0 | 4.0 |
|---|---|---|---|---|---|---|---|---|---|---|---|---|---|---|---|
| | | | | MNIST | | | | | | | | CIFAR10 | | | |
| PostNet | **99.4** | **98.6** | 91.8 | **14.9** | **1.3** | **0.1** | 0.0 | | **91.7** | 21.5 | 10.1 | 0.1 | 1.2 | 0.0 | 1.9 |
| PriorNet | 99.3 | 98.5 | **95.7** | 14.4 | 0.0 | 0.0 | 0.0 | | 87.7 | **28.1** | 11.2 | 9.7 | 5.0 | **8.5** | **9.0** |
| DDNet | **99.4** | **98.6** | 92.4 | 13.3 | 0.7 | 0.0 | 0.0 | | 81.7 | 23.0 | **11.2** | **11.2** | **11.0** | 7.8 | 6.7 |
| EvNet | 99.3 | 96.9 | 81.6 | 11.7 | 0.5 | 0.0 | 0.0 | | 89.5 | 20.7 | 11.1 | 5.2 | 0.5 | 2.3 | 3.9 |
| | | | | Sensorless | | | | | | | | Segment | | | |
| PostNet | 98.1 | 0.1 | **3.7** | **11.7** | **11.7** | **11.7** | **11.7** | | 98.5 | 39.4 | 3.9 | **1.8** | **12.1** | **20.3** | **22.1** |
| PriorNet | **99.3** | 0.2 | 0.0 | 0.0 | 0.0 | 0.3 | 2.4 | | 99.4 | 47.9 | 8.8 | 0.0 | 0.0 | 0.0 | 0.0 |
| DDNet | 99.0 | **0.4** | 0.1 | 0.0 | 0.0 | 0.0 | 0.0 | | 99.1 | 50.0 | 10.3 | 0.0 | 0.0 | 0.3 | 0.0 |
| EvNet | 98.6 | 0.2 | 0.0 | 0.1 | 1.4 | 4.6 | 8.8 | | 99.1 | **50.3** | 10.3 | 1.2 | 0.3 | 0.0 | 1.5 |

### A.4.1 UNCERTAINTY ESTIMATION UNDER LABEL ATTACKS

**Is high certainty a reliable indicator of correct predictions?**

On non-perturbed data uncertainty estimates are an indicator of correctly classified samples, but if the input data is perturbed none of the DBU models maintains its high performance. Thus, uncertainty estimates are not a robust indicator of correctly labeled inputs.

Table 10: Certainty based on differential entropy under PGD label attacks (AUC-PR).

| | | | MNIST | | | | | | | | Segment | | | |
|---|---|---|---|---|---|---|---|---|---|---|---|---|---|---|---|
| Att. Rad. | 0.0 | 0.1 | 0.2 | 0.5 | 1.0 | 2.0 | 4.0 | | 0.0 | 0.1 | 0.2 | 0.5 | 1.0 | 2.0 | 4.0 |
| PostNet | 99.9 | 99.9 | 99.8 | 98.7 | 89.5 | 43.5 | 9.0 | | 99.9 | 77.6 | 31.6 | **11.1** | **5.3** | **4.4** | 8.7 |
| PriorNet | 99.9 | 99.8 | 99.6 | 97.7 | 90.5 | **69.1** | 6.4 | | **100.0** | **96.8** | 44.5 | 4.5 | 0.4 | 0.0 | **15.2** |
| DDNet | **100.0** | **100.0** | **99.9** | **99.7** | **97.6** | 50.2 | 0.1 | | **100.0** | **96.8** | **54.0** | 4.3 | 0.0 | 0.0 | 0.0 |
| EvNet | 99.6 | 99.3 | 98.7 | 96.1 | 88.8 | 63.1 | **31.7** | | **100.0** | 95.9 | 44.3 | 5.9 | 0.8 | 0.6 | 0.7 |

Table 2, 10, 11, and 12 illustrate that neither differential entropy nor precision, nor mutual information are a reliable indicator of correct predictions under PGD attacks. DBU-models achieve significantly

Table 11: Certainty based on precision $\alpha_0$ under PGD label attacks (AUC-PR).

| Att. Rad. | 0.0 | 0.1 | 0.2 | 0.5 | 1.0 | 2.0 | 4.0 | | 0.0 | 0.1 | 0.2 | 0.5 | 1.0 | 2.0 | 4.0 |
|---|---|---|---|---|---|---|---|---|---|---|---|---|---|---|---|
| | | | | MNIST | | | | | | | | CIFAR10 | | | |
| PostNet | **100.0** | 99.9 | 99.7 | 98.2 | 87.9 | 39.1 | 6.9 | | **98.7** | 88.6 | 56.2 | 7.8 | 1.2 | 0.4 | 0.3 |
| PriorNet | 99.9 | 99.8 | 99.6 | 97.7 | 90.4 | **69.1** | 6.6 | | 92.9 | 77.7 | 60.5 | **37.6** | **24.9** | **11.3** | **3.0** |
| DDNet | **100.0** | **100.0** | **100.0** | **99.8** | **98.2** | 51.1 | 0.1 | | 97.6 | **91.8** | **78.3** | 18.1 | 0.8 | 0.0 | 0.0 |
| EvNet | 99.6 | 99.2 | 98.6 | 95.7 | 88.6 | 63.6 | **32.6** | | 97.9 | 85.9 | 57.2 | 10.2 | 4.0 | 2.4 | 0.3 |
| | | | | Sensorless | | | | | | | | Segment | | | |
| PostNet | 99.6 | 7.0 | 3.3 | 3.1 | **6.9** | **9.8** | 11.3 | | 99.9 | 74.2 | 31.6 | **11.1** | **5.0** | **4.2** | **8.6** |
| PriorNet | 99.8 | 10.5 | 3.2 | 0.6 | 0.2 | 0.2 | 1.8 | | **100.0** | 96.9 | **45.2** | 4.4 | 0.4 | 0.0 | 1.2 |
| DDNet | 99.8 | 8.7 | 1.3 | 0.3 | 0.2 | 0.1 | 0.2 | | **100.0** | **97.1** | 45.0 | 4.1 | 0.0 | 0.0 | 0.0 |
| EvNet | **99.9** | **23.2** | **13.2** | **6.0** | 3.7 | 2.7 | 2.1 | | **100.0** | 95.7 | 44.5 | 5.9 | 0.8 | 0.6 | 0.7 |

Table 12: Certainty based on mutual information under PGD label attacks (AUC-PR).

| Att. Rad. | 0.0 | 0.1 | 0.2 | 0.5 | 1.0 | 2.0 | 4.0 | | 0.0 | 0.1 | 0.2 | 0.5 | 1.0 | 2.0 | 4.0 |
|---|---|---|---|---|---|---|---|---|---|---|---|---|---|---|---|
| | | | | MNIST | | | | | | | | CIFAR10 | | | |
| PostNet | 99.7 | 99.7 | 99.6 | 99.2 | 92.4 | 40.0 | 6.9 | | **97.3** | 84.5 | 56.2 | 12.2 | 2.4 | 0.7 | 0.3 |
| PriorNet | 99.9 | 99.8 | 99.6 | 97.7 | 90.3 | **68.9** | 6.4 | | 82.7 | 65.6 | 51.4 | **35.5** | **24.4** | **11.0** | **2.9** |
| DDNet | **100.0** | 99.9 | 99.9 | 99.7 | **97.4** | 50.2 | 0.1 | | 96.9 | **90.8** | **77.2** | 18.8 | 0.8 | 0.0 | 0.0 |
| EvNet | 97.8 | 97.0 | 95.7 | 92.6 | 86.1 | 62.3 | **28.9** | | 91.3 | 72.4 | 47.9 | 11.4 | 1.6 | 0.9 | 1.6 |
| | | | | Sensorless | | | | | | | | Segment | | | |
| PostNet | 99.3 | 7.0 | 3.3 | 3.3 | **7.0** | **9.8** | 11.3 | | 99.9 | 73.2 | 31.5 | **11.1** | **5.0** | **4.3** | **8.7** |
| PriorNet | **99.8** | 10.5 | 3.2 | 0.6 | 0.2 | 0.1 | **11.8** | | **100.0** | **96.6** | **45.2** | 4.5 | 0.4 | 0.0 | 1.1 |
| DDNet | 99.6 | 8.6 | 1.3 | 0.3 | 0.2 | 0.1 | 0.1 | | **100.0** | 96.5 | 42.4 | 4.1 | 0.0 | 0.0 | 0.0 |
| EvNet | 99.1 | **22.0** | **12.6** | **5.9** | 3.7 | 2.7 | 2.2 | | **100.0** | 90.5 | 41.0 | 5.9 | 0.8 | 0.6 | 0.7 |

Table 13: Certainty based on differential entropy under FGSM label attacks (AUC-PR).

| Att. Rad. | 0.0 | 0.1 | 0.2 | 0.5 | 1.0 | 2.0 | 4.0 | | 0.0 | 0.1 | 0.2 | 0.5 | 1.0 | 2.0 | 4.0 |
|---|---|---|---|---|---|---|---|---|---|---|---|---|---|---|---|
| | | | | MNIST | | | | | | | | CIFAR10 | | | |
| PostNet | 99.9 | 99.9 | 99.8 | 99.4 | 97.8 | **92.1** | **83.2** | | **98.5** | 88.7 | 68.9 | 31.0 | 18.6 | 15.5 | 16.7 |
| PriorNet | 99.9 | 99.9 | 99.7 | 98.3 | 94.1 | 88.5 | 78.6 | | 90.1 | 73.6 | 61.6 | **46.1** | **38.5** | **35.6** | **37.3** |
| DDNet | **100.0** | **100.0** | 99.9 | **99.8** | **98.7** | 86.4 | 23.0 | | 97.3 | **90.6** | **78.7** | 39.4 | 13.7 | 6.0 | 5.1 |
| EvNet | 99.6 | 99.4 | 99.1 | 97.8 | 95.8 | 90.4 | 76.8 | | 98.0 | 86.2 | 67.4 | 32.7 | 19.9 | 18.2 | 19.7 |
| | | | | Sensorless | | | | | | | | Segment | | | |
| PostNet | 99.7 | 11.7 | 7.3 | 9.3 | 11.8 | 12.5 | 12.5 | | 99.9 | 73.6 | 40.6 | **23.7** | **17.2** | **19.8** | **20.2** |
| PriorNet | 99.8 | 21.4 | 10.4 | 8.5 | 9.0 | 9.2 | 10.3 | | **100.0** | 93.7 | 37.7 | 5.8 | 1.1 | 0.9 | 0.8 |
| DDNet | 99.7 | 18.5 | 5.4 | 4.3 | 4.2 | 5.7 | 7.9 | | **100.0** | **94.1** | 42.9 | 7.2 | 1.0 | 0.0 | 0.0 |
| EvNet | **99.9** | **44.8** | **29.2** | **18.2** | **15.1** | **14.9** | **15.5** | | **100.0** | 93.7 | **48.7** | 8.7 | 2.4 | 1.6 | 0.5 |

Table 14: Certainty based on differential entropy under Noise label attacks (AUC-PR).

| Noise Std | 0.0 | 0.1 | 0.2 | 0.5 | 1.0 | 2.0 | 4.0 | | 0.0 | 0.1 | 0.2 | 0.5 | 1.0 | 2.0 | 4.0 |
|---|---|---|---|---|---|---|---|---|---|---|---|---|---|---|---|
| | | | | MNIST | | | | | | | | CIFAR10 | | | |
| PostNet | 99.9 | 99.8 | 99.6 | 74.2 | **7.4** | **0.2** | 0.0 | | **98.7** | 76.3 | 24.3 | 0.4 | 4.9 | 0.0 | 1.7 |
| PriorNet | 99.9 | 99.9 | **99.8** | 73.4 | 0.0 | 0.0 | 0.0 | | 85.0 | 27.8 | 15.9 | **20.4** | 7.0 | **7.7** | **8.3** |
| DDNet | **100.0** | **99.9** | 99.4 | 51.1 | 0.6 | 0.1 | 0.0 | | 96.1 | 61.0 | **39.8** | 14.2 | **11.3** | 6.9 | 6.9 |
| EvNet | 99.5 | 98.4 | 88.5 | 20.2 | 0.9 | 0.0 | 0.0 | | 97.5 | 66.1 | 21.4 | 7.7 | 2.3 | 3.0 | 3.8 |
| | | | | Sensorless | | | | | | | | Segment | | | |
| PostNet | 99.7 | 0.3 | **3.2** | 13.3 | 12.0 | 11.7 | 11.7 | | 99.9 | 53.9 | 4.8 | 1.8 | **11.2** | **21.7** | 21.6 |
| PriorNet | **100.0** | 0.3 | 0.0 | 0.0 | 0.0 | 7.8 | 11.5 | | **100.0** | **84.5** | 15.6 | 0.0 | 0.0 | 0.0 | 0.0 |
| DDNet | 99.7 | **0.9** | 0.6 | 0.0 | 0.0 | 0.0 | 0.0 | | **100.0** | 82.7 | **23.9** | 0.0 | 0.0 | 0.6 | 0.0 |
| EvNet | 99.8 | 0.3 | 0.0 | 0.1 | 1.7 | 5.5 | 10.0 | | **100.0** | 78.3 | 19.0 | **3.5** | 0.5 | 0.0 | 1.7 |

better results when they are attacked by FGSM-attacks (Table 13), but as FGSM attacks provide much weaker adversarial examples than PGD attacks, this cannot be seen as real advantage.

**Can we use uncertainty estimates to detect attacks against the classification decision?**

PGD attacks do not explicitly consider uncertainty during the computation of adversarial examples, but they seem to provide perturbed inputs with similar uncertainty as the original input.

Table 15: Attack-Detection based on differential entropy under PGD label attacks (AUC-PR).

| | MNIST | | | | | | | Segment | | | | | |
|---|---|---|---|---|---|---|---|---|---|---|---|---|---|
| Att. Rad. | 0.1 | 0.2 | 0.5 | 1.0 | 2.0 | 4.0 | | 0.1 | 0.2 | 0.5 | 1.0 | 2.0 | 4.0 |
| PostNet | 57.7 | 66.3 | 83.4 | 90.5 | 79.0 | 50.1 | | **95.6** | 73.5 | **47.0** | **42.3** | **53.4** | **82.7** |
| PriorNet | **67.7** | **83.2** | **97.1** | **96.7** | 92.1 | 82.9 | | 86.7 | 83.3 | 38.0 | 31.3 | 30.8 | 31.5 |
| DDNet | 53.4 | 57.1 | 68.5 | 83.9 | **96.0** | **86.3** | | 76.1 | **83.5** | 45.4 | 32.4 | 30.8 | 30.8 |
| EvNet | 54.8 | 59.0 | 68.5 | 75.9 | 72.6 | 59.8 | | 94.9 | 80.9 | 41.5 | 32.5 | 31.1 | 31.1 |

Table 16: Attack-Detection based on precision $\alpha_0$ under PGD label attacks (AUC-PR).

| Att. Rad. | 0.1 | 0.2 | 0.5 | 1.0 | 2.0 | 4.0 | | 0.1 | 0.2 | 0.5 | 1.0 | 2.0 | 4.0 |
|---|---|---|---|---|---|---|---|---|---|---|---|---|---|
| | | | MNIST | | | | | | | CIFAR10 | | | |
| PostNet | 63.3 | 75.7 | 92.6 | 95.1 | 75.3 | 39.5 | | **63.4** | **66.9** | 42.1 | 32.9 | 31.6 | 31.2 |
| PriorNet | **67.6** | **83.2** | **97.1** | **96.9** | **92.7** | **84.7** | | 53.3 | 56.0 | 55.6 | **49.2** | 42.2 | 35.4 |
| DDNet | 52.7 | 55.7 | 64.7 | 78.4 | 91.9 | 80.9 | | 55.8 | 60.5 | **57.3** | 38.7 | 32.3 | 31.4 |
| EvNet | 49.1 | 48.0 | 45.1 | 42.7 | 41.8 | 39.2 | | 48.4 | 46.9 | 46.3 | 46.3 | **44.5** | **42.5** |
| | | | Sensorless | | | | | | | Segment | | | |
| PostNet | 39.8 | 35.8 | 35.4 | **52.0** | **88.2** | **99.0** | | **94.6** | 70.3 | **46.3** | **42.6** | **54.9** | **84.0** |
| PriorNet | 40.9 | 35.1 | 32.0 | 31.1 | 30.7 | 30.7 | | 82.7 | 82.6 | 39.4 | 31.6 | 30.8 | 30.8 |
| DDNet | **47.7** | **40.3** | 35.3 | 32.8 | 31.3 | 30.8 | | 80.0 | **86.0** | 43.3 | 33.6 | 31.0 | 30.8 |
| EvNet | 45.4 | 39.7 | **36.1** | 34.8 | 34.7 | 36.0 | | 90.9 | 72.4 | 40.4 | 32.4 | 31.1 | 31.1 |

Table 17: Attack-Detection based on mutual information under PGD label attacks (AUC-PR).

| Att. Rad. | 0.1 | 0.2 | 0.5 | 1.0 | 2.0 | 4.0 | | 0.1 | 0.2 | 0.5 | 1.0 | 2.0 | 4.0 |
|---|---|---|---|---|---|---|---|---|---|---|---|---|---|
| | | | MNIST | | | | | | | CIFAR10 | | | |
| PostNet | 42.2 | 37.5 | 36.7 | 54.5 | 70.5 | 70.3 | | 52.2 | 52.1 | 50.0 | **65.9** | **76.3** | **80.7** |
| PriorNet | **67.7** | **83.3** | **97.1** | **96.9** | **92.6** | **84.5** | | 54.0 | 56.9 | 56.3 | 49.7 | 42.4 | 35.5 |
| DDNet | 53.1 | 56.3 | 66.5 | 81.0 | **94.0** | 82.9 | | **56.0** | **60.8** | **57.4** | 38.2 | 32.1 | 31.3 |
| EvNet | 49.1 | 48.0 | 45.2 | 42.9 | 41.9 | 39.3 | | 48.7 | 47.3 | 46.3 | 46.0 | 44.1 | 42.2 |
| | | | Sensorless | | | | | | | Segment | | | |
| PostNet | **75.3** | **76.6** | **66.5** | **57.7** | **85.6** | **98.7** | | **94.8** | 73.5 | **55.9** | **47.9** | **58.0** | **84.0** |
| PriorNet | 40.7 | 35.0 | 32.0 | 31.0 | 30.7 | 30.7 | | 83.5 | 82.7 | 39.2 | 31.6 | 30.8 | 30.8 |
| DDNet | 48.0 | 40.0 | 35.2 | 32.6 | 31.2 | 30.8 | | 82.4 | **88.1** | 43.4 | 33.4 | 30.9 | 30.8 |
| EvNet | 45.5 | 39.7 | 36.1 | 34.8 | 34.7 | 36.0 | | 91.7 | 72.9 | 40.5 | 32.4 | 31.1 | 31.1 |

FGSM and Noise attacks are easier to detect, but also weaker thand PGD attacks. This suggests that DBU models are capable of detecting weak attacks by using uncertainty estimation.

Table 18: Attack-Detection based on differential entropy under FGSM label attacks (AUC-PR).

| Att. Rad. | 0.1 | 0.2 | 0.5 | 1.0 | 2.0 | 4.0 | | 0.1 | 0.2 | 0.5 | 1.0 | 2.0 | 4.0 |
|---|---|---|---|---|---|---|---|---|---|---|---|---|---|
| | | | MNIST | | | | | | | CIFAR10 | | | |
| PostNet | 55.9 | 61.8 | 74.8 | 84.0 | 88.9 | 89.9 | | **62.1** | **67.2** | 65.7 | 63.1 | 65.4 | 73.8 |
| PriorNet | **67.4** | **82.4** | **96.9** | **98.3** | **98.9** | **99.6** | | 58.4 | 63.1 | 68.5 | **70.1** | 68.5 | 62.5 |
| DDNet | 53.6 | 57.3 | 68.3 | 82.6 | 95.6 | 98.7 | | 57.2 | 62.9 | **69.1** | 68.7 | **69.7** | **76.5** |
| EvNet | 54.1 | 57.4 | 63.8 | 67.6 | 68.6 | 69.9 | | 57.8 | 61.7 | 63.3 | 62.9 | 65.7 | 72.5 |
| | | | Sensorless | | | | | | | Segment | | | |
| PostNet | **98.4** | **99.8** | **99.9** | **99.9** | **99.9** | **99.9** | | **96.9** | 93.9 | **99.5** | **99.9** | **100.0** | **100.0** |
| PriorNet | 48.7 | 38.6 | 32.7 | 32.9 | 38.6 | 44.3 | | 89.0 | 80.8 | 46.7 | 37.2 | 33.7 | 32.4 |
| DDNet | 61.5 | 47.8 | 37.1 | 33.1 | 32.4 | 33.2 | | 79.6 | **86.2** | 60.2 | 47.5 | 36.6 | 31.6 |
| EvNet | 67.3 | 65.5 | 72.3 | 73.4 | 75.3 | 79.1 | | 95.7 | 87.2 | 59.3 | 51.7 | 51.1 | 53.5 |

Table 19: Attack-Detection based on differential entropy under Noise label attacks (AUC-PR).

| Noise Std. | 0.1 | 0.2 | 0.5 | 1.0 | 2.0 | 4.0 | 0.1 | 0.2 | 0.5 | 1.0 | 2.0 | 4.0 |
|---|---|---|---|---|---|---|---|---|---|---|---|---|
| | | | MNIST | | | | | | CIFAR10 | | | |
| PostNet | 51.3 | 65.3 | 93.8 | 95.1 | 95.2 | 95.2 | **80.8** | **84.5** | **97.6** | **99.5** | 99.3 | 98.2 |
| PriorNet | 32.5 | 36.8 | 88.9 | 99.6 | 99.7 | 92.7 | 34.7 | 32.3 | 34.3 | 60.3 | 95.5 | **100.0** |
| DDNet | **60.7** | **87.6** | **99.8** | **100.0** | **99.9** | **99.8** | 59.1 | 62.6 | 81.5 | 98.6 | **99.8** | 98.7 |
| EvNet | 51.2 | 55.7 | 66.9 | 70.3 | 68.0 | 67.1 | 75.7 | 78.6 | 88.2 | 97.8 | 96.4 | 95.6 |
| | | | Sensorless | | | | | | Segment | | | |
| PostNet | **99.8** | **100.0** | **100.0** | **100.0** | **100.0** | **100.0** | **95.6** | **99.4** | **100.0** | **100.0** | **100.0** | **100.0** |
| PriorNet | 42.0 | 33.8 | 31.5 | 34.7 | 43.7 | 47.0 | 56.7 | 56.7 | 39.8 | 33.7 | 31.9 | 33.7 |
| DDNet | 53.4 | 43.5 | 34.3 | 31.6 | 32.5 | 36.1 | 57.0 | 58.9 | 43.1 | 33.7 | 31.5 | 31.3 |
| EvNet | 67.1 | 78.8 | 88.3 | 95.4 | 96.9 | 97.8 | 60.8 | 63.5 | 61.2 | 64.8 | 73.7 | 85.2 |

### A.4.2 ATTACKING UNCERTAINTY ESTIMATION

**Are uncertainty estimates a robust feature for OOD detection?**

Using uncertainty estimation to distinguish between ID and OOD data is not robust as shown in the following tables.

Table 20: OOD detection based on differential entropy under PGD uncertainty attacks against differential entropy on ID data and OOD data (AUC-PR).

| | ID-Attack (non-attacked OOD) | | | | | | | OOD-Attack (non-attacked ID) | | | | | | |
|---|---|---|---|---|---|---|---|---|---|---|---|---|---|---|
| Att. Rad. | 0.0 | 0.1 | 0.2 | 0.5 | 1.0 | 2.0 | 4.0 | 0.0 | 0.1 | 0.2 | 0.5 | 1.0 | 2.0 | 4.0 |
| | | | | | | | **MNIST – KMNIST** | | | | | | | |
| PostNet | 94.5 | 94.1 | 93.9 | 91.1 | 77.1 | 44.0 | 31.9 | 94.5 | 93.1 | 91.4 | 82.1 | 62.2 | 50.7 | 48.8 |
| PriorNet | **99.6** | **99.4** | **99.1** | **97.8** | **93.8** | **77.6** | **32.0** | **99.6** | **99.4** | **99.1** | 98.0 | 94.6 | 85.5 | 73.9 |
| DDNet | 99.3 | 99.1 | 98.9 | **97.8** | 93.5 | 63.3 | 30.7 | 99.3 | 99.1 | 99.0 | **98.3** | **96.7** | **91.3** | 73.8 |
| EvNet | 69.0 | 67.1 | 65.6 | 61.8 | 57.4 | 50.9 | 43.6 | 69.0 | 55.8 | 48.0 | 39.4 | 36.2 | 34.9 | 34.4 |
| | | | | | | | **Seg. – Seg. class sky** | | | | | | | |
| PostNet | **99.0** | **80.7** | **53.5** | **38.0** | **34.0** | **41.6** | **49.5** | **99.0** | **88.4** | 69.2 | 45.1 | **36.4** | **42.6** | 75.4 |
| PriorNet | 34.8 | 31.4 | 30.9 | 30.8 | 30.8 | 30.8 | 30.8 | 34.8 | 31.8 | 31.0 | 30.8 | 30.8 | 30.8 | 32.1 |
| DDNet | 31.5 | 30.9 | 30.8 | 30.8 | 30.8 | 30.8 | 30.8 | 31.5 | 31.0 | 30.8 | 30.8 | 30.8 | 30.8 | 30.8 |
| EvNet | 92.5 | 67.2 | 43.2 | 31.6 | 30.9 | 30.9 | 31.2 | 92.5 | 86.1 | **82.7** | **48.9** | 32.7 | 30.9 | 30.9 |

Table 21: OOD detection under PGD uncertainty attacks against differential entropy on ID data and OOD data (AUC-ROC).

| | ID-Attack (non-attacked OOD) | | | | | | | OOD-Attack (non-attacked ID) | | | | | | |
|---|---|---|---|---|---|---|---|---|---|---|---|---|---|---|
| Att. Rad. | 0.0 | 0.1 | 0.2 | 0.5 | 1.0 | 2.0 | 4.0 | 0.0 | 0.1 | 0.2 | 0.5 | 1.0 | 2.0 | 4.0 |
| | | | | | | | **MNIST – KMNIST** | | | | | | | |
| PostNet | 91.6 | 91.3 | 91.9 | 91.5 | 80.2 | 38.8 | 9.2 | 91.6 | 90.4 | 89.0 | 81.6 | 62.6 | 45.0 | 43.1 |
| PriorNet | **99.8** | **99.7** | **99.5** | **99.0** | **97.1** | **81.1** | 8.7 | **99.8** | **99.7** | **99.6** | **99.1** | **97.7** | **93.0** | **84.9** |
| DDNet | 99.2 | 98.9 | 98.6 | 97.3 | 92.1 | 58.2 | 1.2 | 99.2 | 99.0 | 98.8 | 97.9 | 95.8 | 89.1 | 69.3 |
| EvNet | 81.2 | 79.6 | 78.2 | 74.6 | 69.5 | 58.7 | **43.0** | 81.2 | 67.2 | 54.8 | 35.4 | 25.5 | 20.7 | 18.5 |
| | | | | | | | **CIFAR10 – SVHN** | | | | | | | |
| PostNet | 87.0 | 71.9 | 56.3 | **30.2** | **20.2** | **15.0** | 9.7 | 87.0 | 71.0 | 54.3 | 33.5 | 30.3 | 26.2 | 19.4 |
| PriorNet | 62.4 | 48.2 | 35.9 | 13.8 | 3.6 | 0.9 | 0.3 | 62.4 | 48.0 | 35.6 | 14.8 | 6.6 | 3.4 | 1.6 |
| DDNet | 87.0 | **76.0** | **63.6** | 29.3 | 6.1 | 1.1 | 0.4 | 87.0 | **78.1** | **66.1** | 26.2 | 5.1 | 0.7 | 0.1 |
| EvNet | **88.0** | 69.1 | 51.7 | 24.6 | 15.5 | 9.5 | 4.2 | **88.0** | 72.0 | 60.7 | **47.9** | **42.1** | **33.3** | **24.0** |
| | | | | | | | **Sens. – Sens. class 10, 11** | | | | | | | |
| PostNet | **85.3** | **49.1** | **38.1** | **7.8** | **8.2** | 8.2 | 8.2 | **85.3** | **57.2** | **54.0** | **27.3** | **31.5** | **86.7** | **99.5** |
| PriorNet | 28.1 | 0.8 | 0.3 | 0.4 | 1.6 | **8.4** | **26.8** | 28.1 | 2.5 | 0.7 | 0.2 | 2.3 | 18.9 | 41.0 |
| DDNet | 21.0 | 3.0 | 0.9 | 0.4 | 0.6 | 2.1 | 7.3 | 21.0 | 4.4 | 2.1 | 1.9 | 2.2 | 2.2 | 4.1 |
| EvNet | 74.2 | 21.4 | 12.2 | 4.3 | 1.4 | 0.6 | 0.3 | 74.2 | 45.3 | 38.5 | 19.6 | 9.6 | 12.1 | 26.0 |
| | | | | | | | **Seg. – Seg. class sky** | | | | | | | |
| PostNet | **99.2** | **84.7** | **55.5** | **23.0** | **9.7** | **4.4** | **4.7** | **99.2** | **92.1** | **77.1** | 41.5 | **24.9** | **41.0** | **80.8** |
| PriorNet | 17.1 | 4.4 | 1.3 | 0.0 | 0.0 | 0.0 | 0.1 | 17.1 | 5.9 | 1.5 | 0.1 | 0.0 | 0.1 | 5.8 |
| DDNet | 4.1 | 1.1 | 0.0 | 0.0 | 0.0 | 0.0 | 0.0 | 4.1 | 1.8 | 0.4 | 0.0 | 0.0 | 0.0 | 0.0 |
| EvNet | 91.2 | 54.5 | 23.3 | 3.9 | 0.9 | 0.4 | 0.2 | 91.2 | 82.9 | 76.4 | **42.2** | 9.7 | 0.8 | 0.6 |

Table 22: OOD detection (AU-PR) under PGD uncertainty attacks against precision $\alpha_0$ on ID data and OOD data.

| | ID-Attack (non-attacked OOD) | | | | | | | | OOD-Attack (non-attacked ID) | | | | | | |
|---|---|---|---|---|---|---|---|---|---|---|---|---|---|---|---|
| Att. Rad. | 0.0 | 0.1 | 0.2 | 0.5 | 1.0 | 2.0 | 4.0 | | 0.0 | 0.1 | 0.2 | 0.5 | 1.0 | 2.0 | 4.0 |
| | | | | | | | **MNIST – KMNIST** | | | | | | | | |
| PostNet | 98.4 | 97.4 | 96.0 | 88.8 | 70.9 | 39.3 | 31.3 | | 98.4 | 97.2 | 95.2 | 82.8 | 52.6 | 34.3 | 32.1 |
| PriorNet | **99.6** | **99.5** | **99.2** | **98.0** | **94.1** | **76.0** | 31.1 | | **99.6** | **99.5** | **99.2** | **98.2** | **95.3** | **87.5** | **75.6** |
| DDNet | 97.2 | 96.7 | 96.1 | 93.8 | 86.4 | 53.2 | 31.0 | | 97.2 | 96.7 | 96.2 | 94.5 | 91.1 | 82.9 | 64.6 |
| EvNet | 39.8 | 39.2 | 38.8 | 37.9 | 37.1 | 36.3 | **35.4** | | 39.8 | 34.5 | 32.5 | 31.2 | 31.0 | 30.9 | 31.0 |
| | | | | | | | **CIFAR10 – SVHN** | | | | | | | | |
| PostNet | **82.4** | 63.8 | 46.1 | 22.3 | 17.4 | 16.7 | 16.4 | | **82.4** | 61.8 | 41.5 | 21.8 | **19.8** | **17.5** | **15.8** |
| PriorNet | 37.9 | 25.0 | 19.2 | 15.8 | 15.4 | 15.4 | 15.4 | | 37.9 | 25.9 | 19.4 | 15.6 | 15.4 | 15.4 | 15.4 |
| DDNet | 81.1 | **70.1** | **58.4** | **30.0** | 16.7 | 15.5 | 15.4 | | 81.1 | **71.2** | **59.9** | **27.8** | 16.5 | 15.5 | 15.4 |
| EvNet | 34.7 | 27.4 | 25.4 | 22.0 | **19.7** | **18.1** | **17.1** | | 34.7 | 19.4 | 18.1 | 17.1 | 16.8 | 16.2 | 15.7 |
| | | | | | | | **Sens. – Sens. class 10, 11** | | | | | | | | |
| PostNet | **77.4** | **39.6** | **35.9** | **31.7** | **44.4** | **44.4** | **44.4** | | **77.4** | 40.3 | **38.6** | 29.5 | **34.0** | **79.4** | **97.4** |
| PriorNet | 35.9 | 27.0 | 26.8 | 26.8 | 26.8 | 27.5 | 36.2 | | 35.9 | 27.7 | 27.0 | 26.7 | 26.6 | 26.5 | 26.5 |
| DDNet | 55.6 | 34.4 | 31.7 | 30.4 | 29.5 | 30.2 | 33.4 | | 55.6 | **40.9** | 34.1 | 28.0 | 26.9 | 26.6 | 26.5 |
| EvNet | 66.3 | 33.3 | 29.7 | 27.0 | 27.1 | 29.2 | 33.9 | | 66.3 | 39.3 | 37.1 | **31.3** | 28.3 | 28.4 | 29.7 |
| | | | | | | | **Seg. – Seg. class sky** | | | | | | | | |
| PostNet | **98.4** | 74.8 | 51.0 | **37.2** | **32.8** | **43.5** | **49.9** | | **98.4** | 84.7 | 66.1 | 42.4 | 34.8 | **40.9** | **71.2** |
| PriorNet | 32.1 | 30.9 | 30.8 | 30.8 | 30.8 | 30.8 | 30.8 | | 32.1 | 31.0 | 30.8 | 30.8 | 30.8 | 30.8 | 30.8 |
| DDNet | 31.0 | 30.8 | 30.8 | 30.8 | 30.8 | 30.8 | 30.8 | | 31.0 | 30.8 | 30.8 | 30.8 | 30.8 | 30.8 | 30.8 |
| EvNet | 98.3 | **83.0** | **60.5** | 34.0 | 31.0 | 30.8 | 30.8 | | 98.3 | **94.4** | **88.8** | **65.6** | **37.0** | 31.4 | 30.9 |

Table 23: OOD detection (AUC-ROC) under PGD uncertainty attacks against precision $\alpha_0$ on ID data and OOD data.

| | ID-Attack (non-attacked OOD) | | | | | | | | OOD-Attack (non-attacked ID) | | | | | | |
|---|---|---|---|---|---|---|---|---|---|---|---|---|---|---|---|
| Att. Rad. | 0.0 | 0.1 | 0.2 | 0.5 | 1.0 | 2.0 | 4.0 | | 0.0 | 0.1 | 0.2 | 0.5 | 1.0 | 2.0 | 4.0 |
| | | | | | | | **MNIST – KMNIST** | | | | | | | | |
| PostNet | 98.4 | 97.6 | 96.4 | 90.9 | 74.0 | 28.9 | 6.3 | | 98.4 | 97.6 | 96.3 | 89.0 | 61.3 | 19.6 | 9.7 |
| PriorNet | **99.8** | **99.7** | **99.6** | **99.1** | **97.2** | **79.4** | 4.4 | | **99.8** | **99.7** | **99.6** | **99.2** | **98.0** | **93.9** | **85.8** |
| DDNet | 96.5 | 95.9 | 95.1 | 92.0 | 82.6 | 44.3 | 3.5 | | 96.5 | 95.9 | 95.2 | 92.9 | 88.6 | 78.7 | 59.4 |
| EvNet | 35.9 | 34.1 | 32.8 | 30.1 | 27.4 | 24.6 | **21.4** | | 35.9 | 18.7 | 10.4 | 3.7 | 2.0 | 1.7 | 2.0 |
| | | | | | | | **CIFAR10 – SVHN** | | | | | | | | |
| PostNet | **87.4** | 71.2 | 54.8 | 29.2 | 19.0 | 14.0 | 9.4 | | **87.4** | 71.4 | 54.1 | 30.1 | **25.8** | **17.5** | **5.8** |
| PriorNet | 45.6 | 31.1 | 20.4 | 6.3 | 1.4 | 0.3 | 0.1 | | 45.6 | 32.2 | 21.7 | 5.4 | 1.0 | 0.3 | 0.1 |
| DDNet | 84.9 | **73.8** | **61.8** | 30.2 | 9.3 | 3.0 | 0.8 | | 84.9 | **76.6** | **66.2** | **34.6** | 10.4 | 2.3 | 0.3 |
| EvNet | 61.2 | 49.4 | 45.2 | **37.6** | **30.5** | **23.4** | **17.0** | | 61.2 | 29.4 | 23.0 | 16.8 | 14.2 | 10.2 | 5.5 |
| | | | | | | | **Sens. – Sens. class 10, 11** | | | | | | | | |
| PostNet | **87.2** | **48.8** | **37.3** | 4.1 | 0.7 | 0.7 | 0.7 | | **87.2** | 50.0 | 45.4 | 16.5 | **27.6** | 81.9 | 98.0 |
| PriorNet | 37.3 | 3.5 | 2.4 | 2.2 | 2.9 | 6.3 | **19.2** | | 37.3 | 8.0 | 3.6 | 1.4 | 0.6 | 0.1 | 0.0 |
| DDNet | 55.2 | 23.7 | 17.7 | **14.1** | **12.5** | **12.7** | 15.7 | | 55.2 | 37.1 | 27.7 | 9.4 | 2.5 | 0.6 | 0.1 |
| EvNet | 75.5 | 30.8 | 18.2 | 5.8 | 1.6 | 0.6 | 0.2 | | 75.5 | 47.8 | 41.9 | **24.1** | 10.2 | 10.2 | 15.6 |
| | | | | | | | **Seg. – Seg. class sky** | | | | | | | | |
| PostNet | **98.6** | 77.7 | 50.8 | 20.3 | 8.2 | 1.3 | **0.5** | | **98.6** | 88.9 | 73.4 | 36.2 | 19.4 | **36.7** | **75.2** |
| PriorNet | 8.5 | 1.3 | 0.2 | 0.0 | 0.0 | 0.0 | 0.1 | | 8.5 | 2.0 | 0.4 | 0.0 | 0.0 | 0.0 | 0.0 |
| DDNet | 2.2 | 0.3 | 0.0 | 0.0 | 0.0 | 0.0 | 0.0 | | 2.2 | 0.5 | 0.1 | 0.0 | 0.0 | 0.0 | 0.0 |
| EvNet | 97.7 | **78.4** | 47.7 | 9.9 | 1.2 | 0.2 | 0.1 | | 97.7 | **93.5** | **86.9** | **62.2** | **21.5** | 3.7 | 1.0 |

Table 24: OOD detection (AU-PR) under PGD uncertainty attacks against distributional uncertainty on ID data and OOD data.

| | ID-Attack (non-attacked OOD) | | | | | | | | OOD-Attack (non-attacked ID) | | | | | | |
|---|---|---|---|---|---|---|---|---|---|---|---|---|---|---|---|
| Att. Rad. | 0.0 | 0.1 | 0.2 | 0.5 | 1.0 | 2.0 | 4.0 | | 0.0 | 0.1 | 0.2 | 0.5 | 1.0 | 2.0 | 4.0 |
| | | | | | | | **MNIST – KMNIST** | | | | | | | | |
| PostNet | 80.5 | 76.2 | 73.4 | 69.1 | 66.6 | 65.4 | **60.2** | | 80.5 | 72.1 | 63.9 | 43.9 | 33.0 | 30.9 | 30.8 |
| PriorNet | **99.6** | **99.4** | **99.2** | **98.0** | **94.1** | **76.3** | 31.2 | | **99.6** | **99.4** | **99.2** | **98.2** | **95.2** | **87.2** | **75.2** |
| DDNet | 98.4 | 98.1 | 97.7 | 95.8 | 89.5 | 56.2 | 30.9 | | 98.4 | 98.1 | 97.8 | 96.5 | 93.8 | 86.3 | 67.7 |
| EvNet | 40.1 | 39.5 | 39.1 | 38.2 | 37.3 | 36.5 | 35.6 | | 40.1 | 34.6 | 32.6 | 31.3 | 31.0 | 31.0 | 31.1 |
| | | | | | | | **CIFAR10 – SVHN** | | | | | | | | |
| PostNet | 64.2 | 44.7 | 37.5 | **31.1** | **28.5** | **25.0** | **19.3** | | 64.2 | 31.0 | 19.5 | 16.3 | 16.4 | **16.5** | **16.3** |
| PriorNet | 40.8 | 27.4 | 20.4 | 15.9 | 15.4 | 15.4 | 15.4 | | 40.8 | 28.3 | 21.1 | 15.9 | 15.4 | 15.4 | 15.4 |
| DDNet | **82.0** | **71.0** | **59.1** | 29.9 | 16.6 | 15.5 | 15.4 | | **82.0** | **72.2** | **60.3** | **26.3** | 16.2 | 15.4 | 15.4 |
| EvNet | 36.4 | 28.7 | 26.5 | 22.8 | 20.2 | 18.4 | 17.2 | | 36.4 | 19.8 | 18.3 | 17.2 | **16.9** | 16.2 | 15.7 |
| | | | | | | | **Sens. – Sens. class 10, 11** | | | | | | | | |
| PostNet | **79.1** | **40.3** | **35.9** | **33.0** | **45.5** | 45.5 | 45.5 | | **79.1** | **47.3** | **43.7** | **36.5** | **37.9** | **74.6** | **96.5** |
| PriorNet | 35.5 | 26.8 | 26.7 | 26.9 | 29.6 | 43.7 | **68.7** | | 35.5 | 27.5 | 26.9 | 26.7 | 26.6 | 26.5 | 26.5 |
| DDNet | 52.9 | 31.7 | 29.8 | 29.1 | 28.4 | 30.1 | 37.6 | | 52.9 | 38.4 | 31.5 | 27.5 | 26.8 | 26.6 | 26.5 |
| EvNet | 66.3 | 33.3 | 29.6 | 27.0 | 27.2 | 29.3 | 35.2 | | 66.3 | 39.3 | 37.1 | 31.3 | 28.3 | 28.4 | 29.7 |
| | | | | | | | **Seg. – Seg. class sky** | | | | | | | | |
| PostNet | 98.0 | 76.3 | 53.1 | **37.4** | **32.9** | 44.6 | **50.2** | | 98.0 | 83.5 | 64.8 | 41.8 | 35.4 | **43.1** | 71.3 |
| PriorNet | 32.3 | 30.9 | 30.8 | 30.8 | 30.8 | 32.5 | 45.0 | | 32.3 | 31.0 | 30.8 | 30.8 | 30.8 | 30.8 | 30.8 |
| DDNet | 30.9 | 30.8 | 30.8 | 30.8 | 30.8 | 30.8 | 30.8 | | 30.9 | 30.8 | 30.8 | 30.8 | 30.8 | 30.8 | 30.8 |
| EvNet | **98.1** | **82.1** | **59.1** | 33.8 | 31.0 | 30.8 | 30.8 | | **98.1** | **93.8** | **88.2** | **64.5** | **36.4** | 31.3 | 31.0 |

Table 25: OOD detection (AUC-ROC) under PGD uncertainty attacks against distributional uncertainty on ID data and OOD data.

| | ID-Attack (non-attacked OOD) | | | | | | | | OOD-Attack (non-attacked ID) | | | | | | |
|---|---|---|---|---|---|---|---|---|---|---|---|---|---|---|---|
| Att. Rad. | 0.0 | 0.1 | 0.2 | 0.5 | 1.0 | 2.0 | 4.0 | | 0.0 | 0.1 | 0.2 | 0.5 | 1.0 | 2.0 | 4.0 |
| | | | | | | | **MNIST – KMNIST** | | | | | | | | |
| PostNet | 90.1 | 88.0 | 86.2 | 82.2 | 79.0 | 77.1 | **66.1** | | 90.1 | 84.5 | 77.2 | 46.4 | 12.9 | 2.7 | 2.4 |
| PriorNet | **99.8** | **99.7** | **99.6** | **99.1** | **97.2** | **79.7** | 4.7 | | **99.8** | **99.7** | **99.6** | **99.2** | **97.9** | **93.7** | **85.6** |
| DDNet | 98.1 | 97.7 | 97.2 | 94.8 | 87.0 | 48.7 | 3.0 | | 98.1 | 97.8 | 97.3 | 95.8 | 92.3 | 83.3 | 63.3 |
| EvNet | 36.8 | 35.0 | 33.7 | 30.9 | 28.2 | 25.3 | 22.1 | | 36.8 | 19.3 | 10.7 | 3.9 | 2.1 | 1.8 | 2.2 |
| | | | | | | | **CIFAR10 – SVHN** | | | | | | | | |
| PostNet | 82.9 | 67.7 | 59.2 | **51.3** | **47.7** | **40.1** | **24.2** | | 82.9 | 51.9 | 26.2 | 8.9 | 9.5 | **11.1** | **9.9** |
| PriorNet | 48.0 | 33.6 | 22.5 | 7.1 | 1.6 | 0.3 | 0.1 | | 48.0 | 34.8 | 24.0 | 6.7 | 1.6 | 0.6 | 0.2 |
| DDNet | **85.9** | **74.9** | **62.7** | 30.1 | 8.3 | 2.3 | 0.6 | | **85.9** | **77.6** | **66.9** | **32.1** | 8.0 | 1.5 | 0.2 |
| EvNet | 63.3 | 51.4 | 47.1 | 39.3 | 32.1 | 24.9 | 17.9 | | 63.3 | 31.1 | 24.4 | 17.7 | **15.0** | 10.7 | 5.7 |
| | | | | | | | **Sens. – Sens. class 10, 11** | | | | | | | | |
| PostNet | **87.1** | **50.9** | **37.8** | 5.5 | 4.5 | 4.5 | 4.5 | | **87.1** | **55.3** | **51.1** | **34.4** | **38.9** | **79.7** | **97.9** |
| PriorNet | 36.5 | 2.9 | 1.8 | 1.8 | 5.2 | 21.5 | **52.8** | | 36.5 | 7.3 | 3.0 | 1.3 | 0.5 | 0.1 | 0.0 |
| DDNet | 52.3 | 18.7 | 13.1 | **10.3** | **9.3** | 10.8 | 18.4 | | 52.3 | 33.1 | 22.0 | 6.7 | 2.2 | 0.6 | 0.1 |
| EvNet | 75.5 | 30.7 | 18.1 | 5.8 | 1.6 | 0.6 | 0.8 | | 75.5 | 47.7 | 41.8 | 23.8 | 10.3 | 10.2 | 15.8 |
| | | | | | | | **Seg. – Seg. class sky** | | | | | | | | |
| PostNet | 98.6 | 78.3 | 51.9 | 20.5 | 8.3 | 2.1 | 1.7 | | 98.6 | 88.8 | 73.1 | 35.9 | **21.4** | **39.9** | 75.9 |
| PriorNet | 9.4 | 1.6 | 0.3 | 0.0 | 0.0 | 1.8 | **15.4** | | 9.4 | 2.4 | 0.4 | 0.0 | 0.0 | 0.0 | 0.0 |
| DDNet | 1.3 | 0.2 | 0.0 | 0.0 | 0.0 | 0.0 | 0.0 | | 1.3 | 0.2 | 0.0 | 0.0 | 0.0 | 0.0 | 0.0 |
| EvNet | **97.4** | **77.1** | **45.9** | 9.4 | 1.3 | 0.2 | 0.1 | | **97.4** | **92.9** | **86.1** | **60.9** | 20.4 | 3.0 | 1.2 |

Table 26: OOD detection (AU-PR) under FGSM uncertainty attacks against differential entropy on ID data and OOD data.

| Att. Rad. | ID-Attack (non-attacked OOD) | | | | | | | | OOD-Attack (non-attacked ID) | | | | | | |
|---|---|---|---|---|---|---|---|---|---|---|---|---|---|---|---|
| | 0.0 | 0.1 | 0.2 | 0.5 | 1.0 | 2.0 | 4.0 | | 0.0 | 0.1 | 0.2 | 0.5 | 1.0 | 2.0 | 4.0 |
| **MNIST – KMNIST** | | | | | | | | | | | | | | | |
| PostNet | 94.5 | 94.2 | 94.1 | 93.5 | 89.9 | 81.2 | **71.6** | | 94.5 | 93.3 | 92.0 | 87.6 | 81.1 | 75.7 | 75.7 |
| PriorNet | **99.6** | **99.4** | **99.2** | 98.1 | 95.6 | 90.0 | 65.3 | | **99.6** | **99.4** | **99.2** | **98.6** | **97.5** | **95.9** | **94.4** |
| DDNet | 99.3 | 99.1 | 98.9 | 98.0 | 95.4 | 80.9 | 48.2 | | 99.3 | 99.2 | 99.0 | 98.5 | **97.6** | 95.5 | 92.0 |
| EvNet | 69.0 | 67.4 | 66.2 | 64.0 | 61.9 | 59.8 | 56.70 | | 9.0 | 60.1 | 56.5 | 53.4 | 52.7 | 52.9 | 53.5 |
| **CIFAR10 – SVHN** | | | | | | | | | | | | | | | |
| PostNet | 81.8 | 66.2 | 61.6 | **64.2** | **65.7** | 61.3 | 48.4 | | 81.8 | 63.1 | 51.9 | 43.4 | 46.6 | **61.7** | **77.0** |
| PriorNet | 54.4 | 40.6 | 33.8 | 27.0 | 25.5 | 27.2 | 35.5 | | 54.4 | 42.3 | 36.8 | 30.6 | 28.3 | 29.5 | 32.1 |
| DDNet | **82.8** | **71.9** | **64.6** | 53.8 | 50.2 | 47.8 | 41.0 | | **82.8** | **71.5** | **60.5** | 39.1 | 31.4 | 41.2 | 66.6 |
| EvNet | 80.3 | 67.8 | 64.0 | 61.9 | 61.6 | 57.4 | **49.6** | | 80.3 | 59.2 | 51.5 | **46.7** | **49.0** | 56.3 | 64.6 |
| **Sens. – Sens. class 10, 11** | | | | | | | | | | | | | | | |
| PostNet | **74.5** | 40.6 | 37.2 | 31.4 | 38.1 | 44.9 | 45.9 | | **74.5** | 99.6 | 99.8 | 99.9 | 99.9 | 99.9 | 99.9 |
| PriorNet | 32.3 | 35.7 | **57.6** | 83.1 | 88.8 | 79.7 | 70.0 | | 32.3 | 28.3 | 28.1 | 27.6 | 28.0 | 32.7 | 38.5 |
| DDNet | 31.7 | 31.3 | 44.4 | 70.3 | 87.9 | **92.5** | 91.9 | | 31.7 | 28.8 | 29.3 | 29.1 | 27.7 | 27.9 | 28.01 |
| EvNet | 66.5 | **45.7** | 46.8 | 42.3 | 42.0 | 41.4 | 41.8 | | 66.5 | 54.7 | 66.5 | 76.2 | 71.1 | 75.3 | 75.8 |
| **Seg. – Seg. class sky** | | | | | | | | | | | | | | | |
| PostNet | **99.0** | 80.8 | 66.4 | 43.6 | 37.0 | 35.5 | 43.0 | | **99.0** | 94.8 | 92.0 | 98.5 | 99.7 | 100.0 | 100.0 |
| PriorNet | 34.8 | 31.2 | 31.4 | 46.3 | **74.0** | **88.8** | **94.5** | | 34.8 | 31.6 | 31.2 | 30.9 | 30.8 | 30.8 | 30.8 |
| DDNet | 31.5 | 30.8 | 30.8 | 30.9 | 37.9 | 56.2 | 84.3 | | 31.5 | 30.9 | 30.8 | 30.8 | 30.8 | 30.8 | 30.8 |
| EvNet | 92.5 | 64.9 | 54.6 | **66.6** | 69.5 | 69.6 | 64.6 | | 92.5 | 85.9 | 83.0 | 66.3 | 66.1 | 61.1 | 56.8 |

Table 27: OOD detection (AU-PR) under Noise uncertainty attacks against differential entropy on ID data and OOD data.

| Noise Std | ID-Attack (non-attacked OOD) | | | | | | | | OOD-Attack (non-attacked ID) | | | | | | |
|---|---|---|---|---|---|---|---|---|---|---|---|---|---|---|---|
| | 0.0 | 0.1 | 0.2 | 0.5 | 1.0 | 2.0 | 4.0 | | 0.0 | 0.1 | 0.2 | 0.5 | 1.0 | 2.0 | 4.0 |
| **MNIST – KMNIST** | | | | | | | | | | | | | | | |
| PostNet | 93.0 | 94.2 | 82.3 | 34.4 | 31.6 | 31.0 | 30.9 | | 92.2 | 91.8 | 91.5 | 92.3 | 92.7 | 93.2 | 93.5 |
| PriorNet | **99.7** | **99.6** | **96.7** | **40.0** | **40.6** | **45.7** | **55.6** | | 99.5 | 97.3 | 96.5 | 99.4 | **100.0** | 99.5 | 72.4 |
| DDNet | 99.1 | 97.5 | 81.2 | 31.3 | 31.0 | 30.9 | 31.2 | | 99.0 | 98.8 | 99.2 | 99.8 | 99.9 | 99.8 | 99.1 |
| EvNet | 65.5 | 60.5 | 51.4 | 35.3 | 34.5 | 35.5 | 35.0 | | 62.5 | 47.2 | 40.9 | 35.1 | 34.6 | 33.5 | 34.9 |
| **CIFAR10 – SVHN** | | | | | | | | | | | | | | | |
| PostNet | 88.5 | 41.4 | 39.8 | 31.0 | 30.7 | 31.6 | 33.9 | | 88.5 | **86.6** | **81.9** | **93.0** | **98.5** | 98.6 | 97.3 |
| PriorNet | 73.3 | **88.3** | **95.3** | **92.4** | **70.4** | 30.9 | 30.8 | | 73.3 | 31.6 | 30.9 | 31.7 | 51.8 | 94.3 | **100.0** |
| DDNet | 87.3 | 69.3 | 78.4 | 55.2 | 31.6 | 30.7 | 31.4 | | 87.3 | 55.8 | 57.9 | 73.9 | 97.3 | **99.5** | 97.2 |
| EvNet | **92.4** | 56.8 | 53.8 | 33.4 | 30.9 | **32.9** | **36.6** | | **92.4** | 73.7 | 73.5 | 77.7 | 93.7 | 92.5 | 92.1 |
| **Sens. – Sens. class 10, 11** | | | | | | | | | | | | | | | |
| PostNet | **85.3** | 30.8 | **39.4** | 50.0 | 50.0 | 50.0 | 50.0 | | **85.3** | 98.9 | 100.0 | 100.0 | 100.0 | 100.0 | 100.0 |
| PriorNet | 32.3 | 30.8 | 34.9 | **83.7** | **77.7** | 49.8 | **80.3** | | 32.3 | 30.7 | 30.7 | 32.5 | 40.1 | 49.9 | 47.6 |
| DDNet | 31.1 | 30.7 | 30.7 | 32.4 | 58.8 | **88.1** | 74.3 | | 31.1 | 30.7 | 30.7 | 30.7 | 30.8 | 31.6 | 39.1 |
| EvNet | 80.3 | **30.8** | 31.2 | 37.9 | 46.3 | 50.0 | 50.0 | | 80.3 | 34.6 | 38.4 | 53.9 | 69.3 | 78.8 | 81.5 |
| **Seg. – Seg. class sky** | | | | | | | | | | | | | | | |
| PostNet | **99.9** | 41.8 | 30.8 | **34.5** | **49.1** | 50.0 | 50.0 | | **99.9** | 97.4 | 96.6 | 99.5 | 100.0 | 100.0 | 100.0 |
| PriorNet | 31.0 | 30.8 | 30.8 | 30.8 | 32.7 | **69.0** | 78.3 | | 31.0 | 30.8 | 30.8 | 30.8 | 30.9 | 31.1 | 32.4 |
| DDNet | 30.8 | 30.8 | 30.8 | 30.8 | 30.8 | 58.2 | **91.3** | | 30.8 | 30.8 | 30.8 | 30.8 | 30.8 | 30.8 | 31.9 |
| EvNet | 99.1 | 38.1 | **32.2** | 30.8 | 30.8 | 32.2 | 37.5 | | 99.1 | 95.6 | 87.6 | 58.0 | 44.9 | 46.6 | 53.8 |

## A.5 ROBUST TRAINING FOR DBU MODELS & ID/OOD VERIFICATION

Table 5 and 29 on adversarial training illustrate that there is a jump between ID-verification and OOD-verification, where robustness on ID data drops while robustness on OOD data increases. These jumps are observed for each model and each training (normal, noise-based, adversarial with label attacks, adversarial with uncertainty attacks). Thus, either ID-verification or OOD-verification perform well, depending on the chosen threshold.

In contrast to that, adversarial training improves robustness w.r.t. the predicted class label for most pair model/data set (Fig. 7, 32).

Table 28: Randomized smoothing verification of CIFAR10 (ID data) and SVHN (OOD data) harmonic mean.

|  | 0.1 | 0.2 | 0.5 |
|---|---|---|---|
| **adv. train. loss: None** | | | |
| PriorNet | 26.7 | 3.7 | 0.0 |
| PostNet | 35.9 | 34.1 | 0.0 |
| DDNet | 45.2 | 18.1 | **46.6** |
| EvNet | **47.6** | **45.4** | 22.6 |
| **adv. train. loss: crossentropy** | | | |
| PriorNet | 0.2 | 0.0 | **41.4** |
| PostNet | 34.4 | **47.9** | 0.0 |
| DDNet | **49.2** | 44.3 | 0.0 |
| EvNet | 41.1 | 22.4 | 4.7 |
| **adv. train. loss: diffE** | | | |
| PriorNet | 2.2 | 0.0 | 0.0 |
| PostNet | 41.9 | 11.4 | 0.0 |
| DDNet | 46.2 | 8.4 | 0.0 |
| EvNet | **47.3** | **34.6** | **2.0** |

Table 29: Randomized smoothing verification of MNIST (ID data) and KMNIST (OOD data): percentage of samples that is certifiably correct (cc) and mean certified radius (R).

| $\sigma$ | ID-Verification | | | | | | OOD-Verification | | | | | |
|---|---|---|---|---|---|---|---|---|---|---|---|---|
| | 0.1 | | 0.2 | | 0.5 | | 0.1 | | 0.2 | | 0.5 | |
| | cc | R | cc | R | cc | R | cc | R | cc | R | cc | R |
| **adv. train. loss: None** | | | | | | | | | | | | |
| PriorNet | **97.0** | **0.36** | **88.2** | **0.52** | 3.0 | 0.20 | 98.7 | 0.37 | 99.5 | 0.74 | 100.0 | 1.88 |
| PostNet | 93.2 | 0.32 | 68.4 | 0.31 | 0.8 | 0.11 | 98.4 | 0.36 | 99.5 | 0.68 | 100.0 | 1.55 |
| DDNet | 90.6 | 0.35 | 52.3 | 0.46 | 0.0 | 0.00 | 97.8 | 0.37 | **99.5** | **0.74** | **100.0** | **1.90** |
| EvNet | 95.0 | 0.31 | 83.0 | 0.30 | **17.3** | **0.21** | 77.3 | 0.17 | 82.7 | 0.24 | 88.6 | 0.39 |
| **adv. train. loss: crossentropy** | | | | | | | | | | | | |
| PriorNet | **97.0** | **0.36** | **94.3** | **0.58** | 1.0 | 0.15 | **99.8** | **0.38** | 99.5 | 0.74 | 100.0 | 1.89 |
| PostNet | 94.4 | 0.31 | 57.7 | 0.32 | 3.2 | 0.13 | 97.2 | 0.33 | 95.6 | 0.51 | 99.6 | 1.02 |
| DDNet | 82.6 | 0.34 | 55.5 | 0.46 | 0.0 | 0.00 | 99.6 | 0.38 | **100.0** | **0.75** | **100.0** | **1.90** |
| EvNet | 96.8 | 0.34 | 70.1 | 0.27 | **18.8** | **0.25** | 58.7 | 0.11 | 85.2 | 0.24 | 89.1 | 0.26 |
| **adv. train. loss: diffE** | | | | | | | | | | | | |
| PriorNet | **98.0** | **0.37** | 83.4 | 0.49 | 0.7 | 0.10 | **99.7** | **0.38** | 100.0 | 0.76 | 100.0 | 1.90 |
| PostNet | 93.5 | 0.33 | 47.1 | 0.23 | 0.6 | 0.15 | 95.8 | 0.34 | 98.8 | 0.63 | 100.0 | 1.38 |
| DDNet | 93.6 | 0.36 | 52.7 | 0.43 | 0.0 | 0.00 | 97.7 | 0.37 | 99.7 | 0.75 | **100.0** | **1.90** |
| EvNet | 95.4 | 0.33 | 81.6 | 0.34 | **23.1** | **0.63** | 81.7 | 0.20 | 82.8 | 0.28 | 99.1 | 1.70 |

Table 30: Randomized smoothing verification of MNIST (ID data) and KMNIST (OOD data): percentage of samples that is certifiably wrong (cw) and mean certified radius (R).

| | 0.1 | | 0.2 | | 0.5 | |
|---|---|---|---|---|---|---|
| | cw | R | cw | R | cw | R |
| adv. train. loss: None | | | | | | |
| PriorNet | **2.8** | **0.16** | **10.7** | **0.21** | 96.0 | 0.97 |
| PostNet | 6.4 | 0.17 | 28.8 | 0.22 | 99.0 | 1.15 |
| DDNet | 9.1 | 0.24 | 46.3 | 0.42 | 100.0 | 1.81 |
| EvNet | 4.5 | 0.10 | 15.1 | 0.13 | **78.8** | **0.31** |
| adv. train. loss: crossentropy | | | | | | |
| PriorNet | **2.9** | **0.20** | **4.9** | **0.24** | 98.4 | 1.05 |
| PostNet | 5.3 | 0.17 | 38.8 | 0.23 | 95.2 | 0.93 |
| DDNet | 16.4 | 0.25 | 43.5 | 0.41 | 100.0 | 1.74 |
| EvNet | 3.0 | 0.08 | 26.3 | 0.13 | **76.3** | **0.27** |
| adv. train. loss: diffE | | | | | | |
| PriorNet | **2.0** | **0.19** | **15.7** | **0.25** | 98.8 | 1.10 |
| PostNet | 6.3 | 0.17 | 49.8 | 0.25 | 99.1 | 1.10 |
| DDNet | 6.2 | 0.22 | 46.2 | 0.42 | 100.0 | 1.81 |
| EvNet | 4.2 | 0.14 | 17.0 | 0.16 | **73.9** | **0.94** |

Table 31: Randomized smoothing verification of MNIST (ID data) and KMNIST (OOD data) harmonic mean.

| | 0.1 | 0.2 | 0.5 |
|---|---|---|---|
| adv. train. loss: None | | | |
| PriorNet | 5.5 | 19.1 | 5.9 |
| PostNet | 12.0 | 40.5 | 1.5 |
| DDNet | **16.5** | **49.2** | 0.0 |
| EvNet | 8.7 | 25.6 | **28.4** |
| adv. train. loss: crossentropy | | | |
| PriorNet | 5.6 | 9.3 | 2.0 |
| PostNet | 10.0 | 46.4 | 6.2 |
| DDNet | **27.4** | **48.8** | 0.0 |
| EvNet | 5.8 | 38.2 | **30.2** |
| adv. train. loss: diffE | | | |
| PriorNet | 3.9 | 26.4 | 1.4 |
| PostNet | **11.8** | 48.4 | 1.2 |
| DDNet | 11.6 | **49.2** | 0.0 |
| EvNet | 8.0 | 28.1 | **35.2** |

Table 32: Adversarial training with CE: Accuracy under PGD label attacks (AUC-PR).

| Att. Rad. | 0.1 | 0.2 | 0.5 | 1.0 | 2.0 | 4.0 | | 0.1 | 0.2 | 0.5 | 1.0 | 2.0 | 4.0 |
|---|---|---|---|---|---|---|---|---|---|---|---|---|---|
| | | | MNIST | | | | | | | CIFAR10 | | | |
| PostNet | **99.1** | 98.7 | 96.7 | 89.3 | 62.4 | 14.8 | | 72.1 | 50.6 | 12.8 | 2.4 | 0.2 | 0.1 |
| PriorNet | **99.1** | 98.8 | **97.6** | **94.8** | **91.1** | **79.5** | | 69.6 | **63.9** | **55.8** | **46.4** | **30.9** | **11.1** |
| DDNet | **99.1** | **98.9** | 97.4 | 90.6 | 47.0 | 0.2 | | 74.7 | **63.9** | 23.6 | 2.0 | 0.0 | 0.0 |
| EvNet | 80.3 | 98.3 | 96.7 | 90.5 | 60.0 | 52.7 | | **78.3** | 51.0 | 14.6 | 2.6 | 2.9 | 0.7 |
| | | | Sensorless | | | | | | | Segment | | | |
| PostNet | 15.5 | 6.4 | 4.7 | **6.7** | **11.1** | **11.7** | | 84.5 | 52.4 | **21.1** | 7.6 | **5.0** | **6.3** |
| PriorNet | 31.3 | 15.8 | 0.2 | 0.0 | 0.3 | 5.3 | | **94.0** | **65.2** | 19.1 | 0.6 | 0.0 | 0.0 |
| DDNet | 12.4 | 4.2 | 0.2 | 0.3 | 0.2 | 0.1 | | 91.4 | 46.2 | 7.4 | 0.2 | 0.0 | 0.0 |
| EvNet | **33.6** | **19.4** | **8.3** | 5.4 | 2.6 | 1.7 | | 93.0 | 55.2 | 15.5 | 2.0 | 1.4 | 1.4 |

Table 33: Adversarial training with CE: Accuracy under FGSM label attacks (AUC-PR).

| Att. Rad. | 0.1 | 0.2 | 0.5 | 1.0 | 2.0 | 4.0 | | 0.1 | 0.2 | 0.5 | 1.0 | 2.0 | 4.0 |
|---|---|---|---|---|---|---|---|---|---|---|---|---|---|
| | | | MNIST | | | | | | | CIFAR10 | | | |
| PostNet | **99.1** | 98.8 | 97.6 | 95.3 | 90.1 | 81.3 | | 71.0 | 54.4 | 30.4 | 19.9 | 17.0 | 17.7 |
| PriorNet | **99.1** | **98.9** | **97.9** | **96.2** | **94.7** | **90.0** | | 69.7 | **65.3** | **58.8** | **52.1** | **38.9** | **22.2** |
| DDNet | **99.1** | **98.9** | 97.8 | 94.6 | 79.1 | 27.9 | | **73.7** | 64.7 | 34.8 | 15.5 | 8.0 | 4.8 |
| EvNet | 80.3 | 98.5 | 97.6 | 94.0 | 72.7 | 81.3 | | 48.0 | 56.9 | 27.0 | 17.1 | 17.8 | 15.7 |
| | | | Sensorless | | | | | | | Segment | | | |
| PostNet | 20.6 | 10.6 | 11.0 | 11.8 | 12.5 | 12.5 | | 82.9 | 60.1 | **27.5** | **22.7** | **19.1** | 24.3 |
| PriorNet | 35.0 | 20.8 | 0.5 | 0.1 | 0.9 | 7.5 | | 91.2 | 55.2 | 19.0 | 0.7 | 0.0 | 0.0 |
| DDNet | 16.4 | 9.7 | 7.0 | 4.6 | 6.4 | 7.5 | | 86.8 | 36.4 | 10.5 | 0.8 | 0.0 | 0.0 |
| EvNet | **41.1** | **27.4** | **20.1** | **15.2** | **14.9** | 12.6 | | **93.8** | **64.2** | 25.0 | 1.5 | 0.0 | 0.4 |

Table 34: Adversarial training with CE: Accuracy under Noise label attacks (AUC-PR).

| Att. Rad. | 0.1 | 0.2 | 0.5 | 1.0 | 2.0 | 4.0 | | 0.1 | 0.2 | 0.5 | 1.0 | 2.0 | 4.0 |
|---|---|---|---|---|---|---|---|---|---|---|---|---|---|
| | | | MNIST | | | | | | | CIFAR10 | | | |
| PostNet | **97.8** | 84.3 | 8.4 | **3.2** | 0.0 | 0.0 | | 19.3 | 7.3 | 3.4 | 0.0 | **10.2** | **10.1** |
| PriorNet | 97.7 | **94.6** | **32.7** | 0.0 | 0.0 | 0.0 | | **39.1** | **16.8** | **3.8** | **10.5** | 3.7 | 0.4 |
| DDNet | 97.6 | 89.1 | 16.5 | 2.1 | 0.0 | **0.3** | | 25.8 | 15.1 | 2.2 | 0.2 | 9.8 | 8.9 |
| EvNet | 94.1 | 74.3 | 5.0 | 0.0 | 0.0 | 0.0 | | 27.1 | 7.1 | 0.1 | 2.8 | 8.6 | 9.7 |
| | | | Sensorless | | | | | | | Segment | | | |
| PostNet | 1.0 | **4.7** | **11.4** | **11.7** | **11.7** | **11.7** | | 27.6 | 1.5 | **3.6** | **15.2** | **20.9** | **21.2** |
| PriorNet | **4.7** | 0.0 | 0.0 | 0.1 | 0.0 | 2.2 | | **56.4** | **16.1** | 2.1 | 0.9 | 0.0 | 0.0 |
| DDNet | 0.3 | 0.0 | 0.0 | 0.0 | 0.0 | 0.1 | | 49.4 | 6.1 | 0.0 | 0.0 | 0.0 | 0.0 |
| EvNet | 0.9 | 0.0 | 0.9 | 0.2 | 3.5 | 3.1 | | 51.2 | 10.6 | 0.3 | 0.0 | 0.0 | 0.6 |

Table 35: Randomized smoothing verification of CIFAR10: percentage of samples that is certifiably correct (cc) w.r.t. the predicted class label and mean certified radius (R) w.r.t. class labels.

| | 0.1 | | 0.2 | | 0.5 | |
|---|---|---|---|---|---|---|
| | cc | R | cc | R | cc | R |
| | | adv. train. loss: None | | | | |
| PriorNet | **42.8** | **0.25** | **21.2** | **0.42** | **11.8** | 1.30 |
| PostNet | 35.0 | 0.22 | 12.3 | 0.51 | 9.4 | 0.12 |
| DDNet | 31.7 | 0.26 | 12.2 | 0.69 | 10.8 | 1.91 |
| EvNet | 34.3 | 0.22 | 15.4 | 0.42 | 11.0 | 0.63 |
| | | adv. train. loss: crossentropy | | | | |
| PriorNet | **56.2** | **0.25** | **25.4** | **0.48** | **13.0** | **0.35** |
| PostNet | 34.7 | 0.22 | 15.6 | 0.45 | 11.0 | 0.32 |
| DDNet | 41.7 | 0.24 | 19.6 | 0.44 | 9.1 | 1.30 |
| EvNet | 34.3 | 0.16 | 11.1 | 0.55 | 10.8 | 0.74 |
| | | adv. train. loss: diffE | | | | |
| PriorNet | 48.1 | 0.23 | **28.0** | **0.40** | 8.4 | 0.22 |
| PostNet | 45.5 | 0.21 | 18.0 | 0.36 | 5.4 | 0.18 |
| DDNet | **49.2** | **0.25** | 26.3 | 0.34 | 9.6 | 0.27 |
| EvNet | 21.9 | 0.30 | 15.2 | 0.24 | **10.8** | **1.06** |

Table 36: Randomized smoothing verification of MNIST: percentage of samples that is certifiably correct (cc) w.r.t. the predicted class label and mean certified radius (R) w.r.t. class labels.

| | 0.1 | | 0.2 | | 0.5 | |
|---|---|---|---|---|---|---|
| | cc | R | cc | R | cc | R |
| | | adv. train. loss: None | | | | |
| PriorNet | 99.2 | 0.38 | **98.8** | **0.71** | **61.4** | **0.45** |
| PostNet | 99.2 | 0.38 | 98.1 | 0.66 | 51.2 | 0.51 |
| DDNet | **99.3** | **0.38** | 98.0 | 0.68 | 47.3 | 0.52 |
| EvNet | 98.9 | 0.37 | 96.2 | 0.56 | 57.1 | 0.42 |
| | | adv. train. loss: crossentropy | | | | |
| PriorNet | 99.1 | 0.38 | **99.0** | **0.72** | 50.4 | 0.53 |
| PostNet | **99.4** | **0.38** | 97.4 | 0.62 | 28.8 | 0.51 |
| DDNet | 99.3 | 0.38 | 98.6 | 0.69 | **75.4** | **0.64** |
| EvNet | 99.1 | 0.37 | 92.1 | 0.43 | 35.0 | 0.40 |
| | | adv. train. loss: diffE | | | | |
| PriorNet | 99.5 | 0.38 | **98.3** | **0.71** | 64.0 | 0.48 |
| PostNet | 99.1 | 0.38 | 96.8 | 0.62 | 48.1 | 0.44 |
| DDNet | **99.6** | **0.38** | 98.1 | 0.69 | 32.4 | 0.64 |
| EvNet | 99.1 | 0.37 | 96.7 | 0.59 | **89.5** | **0.93** |

Table 37: Adversarial training with CE: Certainty based on differential entropy under PGD label attacks (AUC-PR).

| Att. Rad. | 0.1 | 0.2 | 0.5 | 1.0 | 2.0 | 4.0 | | 0.1 | 0.2 | 0.5 | 1.0 | 2.0 | 4.0 |
|---|---|---|---|---|---|---|---|---|---|---|---|---|---|
| | | | MNIST | | | | | | | CIFAR10 | | | |
| PostNet | 99.9 | 99.8 | 98.5 | 88.7 | 47.6 | 9.0 | | 88.1 | 54.1 | 7.5 | 1.3 | 0.1 | 0.1 |
| PriorNet | 99.8 | 99.4 | 97.7 | 92.4 | 79.7 | 67.5 | | 54.0 | 45.5 | 37.9 | 29.7 | 18.1 | 6.1 |
| DDNet | 100.0 | 99.9 | 99.7 | 96.9 | 46.1 | 0.1 | | 92.1 | 83.1 | 24.8 | 1.1 | 0.0 | 0.0 |
| EvNet | 81.2 | 98.4 | 95.5 | 90.4 | 53.2 | 38.3 | | 62.9 | 59.2 | 13.1 | 1.5 | 1.6 | 0.4 |
| | | | Sensorless | | | | | | | Segment | | | |
| PostNet | 8.8 | 4.2 | **4.6** | **6.7** | **11.1** | **11.7** | | 76.1 | 35.8 | 12.6 | **4.9** | **4.9** | **6.3** |
| PriorNet | **22.6** | **11.7** | 0.2 | 0.0 | 0.3 | 3.6 | | **98.1** | **66.2** | **12.8** | 0.6 | 0.0 | 0.0 |
| DDNet | 10.9 | 3.0 | 0.1 | 0.2 | 0.1 | 0.1 | | 95.9 | 52.6 | 4.5 | 0.5 | 0.0 | 0.0 |
| EvNet | 21.4 | 11.0 | 4.4 | 3.1 | 1.7 | 1.4 | | 94.8 | 42.7 | 8.9 | 1.2 | 1.3 | 1.3 |

Table 38: Adversarial training with CE: Certainty based on differential entropy under FGSM label attacks (AUC-PR).

| Att. Rad. | 0.1 | 0.2 | 0.5 | 1.0 | 2.0 | 4.0 | 0.1 | 0.2 | 0.5 | 1.0 | 2.0 | 4.0 |
|---|---|---|---|---|---|---|---|---|---|---|---|---|
| | | | MNIST | | | | | | CIFAR10 | | | |
| PostNet | 99.9 | 99.8 | 99.5 | 97.4 | **90.7** | 83.5 | 88.0 | 66.6 | 28.8 | 18.0 | 15.5 | **20.4** |
| PriorNet | **99.8** | **99.5** | 98.2 | 94.6 | 89.3 | **83.5** | 54.7 | 48.1 | 42.3 | **36.8** | 27.3 | 19.8 |
| DDNet | 100.0 | 99.9 | 99.7 | **98.7** | 87.3 | 27.4 | **91.3** | **83.4** | **45.4** | 15.0 | 6.5 | 3.6 |
| EvNet | 81.2 | 98.7 | 97.4 | 95.9 | 73.4 | 80.7 | 62.4 | 68.7 | 28.6 | 15.1 | 20.5 | 23.1 |
| | | | Sensorless | | | | | | Segment | | | |
| PostNet | 12.5 | 7.5 | 9.9 | 11.6 | 12.5 | 12.5 | 75.1 | 46.2 | **20.5** | **19.5** | **20.2** | 26.3 |
| PriorNet | 30.0 | 22.2 | 0.7 | 0.5 | 3.2 | 8.7 | **96.1** | **55.6** | 13.5 | 2.1 | 0.0 | 0.0 |
| DDNet | 17.5 | 6.7 | 6.2 | 3.1 | 4.8 | 5.5 | 92.0 | 42.0 | 6.4 | 2.3 | 0.0 | 0.0 |
| EvNet | **41.6** | **25.1** | **18.9** | **13.2** | **14.6** | **13.9** | 93.1 | 55.1 | 15.5 | 1.6 | 0.0 | 1.3 |

Table 39: Adversarial training with CE: Certainty based on differential entropy under Noise label attacks (AUC-PR).

| Att. Rad. | 0.1 | 0.2 | 0.5 | 1.0 | 2.0 | 4.0 | 0.1 | 0.2 | 0.5 | 1.0 | 2.0 | 4.0 |
|---|---|---|---|---|---|---|---|---|---|---|---|---|
| | | | MNIST | | | | | | CIFAR10 | | | |
| PostNet | **100.0** | 98.8 | 54.7 | 12.1 | 0.0 | 0.0 | 52.1 | 22.7 | 6.6 | 0.0 | 9.8 | 9.0 |
| PriorNet | **100.0** | **99.9** | **88.5** | 0.0 | 0.0 | 0.0 | 34.9 | 11.2 | 7.8 | **8.7** | **10.0** | 0.4 |
| DDNet | 99.8 | 98.5 | 77.2 | **15.1** | 0.0 | **0.4** | **81.6** | **45.3** | 4.2 | 0.2 | 9.6 | 8.6 |
| EvNet | 98.4 | 86.9 | 13.3 | 0.0 | 0.0 | 0.0 | 54.5 | 17.6 | 0.1 | 3.7 | 8.3 | **10.5** |
| | | | Sensorless | | | | | | Segment | | | |
| PostNet | 0.6 | **5.1** | **12.2** | **11.7** | **11.7** | **11.7** | 36.7 | 2.0 | 3.6 | **17.2** | **20.8** | **21.3** |
| PriorNet | **8.5** | 0.0 | 0.0 | 0.2 | 0.0 | 2.0 | **90.5** | **32.8** | **7.1** | 1.2 | 0.0 | 0.0 |
| DDNet | 1.5 | 0.0 | 0.0 | 0.0 | 0.0 | 0.0 | 79.6 | 21.8 | 0.0 | 0.0 | 0.0 | 0.0 |
| EvNet | 1.5 | 0.0 | 1.0 | 0.2 | 4.9 | 4.8 | 75.7 | 22.0 | 3.2 | 0.0 | 0.0 | 0.7 |

Table 40: Adversarial training with CE: Attack-Detection based on differential entropy under PGD label attacks (AUC-PR).

| Att. Rad. | 0.1 | 0.2 | 0.5 | 1.0 | 2.0 | 4.0 | 0.1 | 0.2 | 0.5 | 1.0 | 2.0 | 4.0 |
|---|---|---|---|---|---|---|---|---|---|---|---|---|
| | | | MNIST | | | | | | CIFAR10 | | | |
| PostNet | 57.8 | 67.0 | 84.1 | 91.0 | 76.8 | 47.9 | **62.5** | **66.7** | 41.6 | 35.0 | 37.5 | 36.6 |
| PriorNet | **71.7** | **83.8** | **96.5** | **96.0** | 90.0 | 79.3 | 54.4 | 55.2 | 54.8 | **51.1** | **45.9** | 40.6 |
| DDNet | 54.4 | 57.4 | 69.9 | 86.4 | **96.2** | **86.3** | 56.7 | 62.4 | **60.8** | 39.3 | 32.9 | 31.8 |
| EvNet | 52.9 | 59.7 | 67.7 | 71.9 | 66.5 | 58.5 | 52.4 | 59.0 | 48.9 | 41.7 | 40.5 | **40.7** |
| | | | Sensorless | | | | | | Segment | | | |
| PostNet | 43.7 | 41.1 | **38.4** | **53.0** | **83.5** | **98.7** | 94.2 | 73.5 | 47.7 | 42.7 | **56.8** | 70.7 |
| PriorNet | **60.9** | **47.5** | 35.8 | 31.1 | 30.8 | 34.5 | 86.2 | **90.1** | **59.5** | **47.6** | 34.0 | 30.8 |
| DDNet | 53.1 | 43.3 | 34.7 | 33.0 | 31.1 | 32.6 | 76.6 | 83.0 | 45.7 | 32.7 | 30.8 | 30.8 |
| EvNet | 48.3 | 42.1 | 37.7 | 36.6 | 39.2 | 48.5 | **95.9** | 79.6 | 43.3 | 33.4 | 31.3 | 31.2 |

Table 41: Adversarial training with CE: Attack-Detection based on differential entropy under FGSM label attacks (AUC-PR).

| Att. Rad. | 0.1 | 0.2 | 0.5 | 1.0 | 2.0 | 4.0 | 0.1 | 0.2 | 0.5 | 1.0 | 2.0 | 4.0 |
|---|---|---|---|---|---|---|---|---|---|---|---|---|
| | | | MNIST | | | | | | CIFAR10 | | | |
| PostNet | 56.1 | 62.8 | 75.5 | 86.3 | 90.4 | 92.6 | **63.0** | **68.9** | 66.1 | 63.2 | 66.2 | **77.5** |
| PriorNet | **68.6** | **80.6** | **96.8** | **98.0** | **98.4** | 98.2 | 55.9 | 59.0 | 63.0 | 65.4 | 63.4 | 58.4 |
| DDNet | 54.4 | 57.4 | 69.5 | 84.0 | 95.5 | **99.0** | 57.6 | 63.7 | 70.3 | **69.0** | **73.4** | 76.4 |
| EvNet | 52.6 | 57.9 | 62.9 | 66.0 | 64.0 | 70.0 | 52.4 | 60.0 | 59.0 | 61.2 | 62.9 | 72.3 |
| | | | Sensorless | | | | | | Segment | | | |
| PostNet | **98.3** | **99.8** | **99.9** | **100.0** | **99.9** | **100.0** | **98.0** | **99.8** | **99.9** | **100.0** | **99.9** | **100.0** |
| PriorNet | 78.6 | 68.1 | 37.6 | 32.0 | 30.7 | 49.1 | 68.8 | 60.0 | 42.9 | 31.4 | 30.7 | 32.3 |
| DDNet | 60.9 | 55.5 | 41.0 | 34.6 | 31.7 | 32.7 | 61.3 | 51.7 | 39.0 | 33.8 | 31.5 | 32.5 |
| EvNet | 70.0 | 70.4 | 67.5 | 63.0 | 77.2 | 76.6 | 69.5 | 70.0 | 66.9 | 62.4 | 77.2 | 76.4 |

Table 42: Adversarial training with CE: Attack-Detection based on differential entropy under Noise label attacks (AUC-PR).

| Att. Rad. | 0.1 | 0.2 | 0.5 | 1.0 | 2.0 | 4.0 | 0.1 | 0.2 | 0.5 | 1.0 | 2.0 | 4.0 |
|---|---|---|---|---|---|---|---|---|---|---|---|---|
| | | | MNIST | | | | | | CIFAR10 | | | |
| PostNet | **59.0** | 66.4 | 97.2 | 95.7 | 95.9 | 99.6 | **80.5** | **89.2** | 95.2 | **99.5** | 85.7 | 99.7 |
| PriorNet | 31.8 | 33.8 | 61.3 | 99.5 | **100.0** | 95.8 | 52.2 | 50.2 | 31.2 | 54.4 | **99.8** | **100.0** |
| DDNet | 51.8 | **86.4** | **99.8** | **100.0** | **100.0** | 99.6 | 80.3 | 88.4 | **99.7** | 98.8 | 99.4 | 68.4 |
| EvNet | 51.7 | 58.6 | 85.3 | 84.9 | 66.3 | **100.0** | 46.9 | 68.1 | 93.4 | 94.9 | 71.4 | 77.8 |
| | | | Sensorless | | | | | | Segment | | | |
| PostNet | **99.9** | **100.0** | **100.0** | **100.0** | **99.9** | **100.0** | 93.2 | 99.3 | **99.9** | **100.0** | **100.0** | **100.0** |
| PriorNet | 56.7 | 42.9 | 32.0 | 30.8 | 31.0 | 30.7 | 68.8 | 60.0 | 42.9 | 31.4 | 30.7 | 32.3 |
| DDNet | 51.2 | 43.9 | 33.8 | 32.4 | 31.7 | 35.0 | 61.3 | 51.7 | 39.0 | 33.8 | 31.5 | 32.5 |
| EvNet | 69.2 | 58.6 | 71.7 | 52.3 | 70.9 | 77.6 | 69.5 | 70.0 | 66.9 | 62.4 | 77.2 | 76.4 |

Table 43: Adversarial training with CE: OOD detection based on differential entropy under PGD uncertainty attacks against differential entropy on ID data and OOD data (AUC-PR).

| | ID-Attack (non-attacked OOD) | | | | | | | OOD-Attack (non-attacked ID) | | | | | |
|---|---|---|---|---|---|---|---|---|---|---|---|---|---|
| Att. Rad. | 0.1 | 0.2 | 0.5 | 1.0 | 2.0 | 4.0 | | 0.1 | 0.2 | 0.5 | 1.0 | 2.0 | 4.0 |
| **MNIST – KMNIST** | | | | | | | | | | | | | |
| PostNet | 95.7 | 93.1 | 88.3 | 78.1 | 46.9 | 32.1 | | 94.8 | 90.3 | 78.6 | 58.7 | 46.4 | 41.2 |
| PriorNet | **99.6** | **99.3** | **98.1** | **95.4** | **86.7** | **62.6** | | 99.7 | 99.3 | 98.3 | 90.7 | 77.7 | 37.3 |
| DDNet | 99.0 | 98.9 | 97.8 | 91.7 | 58.6 | 30.7 | | 99.1 | 99.0 | **98.4** | **96.2** | **90.8** | **75.7** |
| EvNet | 71.3 | 66.9 | 60.6 | 64.4 | 50.4 | 42.7 | | 66.3 | 47.8 | 37.4 | 46.7 | 37.3 | 33.3 |
| **CIFAR10 – SVHN** | | | | | | | | | | | | | |
| PostNet | 65.1 | 45.6 | 21.0 | **17.7** | **16.4** | 15.5 | | 63.8 | 41.1 | 19.6 | 19.4 | 17.0 | 16.1 |
| PriorNet | 17.0 | 16.6 | 16.0 | 15.9 | 16.0 | **16.1** | | 17.1 | 16.4 | 15.8 | 15.8 | 15.6 | 15.7 |
| DDNet | **70.8** | **63.5** | **34.0** | 16.8 | 15.5 | 15.4 | | **72.7** | **64.8** | 28.3 | 17.9 | 15.4 | 15.4 |
| EvNet | 53.9 | 43.7 | 24.2 | 16.6 | 16.1 | 15.5 | | 55.8 | 34.7 | **29.6** | **21.5** | **22.0** | **22.5** |
| **Sens. – Sens. class 10, 11** | | | | | | | | | | | | | |
| PostNet | **40.5** | **37.3** | **43.8** | **46.7** | **47.3** | **45.8** | | **42.6** | **41.7** | **31.7** | **38.5** | **81.9** | **99.3** |
| PriorNet | 26.6 | 26.6 | 26.5 | 26.5 | 30.8 | 40.0 | | 27.9 | 27.7 | 26.5 | 26.5 | 26.5 | 26.5 |
| DDNet | 26.6 | 26.6 | 26.5 | 26.5 | 26.6 | 28.2 | | 26.6 | 26.6 | 26.8 | 26.7 | 26.6 | 26.7 |
| EvNet | 31.8 | 29.7 | 27.2 | 28.0 | 32.8 | 37.8 | | 36.5 | 38.1 | 27.8 | 27.4 | 30.0 | 38.3 |
| **Seg. – Seg. class sky** | | | | | | | | | | | | | |
| PostNet | 61.2 | **50.8** | **53.3** | 32.7 | **45.3** | **49.2** | | **79.9** | **61.6** | **62.7** | 32.6 | **46.0** | **66.7** |
| PriorNet | 31.1 | 30.8 | 30.8 | 30.8 | 30.8 | 30.8 | | 31.4 | 30.8 | 30.8 | 30.8 | 30.8 | 30.8 |
| DDNet | 30.8 | 30.8 | 30.8 | 30.8 | 30.8 | 30.8 | | 30.8 | 30.8 | 30.8 | 30.8 | 30.8 | 30.8 |
| EvNet | **67.0** | 34.9 | 30.9 | 30.8 | 30.8 | 31.6 | | 75.5 | 52.1 | 31.2 | 31.2 | 30.8 | 30.8 |

Table 44: Adversarial training with CE: OOD detection based on differential entropy under FGSM uncertainty attacks against differential entropy on ID data and OOD data (AUC-PR).

| | ID-Attack (non-attacked OOD) | | | | | | | OOD-Attack (non-attacked ID) | | | | | |
|---|---|---|---|---|---|---|---|---|---|---|---|---|---|
| Att. Rad. | 0.1 | 0.2 | 0.5 | 1.0 | 2.0 | 4.0 | | 0.1 | 0.2 | 0.5 | 1.0 | 2.0 | 4.0 |
| **MNIST – KMNIST** | | | | | | | | | | | | | |
| PostNet | 95.8 | 93.7 | 91.7 | 90.2 | 80.3 | 73.5 | | 95.0 | 91.6 | 84.9 | 80.8 | 73.9 | 81.4 |
| PriorNet | **99.5** | **99.2** | **98.3** | **95.7** | **87.2** | **77.0** | | **99.5** | **99.2** | **98.8** | 96.8 | 92.4 | 76.2 |
| DDNet | 99.0 | 99.0 | 98.1 | 94.6 | 80.3 | 47.3 | | 99.1 | 99.1 | 98.6 | **97.3** | **95.3** | **92.4** |
| EvNet | 71.4 | 67.7 | 62.6 | 68.6 | 56.9 | 53.0 | | 68.3 | 57.1 | 50.7 | 62.6 | 50.7 | 46.4 |
| **CIFAR10 – SVHN** | | | | | | | | | | | | | |
| PostNet | 67.4 | 61.4 | **62.9** | **70.7** | 65.0 | 44.4 | | 65.6 | 54.6 | 39.0 | **45.1** | **62.7** | **77.6** |
| PriorNet | 17.0 | 16.9 | 17.1 | 18.1 | 24.0 | 37.5 | | 17.2 | 17.3 | 17.4 | 18.9 | 22.2 | 29.8 |
| DDNet | **71.0** | **66.9** | 56.1 | 55.7 | 48.7 | 44.7 | | **72.2** | **66.4** | **48.0** | 42.2 | 48.7 | 69.1 |
| EvNet | 56.5 | 62.2 | 51.6 | 53.9 | **64.4** | **46.6** | | 55.9 | 42.1 | 35.0 | 37.4 | 56.2 | 68.7 |
| **Sens. – Sens. class 10, 11** | | | | | | | | | | | | | |
| PostNet | 41.2 | 37.9 | 34.3 | 41.4 | 45.7 | 45.7 | | **99.2** | **99.8** | **99.8** | **100.0** | **99.9** | **100.0** |
| PriorNet | 27.2 | 28.0 | 29.3 | 37.5 | **96.5** | 77.7 | | 28.8 | 29.5 | 26.6 | 26.5 | 26.5 | 26.5 |
| DDNet | 30.7 | 32.8 | **65.6** | **72.7** | 92.9 | **94.4** | | 27.5 | 29.5 | 28.7 | 26.6 | 26.5 | 27.2 |
| EvNet | **44.3** | **47.2** | 47.7 | 46.3 | 38.8 | 40.0 | | 51.6 | 69.3 | 50.4 | 48.5 | 65.4 | 72.3 |
| **Seg. – Seg. class sky** | | | | | | | | | | | | | |
| PostNet | 61.9 | **54.3** | **57.4** | 34.0 | 37.5 | 43.3 | | 92.9 | 92.2 | 91.1 | **99.9** | **99.9** | **100.0** |
| PriorNet | 31.0 | 30.8 | 30.8 | 30.8 | 30.8 | 36.9 | | 31.2 | 30.8 | 30.8 | 30.8 | 30.8 | 30.8 |
| DDNet | 30.8 | 30.8 | 33.0 | 37.8 | 59.4 | **92.1** | | 30.8 | 30.8 | 30.8 | 30.8 | 30.8 | 30.8 |
| EvNet | **66.7** | 42.3 | 45.6 | **49.0** | **61.5** | 50.1 | | 74.6 | 57.3 | 51.2 | 45.8 | 60.0 | 63.2 |

Table 45: Adversarial training with CE: OOD detection based on differential entropy under Noise uncertainty attacks against differential entropy on ID data and OOD data (AUC-PR).

| | ID-Attack (non-attacked OOD) | | | | | | | OOD-Attack (non-attacked ID) | | | | | |
|---|---|---|---|---|---|---|---|---|---|---|---|---|---|
| Att. Rad. | 0.1 | 0.2 | 0.5 | 1.0 | 2.0 | 4.0 | | 0.1 | 0.2 | 0.5 | 1.0 | 2.0 | 4.0 |
| | MNIST – KMNIST | | | | | | | | | | | | |
| PostNet | 91.7 | 72.7 | 32.4 | **36.2** | 33.2 | 30.7 | | 93.0 | 85.5 | 89.9 | 90.5 | 73.1 | 99.1 |
| PriorNet | **99.7** | **98.4** | **71.5** | 34.3 | **37.3** | **38.7** | | **99.2** | 96.8 | 96.8 | 99.3 | **100.0** | 77.4 |
| DDNet | 96.6 | 79.5 | 31.9 | 31.2 | 30.9 | 35.0 | | 98.6 | **99.5** | **99.9** | **100.0** | 99.8 | 98.5 |
| EvNet | 87.4 | 49.9 | 32.8 | 32.0 | 33.2 | 36.3 | | 87.4 | 46.8 | 48.6 | 45.1 | 33.3 | **100.0** |
| | CIFAR10 – SVHN | | | | | | | | | | | | |
| PostNet | 43.9 | 31.9 | 30.7 | 30.7 | **56.2** | 31.5 | | 85.1 | **84.8** | 85.2 | 97.0 | 82.2 | 99.5 |
| PriorNet | 31.4 | 32.8 | **85.8** | **37.1** | 30.7 | 30.8 | | 35.2 | 40.1 | 30.8 | 42.1 | **99.0** | **100.0** |
| DDNet | 50.8 | **42.8** | 30.7 | 32.8 | 30.7 | **94.9** | | 82.3 | 80.3 | **99.2** | 97.8 | 98.9 | 63.9 |
| EvNet | **56.2** | 34.4 | 32.1 | 32.4 | 37.9 | 46.7 | | 59.8 | 63.2 | 82.5 | 92.0 | 41.6 | 51.2 |
| | Sens. – Sens. class 10, 11 | | | | | | | | | | | | |
| PostNet | 30.8 | 47.8 | **50.0** | 50.0 | 50.0 | 50.0 | | 98.7 | **99.8** | **100.0** | **100.0** | 99.8 | **100.0** |
| PriorNet | 30.7 | 30.7 | 30.9 | **87.7** | **100.0** | **100.0** | | 31.0 | 30.7 | 32.4 | 31.4 | 34.2 | 30.7 |
| DDNet | 30.7 | 30.7 | 47.4 | 75.0 | 92.9 | 79.7 | | 30.7 | 30.7 | 30.7 | 41.3 | 34.3 | 39.6 |
| EvNet | **30.8** | 30.8 | 40.8 | 36.3 | 50.7 | 34.2 | | 34.5 | 31.0 | 34.4 | 38.8 | 47.4 | 51.1 |
| | Seg. – Seg. class sky | | | | | | | | | | | | |
| PostNet | 34.2 | **31.0** | 42.6 | **49.9** | 50.0 | 50.0 | | 97.7 | 93.7 | **99.8** | **100.0** | **100.0** | **100.0** |
| PriorNet | 30.8 | 30.8 | 30.8 | 30.8 | 30.9 | **100.0** | | 30.9 | 30.8 | 30.8 | 30.9 | 31.2 | 32.9 |
| DDNet | 30.8 | 30.8 | 30.8 | 31.3 | **77.8** | 93.3 | | 30.8 | 30.8 | 30.8 | 30.8 | 30.8 | 32.3 |
| EvNet | **63.3** | 30.8 | 30.8 | 30.8 | 32.7 | 36.0 | | 98.8 | 40.2 | 40.4 | 34.8 | 50.0 | 32.5 |

Table 46: Adversarial training with Diff. Ent.: Accuracy based on differential entropy under PGD label attacks (AUC-PR).

| Att. Rad. | 0.1 | 0.2 | 0.5 | 1.0 | 2.0 | 4.0 | | 0.1 | 0.2 | 0.5 | 1.0 | 2.0 | 4.0 |
|---|---|---|---|---|---|---|---|---|---|---|---|---|---|
| | MNIST | | | | | | | CIFAR10 | | | | | |
| PostNet | **99.1** | 98.8 | 96.8 | 88.8 | 65.9 | 15.6 | | 72.9 | 50.7 | 12.9 | 3.1 | 0.4 | 0.4 |
| PriorNet | 99.1 | 98.8 | **97.7** | **94.7** | **88.8** | **73.4** | | 66.5 | 62.8 | **52.9** | **35.8** | **23.0** | **9.6** |
| DDNet | 99.1 | **98.9** | 97.4 | 91.9 | 48.7 | 0.3 | | **78.9** | **63.1** | 22.0 | 1.9 | 0.0 | 0.0 |
| EvNet | 98.3 | 98.1 | 95.2 | 91.0 | 72.7 | 40.1 | | 65.6 | 48.9 | 14.8 | 8.4 | 3.8 | 1.8 |
| | Sensorless | | | | | | | Segment | | | | | |
| PostNet | 16.1 | 7.4 | 5.8 | **7.5** | **9.4** | **12.5** | | 84.7 | 47.1 | **22.3** | 6.4 | 10.8 | **3.8** |
| PriorNet | 33.3 | 15.6 | 3.7 | 0.0 | 0.0 | 0.0 | | **93.9** | **65.9** | 18.1 | 2.9 | 0.0 | 0.0 |
| DDNet | 12.9 | 3.0 | 0.5 | 0.3 | 0.2 | 0.2 | | 90.6 | 47.5 | 8.4 | 0.1 | 0.0 | 0.0 |
| EvNet | **36.1** | **22.1** | **10.8** | 3.8 | 1.7 | 3.1 | | 92.0 | 56.2 | 11.9 | 2.1 | 0.4 | 2.8 |

Table 47: Adversarial training with Diff. Ent.: Accuracy based on differential entropy under FGSM label attacks (AUC-PR).

| Att. Rad. | 0.1 | 0.2 | 0.5 | 1.0 | 2.0 | 4.0 | | 0.1 | 0.2 | 0.5 | 1.0 | 2.0 | 4.0 |
|---|---|---|---|---|---|---|---|---|---|---|---|---|---|
| | MNIST | | | | | | | CIFAR10 | | | | | |
| PostNet | **99.1** | 98.9 | 97.6 | 95.0 | 91.2 | 80.9 | | 23.5 | 11.2 | 0.2 | 0.1 | 9.0 | 1.6 |
| PriorNet | **99.1** | 98.8 | **97.9** | **95.9** | **93.5** | **87.4** | | 31.6 | **14.4** | 6.2 | 7.8 | 0.2 | 1.4 |
| DDNet | **99.1** | **99.0** | **97.9** | 94.9 | 78.4 | 23.3 | | **36.8** | 13.9 | **9.2** | **10.3** | **10.0** | **10.0** |
| EvNet | 98.3 | 98.3 | 95.7 | 95.4 | 88.9 | 63.7 | | 24.4 | 4.8 | 0.5 | 9.0 | 11.2 | 3.9 |
| | Sensorless | | | | | | | Segment | | | | | |
| PostNet | 21.4 | 10.6 | 10.3 | 12.3 | 12.4 | 12.5 | | 83.9 | 53.7 | **27.8** | **15.4** | **20.1** | **19.1** |
| PriorNet | **40.2** | 21.7 | 8.1 | 0.0 | 0.0 | 2.3 | | 91.3 | 57.2 | 18.1 | 3.2 | 0.0 | 0.0 |
| DDNet | 17.6 | 5.0 | 4.9 | 8.0 | 7.0 | 5.7 | | 86.1 | 39.2 | 12.1 | 0.4 | 0.0 | 2.9 |
| EvNet | 43.1 | **29.3** | **21.3** | **14.4** | **13.4** | **13.5** | | **91.9** | **65.3** | 17.9 | 2.9 | 0.1 | 0.8 |

Table 48: Adversarial training with Diff. Ent.: Accuracy based on differential entropy under Noise label attacks (AUC-PR).

| Att. Rad. | 0.1 | 0.2 | 0.5 | 1.0 | 2.0 | 4.0 | | 0.1 | 0.2 | 0.5 | 1.0 | 2.0 | 4.0 |
|---|---|---|---|---|---|---|---|---|---|---|---|---|---|
| | MNIST | | | | | | | CIFAR10 | | | | | |
| PostNet | 97.0 | 88.5 | 10.0 | 2.2 | 0.0 | **0.4** | | 71.8 | 54.2 | 30.7 | 21.6 | 16.8 | 14.6 |
| PriorNet | **98.1** | 88.5 | **24.5** | **4.1** | 0.0 | 0.0 | | 67.2 | **65.2** | **58.9** | **48.5** | **40.5** | **31.4** |
| DDNet | 97.8 | **92.5** | 6.4 | 2.2 | **0.1** | 0.1 | | **78.1** | 63.8 | 34.2 | 16.0 | 8.0 | 6.0 |
| EvNet | 96.2 | 87.0 | 2.3 | 0.1 | 0.0 | 0.0 | | 65.2 | 54.8 | 29.2 | 18.7 | 17.4 | 16.2 |
| | Sensorless | | | | | | | Segment | | | | | |
| PostNet | 1.0 | **4.7** | **11.4** | **11.7** | **11.7** | **11.7** | | 27.6 | 1.5 | **3.6** | **15.2** | **20.9** | **21.2** |
| PriorNet | **4.7** | 0.0 | 0.0 | 0.1 | 0.0 | 2.2 | | **56.4** | **16.1** | 2.1 | 0.9 | 0.0 | 0.0 |
| DDNet | 0.3 | 0.0 | 0.0 | 0.0 | 0.0 | 0.1 | | 49.4 | 6.1 | 0.0 | 0.0 | 0.0 | 0.0 |
| EvNet | 0.9 | 0.0 | 0.9 | 0.2 | 3.5 | 3.1 | | 51.2 | 10.6 | 0.3 | 0.0 | 0.0 | 0.6 |

Table 49: Adversarial training with Diff. Ent.: Certainty based on differential entropy under PGD label attacks (AUC-PR).

| Att. Rad. | 0.1 | 0.2 | 0.5 | 1.0 | 2.0 | 4.0 | | 0.1 | 0.2 | 0.5 | 1.0 | 2.0 | 4.0 |
|---|---|---|---|---|---|---|---|---|---|---|---|---|---|
| | | | MNIST | | | | | | | CIFAR10 | | | |
| PostNet | 99.9 | 99.8 | 98.5 | 86.8 | 53.0 | 10.2 | | 88.5 | 56.6 | 7.5 | 1.7 | 0.3 | 0.2 |
| PriorNet | 99.7 | 99.6 | 98.0 | 91.4 | **76.2** | **54.8** | | 51.0 | 44.7 | **36.3** | **23.7** | **13.8** | **5.5** |
| DDNet | **100.0** | **99.9** | **99.7** | **97.6** | 47.9 | 0.1 | | **94.7** | **82.4** | 21.2 | 1.1 | 0.0 | 0.0 |
| EvNet | 99.2 | 98.9 | 96.8 | 86.5 | 60.8 | 33.73 | | 80.6 | 50.4 | 14.1 | 9.1 | 9.7 | 2.2 |
| | | | Sensorless | | | | | | | Segment | | | |
| PostNet | 10.6 | 5.3 | 5.8 | **7.5** | **9.5** | **12.5** | | 76.1 | 30.1 | **13.4** | **4.9** | **13.2** | **3.8** |
| PriorNet | **22.6** | 10.3 | 3.8 | 0.0 | 0.0 | 0.0 | | **97.9** | **63.8** | 11.4 | 1.7 | 0.0 | 0.0 |
| DDNet | 13.2 | 2.2 | 0.4 | 0.3 | 0.1 | 0.2 | | 95.8 | 51.1 | 5.0 | 0.3 | 0.0 | 0.0 |
| EvNet | **22.6** | **12.8** | **5.9** | 2.0 | 1.1 | 2.9 | | 94.5 | 44.3 | 7.0 | 1.2 | 0.4 | 2.0 |

Table 50: Adversarial training with Diff. Ent.: Certainty based on differential entropy under FGSM label attacks (AUC-PR).

| Att. Rad. | 0.1 | 0.2 | 0.5 | 1.0 | 2.0 | 4.0 | | 0.1 | 0.2 | 0.5 | 1.0 | 2.0 | 4.0 |
|---|---|---|---|---|---|---|---|---|---|---|---|---|---|
| | | | MNIST | | | | | | | CIFAR10 | | | |
| PostNet | 99.9 | **99.9** | 99.4 | 97.1 | **92.3** | **82.2** | | 88.3 | 67.8 | 30.9 | 19.2 | 15.9 | 16.7 |
| PriorNet | 99.7 | 99.6 | 98.4 | 93.8 | 87.0 | 79.5 | | 52.9 | 49.1 | **44.1** | **36.3** | **28.6** | **20.9** |
| DDNet | **100.0** | **99.9** | **99.8** | **98.8** | 85.8 | 21.0 | | **94.1** | **82.8** | 43.1 | 15.4 | 6.8 | 5.1 |
| EvNet | 99.2 | 99.1 | 97.8 | 95.3 | 87.8 | 67.0 | | 80.2 | 61.1 | 32.5 | 21.9 | 21.6 | 22.2 |
| | | | Sensorless | | | | | | | Segment | | | |
| PostNet | 13.8 | 7.2 | 9.7 | 12.3 | 12.4 | 12.5 | | 74.4 | 37.9 | 20.2 | **16.7** | **19.5** | **19.9** |
| PriorNet | 32.3 | 19.1 | 9.1 | 0.0 | 0.0 | 5.2 | | **96.2** | 56.6 | 11.4 | 10.7 | 0.0 | 0.0 |
| DDNet | 19.7 | 7.0 | 3.7 | 7.6 | 7.6 | 7.1 | | 92.2 | 40.5 | 7.0 | 0.4 | 0.0 | 4.9 |
| EvNet | **37.8** | **30.5** | **26.0** | **14.7** | **13.9** | **13.6** | | 91.9 | **57.7** | 11.8 | 2.3 | 0.7 | 1.1 |

Table 51: Adversarial training with Diff. Ent.: Certainty based on differential entropy under Noise label attacks (AUC-PR).

| Att. Rad. | 0.1 | 0.2 | 0.5 | 1.0 | 2.0 | 4.0 | | 0.1 | 0.2 | 0.5 | 1.0 | 2.0 | 4.0 |
|---|---|---|---|---|---|---|---|---|---|---|---|---|---|
| | | | MNIST | | | | | | | CIFAR10 | | | |
| PostNet | 99.7 | 99.2 | 64.5 | 8.7 | 0.0 | **0.5** | | 77.4 | 14.6 | 0.3 | 0.1 | 9.1 | 3.7 |
| PriorNet | **99.9** | 98.8 | **84.5** | **17.5** | 0.0 | 0.0 | | 44.7 | 11.3 | 13.6 | 9.0 | 0.4 | 2.6 |
| DDNet | 99.7 | **99.4** | 39.5 | 5.9 | **0.6** | 0.3 | | **86.1** | **46.5** | **15.5** | **17.4** | **10.2** | **10.1** |
| EvNet | 99.2 | 97.0 | 19.4 | 0.1 | 0.0 | 0.0 | | 67.1 | 12.7 | 4.5 | 12.8 | 13.2 | 3.3 |
| | | | Sensorless | | | | | | | Segment | | | |
| PostNet | 0.6 | **5.1** | **12.2** | **11.7** | **11.7** | **11.7** | | 36.7 | 2.0 | 3.6 | **17.2** | **20.8** | **21.3** |
| PriorNet | **8.5** | 0.0 | 0.0 | 0.2 | 0.0 | 2.0 | | **90.5** | **32.8** | **7.1** | 1.2 | 0.0 | 0.0 |
| DDNet | 1.5 | 0.0 | 0.0 | 0.0 | 0.0 | 0.0 | | 79.6 | 21.8 | 0.0 | 0.0 | 0.0 | 0.0 |
| EvNet | 1.5 | 0.0 | 1.0 | 0.2 | 4.9 | 4.8 | | 75.7 | 22.0 | 3.2 | 0.0 | 0.0 | 0.7 |

Table 52: Adversarial training with Diff. Ent.: Attack-Detection based on differential entropy under PGD label attacks (AUC-PR).

| Att. Rad. | 0.1 | 0.2 | 0.5 | 1.0 | 2.0 | 4.0 | | 0.1 | 0.2 | 0.5 | 1.0 | 2.0 | 4.0 |
|---|---|---|---|---|---|---|---|---|---|---|---|---|---|
| | | | MNIST | | | | | | | CIFAR10 | | | |
| PostNet | 57.9 | 65.4 | 87.1 | 93.6 | 79.3 | 47.7 | | **63.1** | **67.1** | 41.7 | 34.4 | 35.0 | 36.5 |
| PriorNet | **66.9** | **76.0** | **95.1** | **96.4** | 88.7 | 74.8 | | 55.7 | 55.8 | 53.1 | **48.9** | **43.3** | 37.8 |
| DDNet | 53.7 | 58.5 | 69.3 | 85.5 | **96.1** | **87.7** | | 56.7 | 62.5 | **60.4** | 41.2 | 32.6 | 31.8 |
| EvNet | 54.3 | 58.9 | 63.2 | 72.3 | 69.4 | 59.1 | | 55.9 | 60.2 | 49.7 | 44.6 | 41.4 | **39.4** |
| | | | Sensorless | | | | | | | Segment | | | |
| PostNet | 49.8 | 41.5 | 36.3 | **51.0** | **85.9** | **99.0** | | 95.0 | 77.6 | 48.9 | **42.9** | **45.2** | 68.5 |
| PriorNet | 50.4 | 39.4 | 31.6 | 30.8 | 30.7 | 30.7 | | 86.1 | **89.7** | **50.8** | 37.5 | 32.9 | 30.8 |
| DDNet | **52.2** | 41.5 | 35.5 | 32.3 | 31.5 | 35.9 | | 77.9 | 87.3 | 43.4 | 32.4 | 30.9 | 30.8 |
| EvNet | 48.0 | **44.3** | **38.8** | 35.2 | 39.1 | 48.9 | | **95.4** | 77.9 | 42.6 | 33.7 | 31.3 | 31.6 |

Table 53: Adversarial training with Diff. Ent.: Attack-Detection based on differential entropy under FGSM label attacks (AUC-PR).

| Att. Rad. | 0.1 | 0.2 | 0.5 | 1.0 | 2.0 | 4.0 | | 0.1 | 0.2 | 0.5 | 1.0 | 2.0 | 4.0 |
|---|---|---|---|---|---|---|---|---|---|---|---|---|---|
| | | | MNIST | | | | | | | CIFAR10 | | | |
| PostNet | 56.2 | 61.6 | 78.0 | 89.5 | 92.1 | 90.7 | | **63.8** | **68.8** | 65.8 | 62.9 | 67.0 | 73.6 |
| PriorNet | **67.3** | **76.9** | **95.3** | **98.0** | **98.0** | 98.0 | | 58.0 | 62.5 | **67.3** | **67.3** | 65.9 | 62.2 |
| DDNet | 53.8 | 58.6 | 68.7 | 83.9 | 95.8 | **98.9** | | 57.7 | 63.8 | 71.0 | 72.8 | 74.8 | 79.3 |
| EvNet | 53.7 | 57.2 | 59.8 | 65.2 | 71.5 | 72.4 | | 56.2 | 62.4 | 58.9 | 59.0 | 63.2 | 70.5 |
| | | | Sensorless | | | | | | | Segment | | | |
| PostNet | **98.3** | **99.8** | **99.9** | **99.9** | **99.9** | **100.0** | | **96.0** | 93.3 | 93.1 | 97.4 | **99.8** | **99.6** |
| PriorNet | 67.0 | 56.0 | 35.8 | 30.8 | 30.7 | 30.7 | | 88.6 | **89.3** | 55.1 | 45.2 | 37.6 | 30.8 |
| DDNet | 60.1 | 50.0 | 40.3 | 32.5 | 31.5 | 33.8 | | 81.6 | 89.2 | 56.9 | 49.1 | 35.8 | 31.8 |
| EvNet | 68.7 | 69.8 | 69.9 | 68.6 | 68.8 | 73.2 | | 95.9 | 86.6 | 66.2 | 63.2 | 74.2 | 77.2 |

Table 54: Adversarial training with Diff. Ent.: Attack-Detection based on differential entropy under Noise label attacks (AUC-PR).

| Att. Rad. | 0.1 | 0.2 | 0.5 | 1.0 | 2.0 | 4.0 | | 0.1 | 0.2 | 0.5 | 1.0 | 2.0 | 4.0 |
|---|---|---|---|---|---|---|---|---|---|---|---|---|---|
| | | | MNIST | | | | | | | CIFAR10 | | | |
| PostNet | 53.7 | 67.7 | 94.5 | 94.1 | 98.1 | **100.0** | | **85.8** | 79.1 | 93.0 | **98.5** | **96.3** | 96.6 |
| PriorNet | 33.1 | 42.4 | 80.8 | 93.8 | 78.8 | 56.0 | | 52.3 | 38.2 | 51.8 | 38.1 | 87.4 | **99.9** |
| DDNet | **56.0** | **86.7** | **99.9** | **100.0** | **100.0** | 99.4 | | 79.5 | **90.9** | **99.6** | 98.2 | 96.2 | 85.0 |
| EvNet | 50.8 | 71.7 | 83.1 | 88.2 | 68.4 | 88.0 | | 67.9 | 87.0 | 93.3 | 89.6 | 93.7 | 97.3 |
| | | | Sensorless | | | | | | | Segment | | | |
| PostNet | **99.9** | **100.0** | **100.0** | **100.0** | **99.9** | **100.0** | | 93.2 | **99.3** | **99.9** | **100.0** | **100.0** | **100.0** |
| PriorNet | 56.7 | 42.9 | 32.0 | 30.8 | 31.0 | 30.7 | | 55.9 | 63.1 | 41.7 | 33.3 | 30.8 | 30.8 |
| DDNet | 51.2 | 43.9 | 33.8 | 32.4 | 31.7 | 35.0 | | 55.9 | 58.9 | 42.1 | 33.4 | 31.1 | 32.7 |
| EvNet | 69.2 | 58.6 | 71.7 | 52.3 | 70.9 | 77.6 | | 63.0 | 62.7 | 63.9 | 54.4 | 74.6 | 61.2 |

Table 55: Adversarial training with Diff. Ent.: OOD detection based on differential entropy under PGD uncertainty attacks against differential entropy on ID data and OOD data (AUC-PR).

| | ID-Attack (non-attacked OOD) | | | | | | | OOD-Attack (non-attacked ID) | | | | | |
|---|---|---|---|---|---|---|---|---|---|---|---|---|---|
| Att. Rad. | 0.1 | 0.2 | 0.5 | 1.0 | 2.0 | 4.0 | | 0.1 | 0.2 | 0.5 | 1.0 | 2.0 | 4.0 |
| | | | | | | **MNIST – KMNIST** | | | | | | | |
| PostNet | 92.6 | 94.0 | 89.5 | 69.1 | 49.6 | 31.6 | | 91.9 | 91.4 | 79.8 | 54.2 | 42.4 | 46.0 |
| PriorNet | **99.6** | **99.3** | **97.6** | **93.7** | **81.4** | **50.8** | | **99.6** | **99.2** | 97.4 | 92.1 | 66.3 | 37.6 |
| DDNet | 99.1 | 98.9 | **97.6** | 93.6 | 60.7 | 30.7 | | 99.1 | 99.0 | **98.3** | **97.0** | **91.4** | **77.5** |
| EvNet | 73.4 | 66.7 | 72.9 | 57.7 | 49.3 | 45.0 | | 63.7 | 51.3 | 58.8 | 35.1 | 33.4 | 36.4 |
| | | | | | | **CIFAR10 – SVHN** | | | | | | | |
| PostNet | 68.6 | 46.1 | 21.7 | 17.5 | 16.2 | 15.6 | | 63.3 | 37.4 | 19.0 | 17.4 | 16.7 | 16.8 |
| PriorNet | 17.3 | 15.9 | 17.4 | 15.5 | 15.4 | 15.4 | | 17.1 | 15.7 | 16.6 | 15.4 | 15.4 | 15.4 |
| DDNet | **77.5** | **66.0** | **34.5** | 16.4 | 15.4 | 15.4 | | **79.6** | **67.4** | **33.2** | 16.9 | 15.4 | 15.4 |
| EvNet | 57.8 | 35.2 | 22.0 | **21.5** | **17.5** | **16.0** | | 52.7 | 30.7 | 30.3 | **31.1** | **20.8** | **18.0** |
| | | | | | | **Sens. – Sens. class 10, 11** | | | | | | | |
| PostNet | **39.6** | **34.8** | **41.8** | **46.0** | **44.9** | **47.6** | | **41.1** | **40.6** | 30.8 | 35.6 | 83.0 | 99.5 |
| PriorNet | 26.6 | 26.5 | 26.5 | 26.5 | 26.5 | 26.6 | | 28.8 | 27.0 | 26.6 | 26.9 | 26.5 | 26.5 |
| DDNet | 26.8 | 26.5 | 26.5 | 26.6 | 26.6 | 28.0 | | 26.8 | 26.6 | 26.7 | 26.7 | 26.6 | 26.6 |
| EvNet | 31.0 | 29.4 | 27.2 | 29.1 | 32.4 | 36.5 | | 39.1 | 35.1 | 28.9 | 28.7 | 30.0 | 38.3 |
| | | | | | | **Seg. – Seg. class sky** | | | | | | | |
| PostNet | **91.7** | **45.3** | **44.6** | **38.8** | **46.0** | **49.4** | | **98.7** | 67.3 | **44.1** | **47.7** | **37.5** | **59.4** |
| PriorNet | 31.2 | 30.8 | 30.8 | 30.8 | 30.8 | 30.8 | | 31.7 | 30.8 | 30.8 | 30.8 | 30.8 | 30.8 |
| DDNet | 31.0 | 30.8 | 30.8 | 30.8 | 30.8 | 30.8 | | 31.2 | 30.8 | 30.8 | 30.8 | 30.8 | 30.8 |
| EvNet | 58.0 | 39.4 | 31.0 | 30.8 | 30.8 | 31.5 | | 84.1 | **71.7** | 36.1 | 31.1 | 30.8 | 30.8 |

Table 56: Adversarial training with Diff. Ent.: OOD detection based on differential entropy under FGSM uncertainty attacks against differential entropy on ID data and OOD data (AUC-PR).

| | ID-Attack (non-attacked OOD) | | | | | | | OOD-Attack (non-attacked ID) | | | | | |
|---|---|---|---|---|---|---|---|---|---|---|---|---|---|
| Att. Rad. | 0.1 | 0.2 | 0.5 | 1.0 | 2.0 | 4.0 | | 0.1 | 0.2 | 0.5 | 1.0 | 2.0 | 4.0 |
| | | | | | | **MNIST – KMNIST** | | | | | | | |
| PostNet | 92.8 | 94.2 | 93.1 | 84.7 | 81.2 | **72.9** | | 92.1 | 92.2 | 88.7 | 79.0 | 78.2 | 75.9 |
| PriorNet | **99.6** | **99.3** | 97.7 | 94.5 | **86.6** | 71.0 | | **99.6** | **99.3** | 98.1 | 96.5 | 90.2 | 71.5 |
| DDNet | 99.1 | 98.9 | **97.9** | **95.5** | 79.2 | 46.3 | | 99.1 | 99.0 | **98.5** | **97.8** | **95.4** | **92.1** |
| EvNet | 73.7 | 67.6 | 74.4 | 62.2 | 56.5 | 59.5 | | 67.7 | 57.6 | 67.5 | 49.8 | 49.0 | 55.0 |
| | | | | | | **CIFAR10 – SVHN** | | | | | | | |
| PostNet | 71.3 | 60.8 | **67.1** | **68.7** | **61.8** | **54.4** | | 65.7 | 49.9 | 41.7 | 37.8 | **61.0** | **78.4** |
| PriorNet | 17.5 | 16.3 | 20.8 | 17.4 | 19.8 | 19.6 | | 17.5 | 16.4 | 21.7 | 17.2 | 21.0 | 20.1 |
| DDNet | **77.7** | **69.9** | 61.4 | 53.8 | 45.7 | 37.2 | | **79.3** | **69.3** | **54.2** | 47.5 | 52.4 | 73.9 |
| EvNet | 62.9 | 51.2 | 48.9 | 58.7 | 58.1 | 46.3 | | 53.9 | 35.0 | 37.9 | **51.4** | 56.4 | 59.6 |
| | | | | | | **Sens. – Sens. class 10, 11** | | | | | | | |
| PostNet | 40.1 | 35.5 | 33.8 | 43.0 | 43.6 | 47.5 | | **99.7** | **99.9** | **99.9** | **99.9** | **99.9** | **100.0** |
| PriorNet | 28.4 | 29.5 | 39.4 | 53.3 | 93.0 | **98.3** | | 30.2 | 28.3 | 27.2 | 27.8 | 26.9 | 26.5 |
| DDNet | 32.2 | 34.4 | **53.5** | **84.2** | **93.8** | 92.2 | | 26.8 | 26.6 | 26.7 | 26.7 | 26.6 | 26.6 |
| EvNet | **43.1** | **41.0** | 37.0 | 43.5 | 42.9 | 41.9 | | 56.8 | 56.4 | 57.8 | 60.0 | 66.0 | 67.8 |
| | | | | | | **Seg. – Seg. class sky** | | | | | | | |
| PostNet | **91.4** | **54.2** | **53.1** | 45.4 | 38.1 | 40.6 | | **98.8** | 75.9 | 87.7 | **99.1** | **99.9** | **100.0** |
| PriorNet | 31.1 | 30.8 | 30.8 | 30.8 | 30.8 | 39.2 | | 31.5 | 30.8 | 30.8 | 30.8 | 30.8 | 30.8 |
| DDNet | 30.9 | 30.8 | 30.8 | 34.9 | **58.2** | **85.4** | | 31.1 | 30.8 | 30.8 | 30.8 | 30.8 | 30.8 |
| EvNet | 55.3 | 46.6 | 51.8 | **51.0** | 48.9 | 41.4 | | 86.6 | 71.6 | 49.5 | 57.4 | 76.6 | 58.6 |

Table 57: Adversarial training with Diff. Ent.: OOD detection based on differential entropy under Noise uncertainty attacks against differential entropy on ID data and OOD data (AUC-PR).

| | ID-Attack (non-attacked OOD) | | | | | | OOD-Attack (non-attacked ID) | | | | | |
|---|---|---|---|---|---|---|---|---|---|---|---|---|
| Att. Rad. | 0.1 | 0.2 | 0.5 | 1.0 | 2.0 | 4.0 | 0.1 | 0.2 | 0.5 | 1.0 | 2.0 | 4.0 |
| | | | | | | **MNIST – KMNIST** | | | | | | |
| PostNet | 90.8 | 81.0 | 39.1 | 38.3 | 30.8 | 30.7 | 88.1 | 88.9 | 81.0 | 70.8 | 95.4 | **100.0** |
| PriorNet | 99.8 | **93.6** | **41.0** | 42.1 | **40.0** | **51.7** | **100.0** | 99.3 | 98.1 | 94.0 | 44.8 | 32.1 |
| DDNet | **97.2** | 87.1 | 31.0 | 31.1 | 31.6 | 40.6 | 98.8 | **99.5** | **99.9** | **99.9** | **100.0** | 98.1 |
| EvNet | 68.6 | 52.2 | 31.9 | 33.1 | 32.9 | 31.3 | 53.2 | 62.2 | 39.7 | 45.9 | 33.7 | 50.3 |
| | | | | | | **CIFAR10 – SVHN** | | | | | | |
| PostNet | 46.7 | **67.3** | **33.9** | 30.7 | **39.4** | 44.0 | 88.0 | 78.2 | 84.8 | 92.1 | 93.6 | 95.9 |
| PriorNet | 41.9 | 37.0 | **33.9** | 42.5 | 32.6 | 30.7 | 35.4 | 32.4 | 38.2 | 33.3 | 76.1 | **99.5** |
| DDNet | **58.7** | 37.3 | 30.7 | 33.5 | 33.6 | **61.0** | 90.6 | 88.3 | 99.0 | 97.5 | 93.9 | 80.9 |
| EvNet | 48.9 | 34.8 | 30.9 | 31.8 | 37.7 | 34.4 | 69.6 | 77.7 | 85.3 | 87.4 | 93.1 | 95.7 |
| | | | | | | **Sens. – Sens. class 10, 11** | | | | | | |
| PostNet | **30.8** | **47.8** | **50.0** | 50.0 | 50.0 | 50.0 | 98.7 | 99.8 | 100.0 | 100.0 | 99.8 | 100.0 |
| PriorNet | 30.7 | 30.7 | 30.9 | **87.7** | **100.0** | **100.0** | 31.0 | 30.7 | 32.4 | 31.4 | 34.2 | 30.7 |
| DDNet | 30.7 | 30.7 | 47.4 | 75.0 | 92.9 | 79.7 | 30.7 | 30.7 | 30.7 | 41.3 | 34.3 | 39.6 |
| EvNet | **30.8** | 30.8 | 40.8 | 36.3 | 50.7 | 34.2 | 34.5 | 31.0 | 34.4 | 38.8 | 47.4 | 51.1 |
| | | | | | | **Seg. – Seg. class sky** | | | | | | |
| PostNet | 34.2 | **31.0** | **42.6** | **49.9** | 50.0 | 50.0 | 97.7 | **93.7** | **99.8** | **100.0** | **100.0** | **100.0** |
| PriorNet | 30.8 | 30.8 | 30.8 | 30.8 | 30.9 | **100.0** | 30.9 | 30.8 | 30.8 | 30.9 | 31.2 | 32.9 |
| DDNet | 30.8 | 30.8 | 30.8 | 31.3 | **77.8** | 93.3 | 30.8 | 30.8 | 30.8 | 30.8 | 30.8 | 32.3 |
| EvNet | **63.3** | 30.8 | 30.8 | 30.8 | 32.7 | 36.0 | **98.8** | 40.2 | 40.4 | 34.8 | 50.0 | 32.5 |

## A.6 VISUALIZATION OF DIFFERENTIAL ENTROPY DISTRIBUTIONS ON ID DATA AND OOD DATA

The following Figures visualize the differential entropy distribution for ID data and OOD data for all models with standard training. We used label attacks and uncertainty attacks for CIFAR10 and MNIST. Thus, they show how well the DBU models separate on clean and perturbed ID data and OOD data.

Figures 4 and 5 visualizes the differential entropy distribution of ID data and OOD data under label attacks. On CIFAR10, PriorNet and DDNet can barely distinguish between clean ID and OOD data. We observe a better ID/OOD distinction for PostNet and EvNet for clean data. However, we do not observe for any model an increase of the uncertainty estimates on label attacked data. Even worse, PostNet, PriorNet and DDNet seem to assign higher confidence on class label attacks. On MNIST, models show a slightly better behavior. They are capable to assign a higher uncertainty to label attacks up to some attack radius.

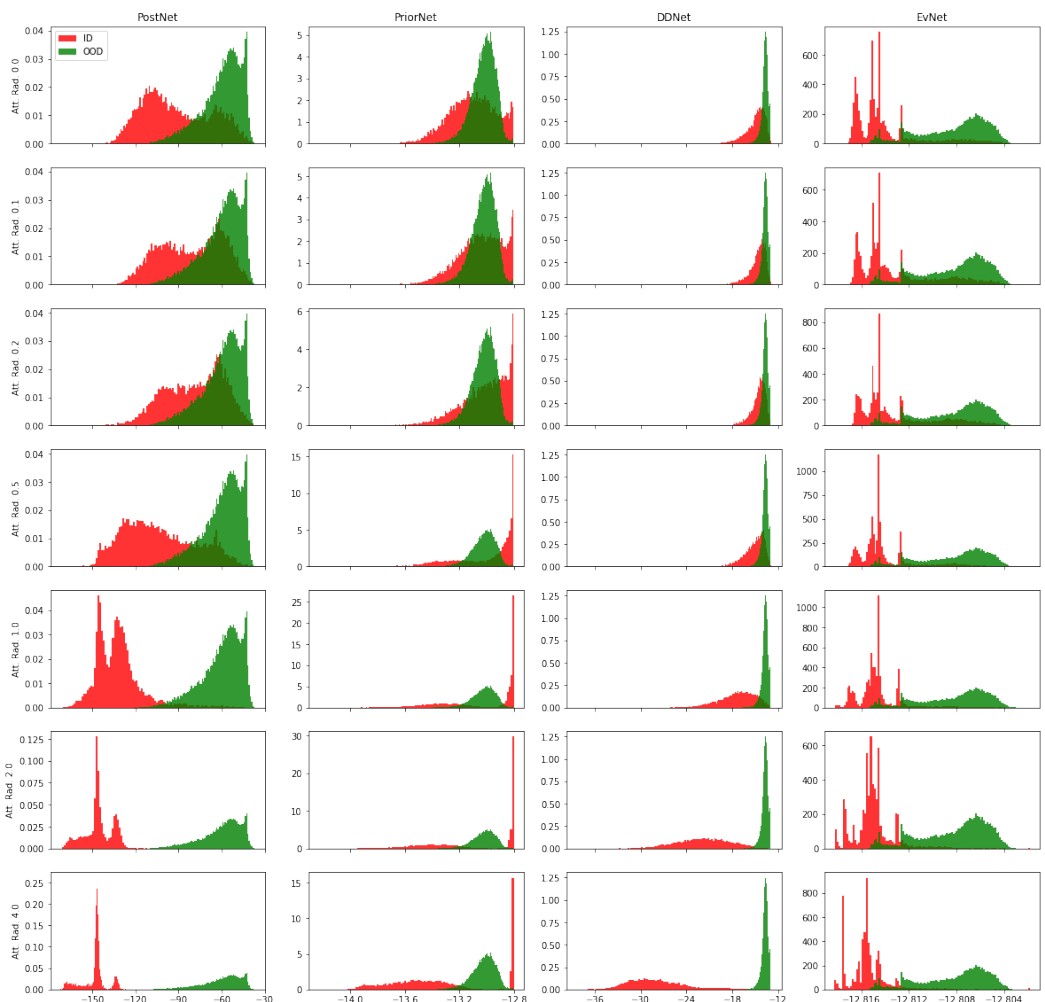

Figure 4: Visualization of the differential entropy distribution of ID data (CIFAR10) and OOD data (SVHN) under label attack. The first row corresponds to no attack. The other rows correspond do increasingly stronger attack strength.

Figures 6, 7, 8 and 9 visualizes the differential entropy distribution of ID data and OOD data under uncertainty attacks. For both CIFAR10 and MNIST data sets, we observed that uncertainty estimations of all models can be manipulated. That is, OOD uncertainty attacks can shift the OOD uncertainty distribution to more certain predcitions, and ID uncertainty attacks can shift the ID uncertainty distribution to less certain predictions.

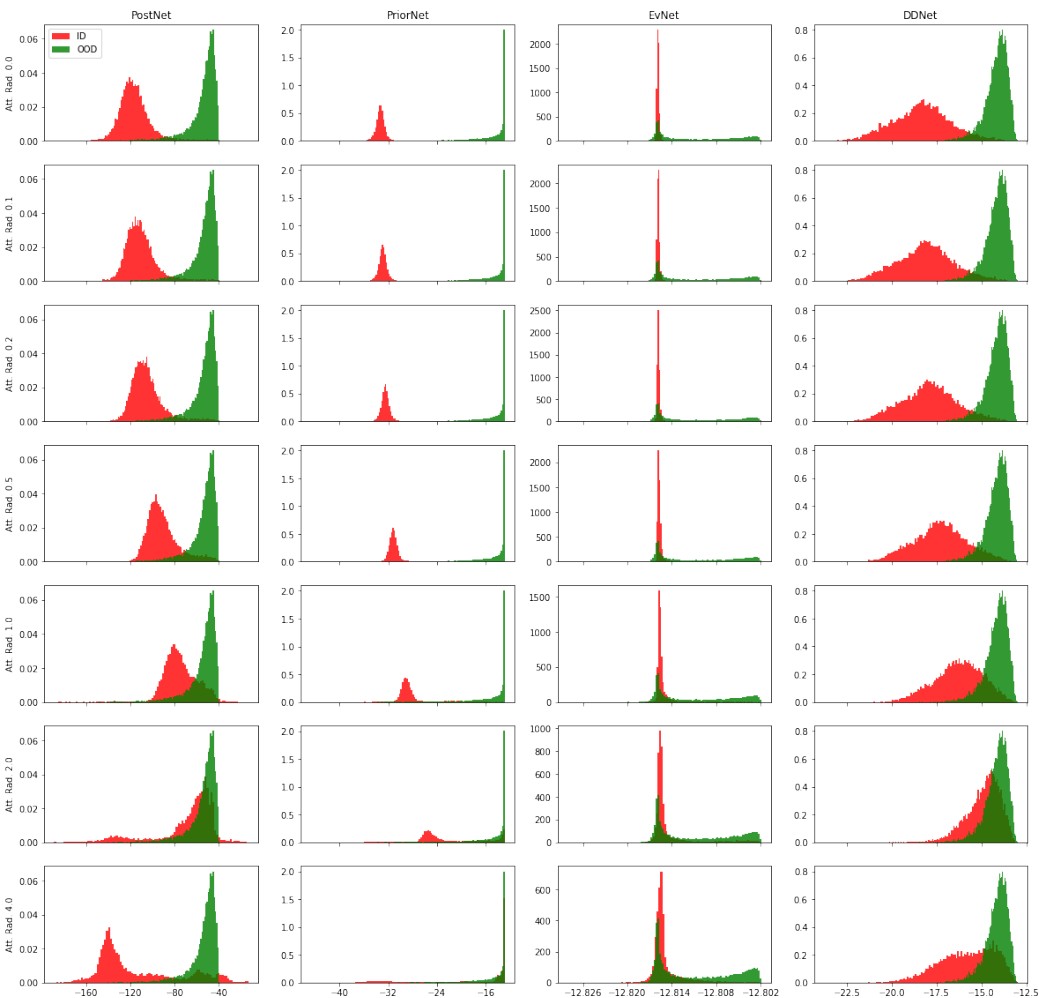

Figure 5: Visualization of the differential entropy distribution of ID data (MNIST) and OOD data (KMNIST) under label attack. The first row corresponds to no attack. The other rows correspond do increasingly stronger attack strength.

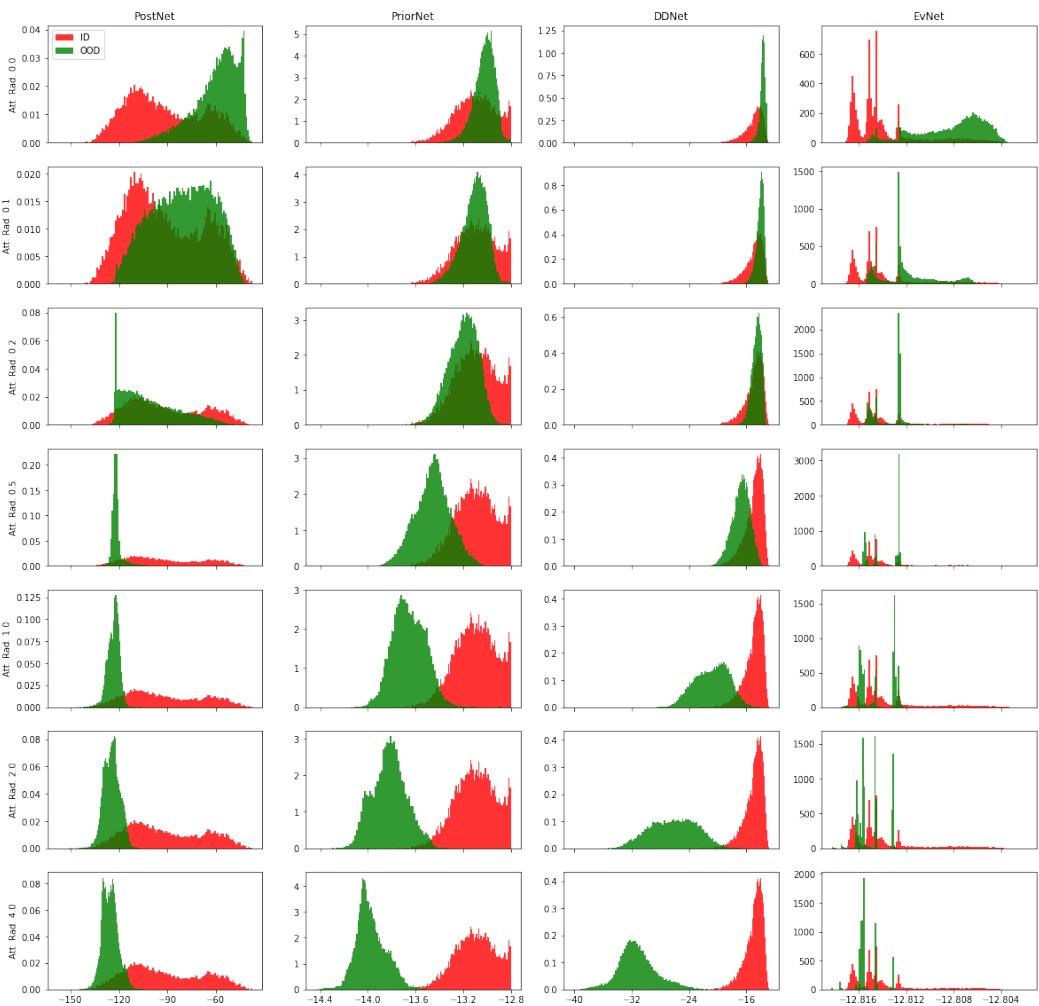

Figure 6: Visualization of the differential entropy distribution of ID data (CIFAR10) and OOD data (SVHN) under OOD uncertainty attack. The first row corresponds to no attack. The other rows correspond do increasingly stronger attack strength.

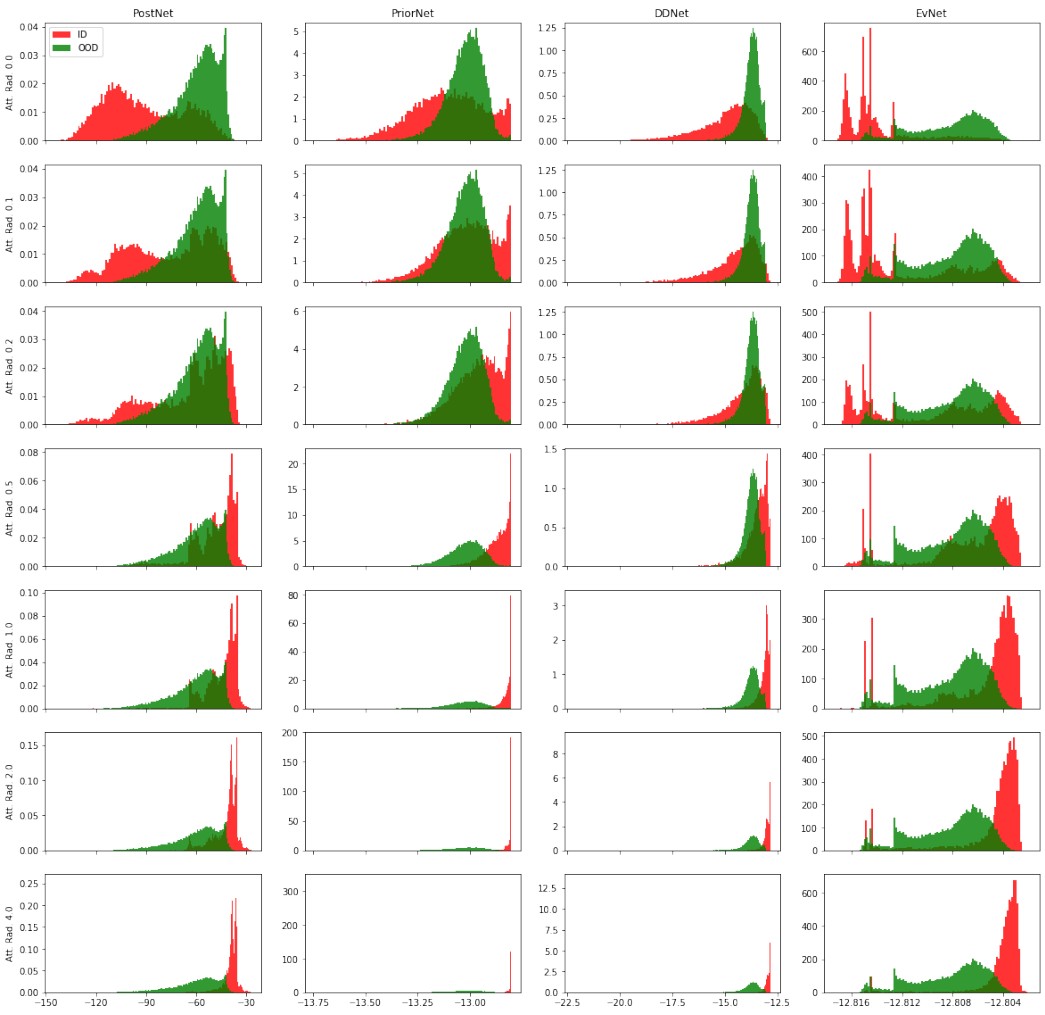

Figure 7: Visualization of the differential entropy distribution of ID data (CIFAR10) and OOD data (SVHN) under ID uncertainty attack. The first row corresponds to no attack. The other rows correspond do increasingly stronger attack strength.

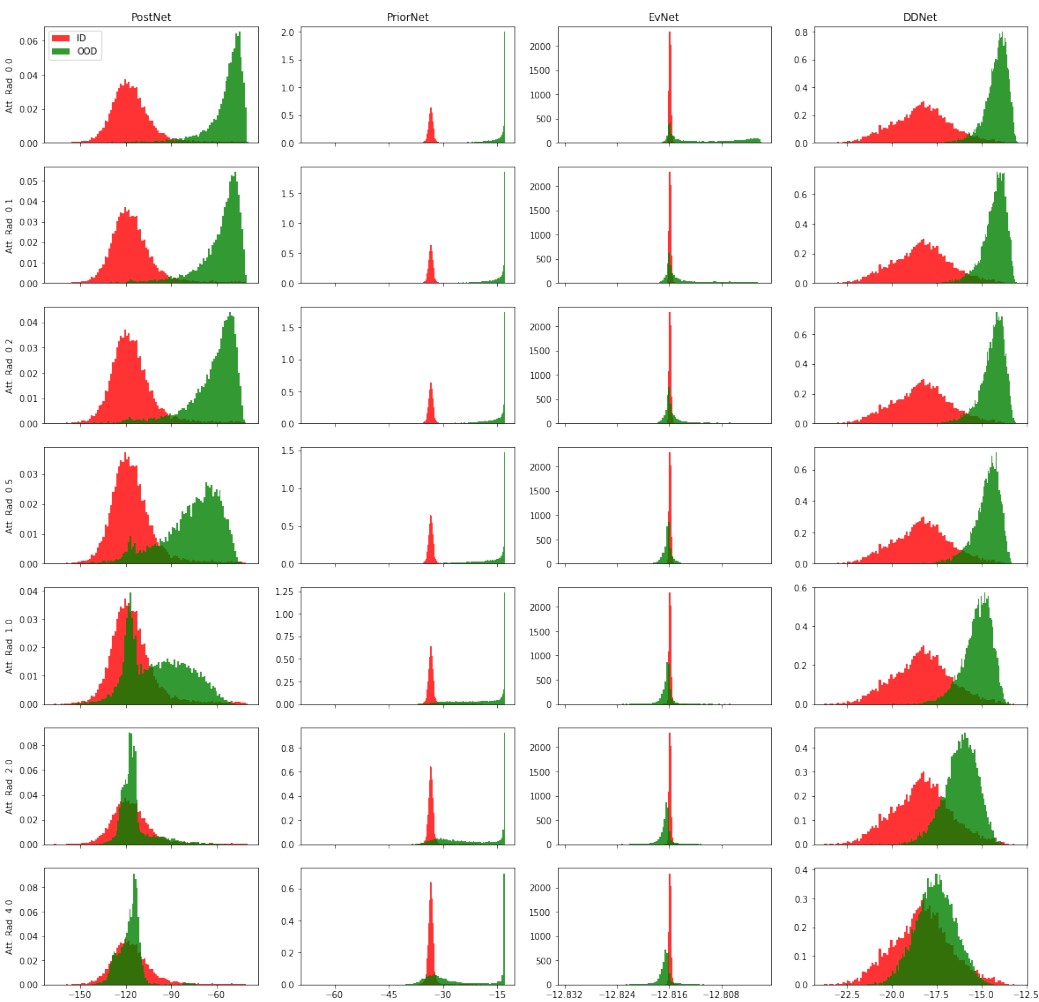

Figure 8: Visualization of the differential entropy distribution of ID data (MNIST) and OOD data (KMNIST) under OOD uncertainty attack. The first row corresponds to no attack. The other rows correspond do increasingly stronger attack strength.

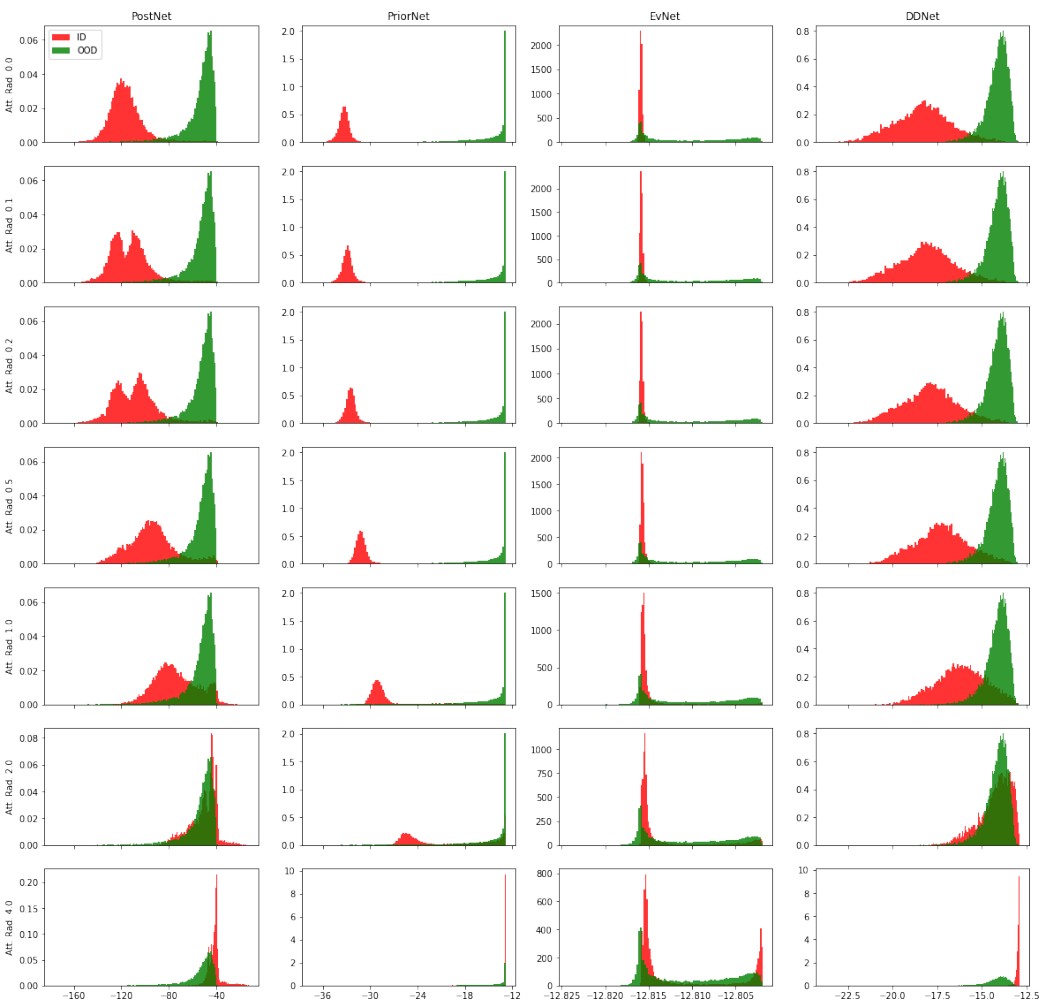

Figure 9: Visualization of the differential entropy distribution of ID data (MNIST) and OOD data (KMNIST) under ID uncertainty attack. The first row corresponds to no attack. The other rows correspond do increasingly stronger attack strength.

