# OpenReview forum: "Evaluating Robustness of Predictive Uncertainty Estimation: Are Dirichlet-based Models Reliable?"
_ICLR.cc/2021/Conference — Reject_

### Official Review · AnonReviewer3 · 2020-10-26
**Connecting Uncertainty Measures to Adversarial Perturbations**

**Rating:** 5
**Confidence:** 2

**Review:**

This manuscript addresses an important question of how Drichlet-based uncertainty (DBU) measures can be used to quantify robustness to adversarial label attacks. Robustness to OOD samples of these models were already shown in the papers they were proposed but this work differs from them in using adversarial samples as OOD samples. Via extensive experimental results on various datasets, the authors conclude that the uncertainty estimates are not good indicators for identifying correctly classified samples for adversarially perturbed data.

Although I find this contribution valuable, I am still not sure how generalizable the conclusions are. My comments/questions are below:

- What makes adversarial samples so different than other OOD samples? In other words, how come these measures are good indicators in detecting different OOD samples as reported in Malinin & Gates, 2018a but fail in adversarial samples.

- I am surprised that the authors did not mention calibration in the context of uncertainty estimation. Is it possible to relate these findings to calibration?

- I understand that the main focus of this work is Dirichlet-based models but I wonder if it is possible to generalize these findings to other families of models such as deep ensembles (Lakshminarayanan et al., 2017)?

- Figure 1 was never mentioned in text.

- It is not clear what y-axis is in Figure 2. Is it 1-uncertainty?

- I am skeptical about the results on tabular data shown in Table 4 for PostNet. How is possible the detection rate becomes almost perfect for the strongest attack? The authors mention that it is due to density estimation but it is not clear to me how it is possible.

- For the results in section 4.3, I do not understand why the authors decided to report results based on a threshold instead of computing AUCPR as was done in other sections.

---

> ### Author Response · Authors · 2020-11-19
> **Answer to Reviewer 3**
>
> Dear Reviewer 3,
>
> Thank you for the detailed feedback and your comments.
>
> Q1: What makes adversarial samples so different than other OOD samples? In other words, how come these measures are good indicators in detecting different OOD samples as reported in Malinin & Gates, 2018a but fail in adversarial samples.
>
> A1: Adversarial examples are specifically designed to fool the models which make them usually harder to detect than OOD samples. In particular, an adversarial example should be very close to the original sample contrary to OOD samples.
>
> Q2: I am surprised that the authors did not mention calibration in the context of uncertainty estimation. Is it possible to relate these findings to calibration?
>
> A2: Calibration scores evaluate if the predicted probabilities are aligned with the real probability that the prediction is correct. The certainty score addresses the following, similar question "Is high certainty a reliable indicator of correct predictions?". We show (see Table 2 and appendix) that the models do not achieve good certainty scores. Thus, a good calibration of the models is very unlikely. Note that models which achieve an accuracy close to 0 cannot be calibrated since the max predicted probability has to be strictly larger than 0, i.e. max_c p_c >= 1/C >0.
>
> Q3: I understand that the main focus of this work is Dirichlet-based models but I wonder if it is possible to generalize these findings to other families of models such as deep ensembles (Lakshminarayanan et al., 2017)?
>
> A3: Please have a look at the sections “Choice of models” in the comment addressed to all reviewers.
>
> Q4: I am skeptical about the results on tabular data shown in Table 4 for PostNet. How is possible the detection rate becomes almost perfect for the strongest attack? The authors mention that it is due to density estimation but it is not clear to me how it is possible.
>
> A4: In (Charpentier et al., 2020), PostNet shows particularly good performances for OOD detection on tabular datasets. This is due to the density estimation performed by the model on the ID training data. This allows to assign low density to data far from ID training data in the unbounded input space. As a stronger perturbation allows to push away the attacked sample further from the original ID sample, PostNet mechanically assigns a lower density to attacks with larger magnitudes, and thus detects them as OOD data.
>
> Q5: For the results in section 4.3, I do not understand why the authors decided to report results based on a threshold instead of computing AUCPR as was done in other sections.
>
> A5: In section 4.3, we verify if an input is robustly classified as ID/OOD data based on randomized smoothing which requires an explicit decision boundary. To this end, we have to define a threshold on the uncertainty measure (i.e. differential entropy) used for ID/OOD classification. Note that using a threshold is a very practical setting.

---

### Official Review · AnonReviewer2 · 2020-10-28
**A nice contribution for evaluation/quantification of uncertainty**

**Rating:** 7
**Confidence:** 4

**Review:**

In this work, the authors seek to evaluate the robustness and uncertainty of Dirichlet-Based Uncertainty models (DBUs). DBUs are models which rather than outputting a multinoulli distribution (as is the case with DNNs and BNNs) output the parameters of a Dirichet distribution. The authors clearly state and review the history and benefits of such models. While models with calibrated uncertainty are becoming increasingly popular for addressing some of the fundamental shortcomings of standard DNNs, there has been less work in evaluating the relationship between robustness and uncertainty and quantifying the robustness of this uncertainty. This paper make a valuable contribution in that it performs a systematic analysis of the robustness and uncertainty of DBUs. I think the authors evaluation framework is thorough and well-motivated and can serve as a template for how developers of methods which seek to intrinsically capture uncertainty should benchmark their performances wrt adversaries.

The one minor weak point of the paper is some of its contextualizations with the literature. For example, there are several papers (albeit very recently published ones) which I think it would benefit the authors to reference. In [1] the authors analyze the robustness of a model which outputs the parameters of a Gaussian distribution and find that directly using the parameters of the outputted Gaussian can lead to a strong attack. In [2,3] the authors perform statistical and probabilistic certification of Bayesian neural networks which is strongly related to the claim that the authors are the first to certify methods for uncertainty estimation models. Of course, there is a distinction in the approaches, but it is one the authors should probably make explicit for completeness.

Also, from a presentation/structural point of view, the authors restate the “assessment metric” paragraph essentially verbatim 3-4 times in the paper and I think perhaps having one global explanation and then discussing the small changes in each subsequent section would make the paper much easier to read. That being said, I do like the format of the presented results, I would just cut down on the redundancy.

[1] - https://arxiv.org/pdf/2003.03778.pdf [ICML]
[2] - https://www.ijcai.org/Proceedings/2019/789 [IJCAI]
[3] -http://proceedings.mlr.press/v124/wicker20a.html [UAI]


Post Rebuttal Response:

I would like to thank the authors for considering my review and for making some of the suggested edits. I think this paper provides a valuable contribution and point of discussion for those interested in the interplay between robustness and uncertainty in deep learning. I do consider the experimental evaluation in this work to be sufficient given that the authors consider many applications which already exist in the literature, and in my view it is out of the scope of an evaluation/methodology paper to necessarily advance the state-of-the-art in applications of the method they seek to evaluate.

As I have no major standing criticism of this work, and believe that it provides a useful and interesting contribution to the conference I have increased my score.

---

> ### Author Response · Authors · 2020-11-19
> **Answer to Reviewer 2**
>
> Dear Reviewer 2,
>
> Thank you for the feedback and your comments.
>
> Q1: Could you provide a better contextualization with the literature ?
>
> A1: We improved the contextualization (see section 2) with the literature and added the suggested literature.
>
> Q2: Could you cut down the redundancy in the paper format ?
>
> A2: We improved the paper structure and cut down redundancy in “assessment metric” paragraphs.

---

### Official Review · AnonReviewer1 · 2020-10-28
**Official Blind Review #1 - too narrow in scope and experiments too limited for negative results to be valuable enough**

**Rating:** 2
**Confidence:** 5

**Review:**

The authors present an analysis of previously proposed Dirichlet based models for adversarial robustness and empirically evaluate exiting methods on two image datasets, MNIST and CIFAR10, as well as 2 tabular datasets.
While in principle adversarial robustness is an interesting topic, the scope of the paper is extremely narrow and I would have liked to see a broader set of Bayesian models being included.
In addition, I find the set of experiments very limiting. MNIST is not a representative dataset at all and for a meaningful comparison analysing a large-scale dataset (such as Imagenet) is crucial. Furthermore, I would have liked to see a broader type of data - how about a text dataset (such as 20 newsgroup) or a different sequential dataset with a recurrent architecture? I feel with the limited  experiments is not a very useful resource for practitioners. Also, performance for other attacks (deepfool, black-box attack) would have been interesting.
Even with the very limited set of experiments, the authors basically report negative results, showing that neither of the approaches could detect adversarial attacks, OOD samples or highly perturbed data
The authors also propose a robust training strategy, but concede that this increases performance either for either ID data or OOD data, but not both.

While in principle also such negative results can be valuable for the community, I feel in this case the scope of the paper is too narrow and a broader class of Bayesian and ideally non-Bayesian but uncertainty-aware methods should have been analysed.

---

> ### Author Response · Authors · 2020-11-19
> **Answer to Reviewer 1**
>
> Dear Reviewer 1,
>
> Thank you for the feedback and your comments.
>
> Please have a look at the sections “Choice of models” and “Choice of attacks” in the comment addressed to all reviewers.
>
> **Choice of data sets**
>
> First, all DBU models are proposed for classification on standard data sets such as the ones we used in our work. As we wanted each DBU to achieve high performance, we chose the same/similar data sets (and architecture) as the ones proposed in the corresponding papers, i.e.:
>
> - MNIST (images)
> - CIFAR10 (images)
> - Segment (tabular data)
> - Sensorless (tabular data)
>
> The additional data sets you suggested comprise sequential data and text data (i.e. discrete data).
> These data sets are special and require addressing questions such as “What is an adversarial attack on text data?” first (which is an own research focus).
> Second, uncertainty estimation is not robust, even on basic, standard data sets such as the ones we used. While some models showed encouraging positive results on weaker FGSM attacks (e.g. good attack detection, see appendix Table 18), we showed that stronger PGD attacks were sufficient to fool all models and metrics without being detected. Therefore, we viewed it as unnecessary to consider a more complex settings since it is very unlikely that the models are more robust on such settings.
>
> As described in the general answer, we analyze robustness of all state-of-the-art DBU models on many different combinations of attack types, measures and data sets. Our results are presented by 56 Tables, requiring an appendix of 26 pages. We believe that this is an exhaustive robustness analysis of DBU models.

---

### Official Review · AnonReviewer4 · 2020-10-28
**Recommendation to Accept**

**Rating:** 6
**Confidence:** 3

**Review:**

The paper focuses on quantifying uncertainty for classification problems using Dirichlet based uncertainty (DBU) estimation techniques. The authors study these techniques for their robustness properties under adversarial attacks, proposes a novel attack type targeting uncertainty estimates through differential entropy, and investigates robust training for detecting in and out of distribution data points. Experiments using image datasets showed that uncertainty from DBU models 1) do not provide robust identification of correct predictions under adversarial attacks, 2) are only able to detect weak attacks and do not perform well under strong attacks, 3) robust training does not guarantee generalization to both in and out of distribution datasets.

Overall, I vote for accepting the paper. I like the idea of using uncertainty estimation for adversarial machine learning. My major concern is the limited scope of the paper and the generalization of the results. Hopefully, the authors can address my concerns in the rebuttal period.

Positives:
-	The paper is nicely structured and the organization of the experiments was easy to follow.
-	The experiments were nicely guided by the research questions. Even though the observed results were surprising and do not match the expected behaviors, nonetheless, the observations are important to guide future research in this direction.
-	The idea of attacking the uncertainty estimates is novel and provides an interesting research direction.

Negatives:
-	The scope of the uncertainty estimation to Dirichlet based uncertainty estimation techniques was limited. The authors already cited prominent techniques in the paper, ranging from MC Dropout to ensembles and including the results using these techniques would have made the results more compelling. This will also help generalize the results to the using predictive uncertainty techniques in the context of adversarial robustness.
-	The scope of the adversarial attacks employed in this study were also limited. Even though the authors claim that using uncertainty from DBU models for identifying correct estimates do not produce robust estimates under adversarial attacks, the experiments are performed only using PGD attacks.
-	The uncertainty attacks provide an interesting contribution but I found the section describing the proposed attacks very limited. I would suggest the authors elaborate more on this in the manuscript or in an appendix. Furthermore, I also recommend that the authors elaborate more on the performance of the uncertainty attacks in general and compare success probabilities to the state-of-the-art adversarial attacks.

Minor comments:
-	Please indicate the best results among the methods used in the tables by highlighting them.
-	In figures 2 and 3, please provide the axes labels.
-	I would suggest defining differential entropy in the manuscript for completeness since it is used throughout the paper.
-	I would suggest adding a brief summary of the conclusions in the abstract.

---

> ### Author Response · Authors · 2020-11-19
> **Answer to Reviewer 4**
>
> Dear Reviewer 4,
>
> Thank you for the detailed feedback and your comments.
>
> Q1: The set of considered uncertainty models is limited.
>
> A1: Please have a look at the section “Choice of models” in the general answer.
>
> Q2: The set of considered attacks is limited.
>
> A2: Please have a look at the section “Choice of attacks” in the general answer.
>
> Q3: Could you elaborate more in the manuscript on the proposed uncertainty attacks?
>
> A3: We updated the manuscript to clarify the proposed uncertainty attacks (see Appendix A2).

---

### Author Response · Authors · 2020-11-19
**General Answer**

Dear Reviewers,

thank you for the detailed feedback and your comments. First, we want to address the main questions.

**Choice of models**

We focus on Dirichlet based uncertainty estimation (DBU) models because of their two major benefits: First, DBU models estimate several types of uncertainty by one pass through the neural network. Thus, in contrast to Bayesian neural networks, Monte-Carlo drop-out and ensemble, they are very efficient. Second, DBU models provide sophisticated uncertainty estimates that go beyond simple confidence (which makes them beneficial in comparison to calibration techniques).
Our evaluation includes *all* state-of-the-art approaches from the DBU family, i.e.:

- Prior Networks [1,2]
- Ensemble Distribution Distillation Networks [3]
- Posterior Networks [4]
- Evidential Networks [5]

As these models differ in various aspects such as training, density estimation, use of auxiliary data sets during training and knowledge generation (ensemble knowledge distillation) our evaluation is exhaustive. Including models from other families would result in an overloaded paper.

**Choice of attacks**

The goal of this paper is to evaluate whether Dirichlet-based uncertainty estimation is robust.
Thus, we analyzed model performance under attacks of increasing strength until attacks fooled the models completely. In particular, our exhaustive attack setting covers:

- 2 Attack types: FGSM, PGD
- 4 Attack losses: cross entropy, differential entropy, mutual information, alpha0
- 6 Attack radii: 0.1, 0.2, .5, 1., 2., 4.
- 9 Evaluation metrics: accuracy, certainty (AUC-ROC, AU-PR), attack detection (AUC-ROC, AU-PR), OOD scores under ID and OOD attacks (AUC-ROC, AU-PR).

This setting is tested for all 4 models (averaged over 5 different initializations) on 4 data sets of different complexity. While some models showed encouraging positive results on weaker FGSM attacks (e.g. good attack detection), we showed that stronger PGD attacks were sufficient to fool all models and metrics without being detected. Therefore, we viewed it as unnecessary to consider stronger attacks on more complex settings since it is very unlikely that the models are more robust in such settings.

Nonetheless, since multiple reviewers suggested to consider additional attacks, we performed a black-box attack based on random noise sampling. This black-box attack can fool the model accuracy, certainty and OOD scores but turns out to be detectable (similarly to FGSM attacks, see Appendix).

In summary we analyze robustness of all state-of-the-art DBU models, trained on four different data sets (image data as well as tabular data) using attacks against four different measures based on three attack types (FGSM, PGD and the recently added random noise black-box setting) as well as corresponding verification approaches. These results are presented in detail by 56 Tables and eight Figures, requiring an appendix of 26 pages. We believe that this is an exhaustive robustness analysis of DBU models.

We updated the paper with new results, your minor comments and proposed literature.


[1] Andrey Malinin and Mark Gales, Predictive Uncertainty Estimation via Prior Networks, Neural Information Processing Systems, 2018.

[2] Andrey Malinin and Mark Gales, Reverse KL-Divergence Training of Prior Networks: Improved Uncertainty and Adversarial Robustness, Advances in Neural Information Processing Systems, 2019.

[3] Andrey Malinin, Bruno Mlodozeniec and Mark Gales, Ensemble Distribution Distillation, International Conference on Learning Representations, 2019.

[4] Bertrand Charpentier, Daniel Zügner and Stephan Günnemann, Posterior Network: Uncertainty Estimation without OOD Samples via Density-Based Pseudo-Counts, Advances in Neural Information Processing Systems, 2020.

[5] Murat Sensoy, Lance Kaplan and Melih Kandemir, Evidential deep learning to quantify classification uncertainty, Advances in Neural Information Processing Systems, 2018.

---

### Decision · Program_Chairs · 2021-01-07
**Final Decision**

**Decision:**

Reject

**Comment:**

All reviewers except for AnonReviewer1 were in favour of accept.  AnonReviewer1 was strongly in favour of reject, but AnonReviewer2 argued against some of AnonReviewer1's opinion.  The authors also gave a coherent, well argued statement of their contribution.  Nevertheless, there are some improvements still needed.

Position:   the scope of the uncertainty estimation to Dirichlet based uncertainty estimation techniques was limited.

Sticking to Dirichlet-based uncertainty is limited, although the coverage of methods within the Dirichlet-based family is OK but could be improved. Note (from AnonReviewer1's comments) Joo Chung and Seo, ICML 2020, is one paper that should be included and Chan, Alaa, and van der Schaar, ICML2020 is also relevant.  While its not about adversial attacks it covers a related idea with a good technique.   Finally, these papers cite Ovadia, Fertig Ren etal. NeurIPS 2019, which is an excellent summary of calibration and estimation under shift, not exactly adversarial attacks but surely related.  The big winner is deep ensembles (Lakshminarayanan etal, NeurIPS 2017).  I think using deep ensembles directly would be a good complement to the Dirichlet methods in this paper.
Note, also, the authors already included additional works mentioned by AnonReviewer2.

Critique:   The authors proposed a robust training strategy but this didn't lead to uniform improvement.

Position:  The scope of the adversarial attacks is limited.

The attacks covered are a good though basic range.  But because these show problems, the argument is that more sophisticated attacks do not need to be studied.

Position:  The datasets covered is limited.

Certainly, there are problems with extending experiments to text data.  But the argument is that if things don't work well for the smaller datasets given, then that is still a problem, so why bother extending the evaluation to larger datasets.

Arguably, the latter two positions have been addressed by the authors, but not the first two.  This makes the paper marginal.
So this is a good publishable paper, but comparatively marginal.